# Accelerating data-driven algorithm selection for combinatorial partitioning problems

**Vaggos Chatziafratis**
UC Santa Cruz
vaggos@ucsc.edu

**Ishani Karmarkar**
Stanford University
ishanik@stanford.edu

**Yingxi Li**
Stanford University
yingxi@stanford.edu

**Ellen Vitercik**
Stanford University
vitercik@stanford.edu

## Abstract

*Data-driven algorithm selection* is a powerful approach for choosing effective heuristics for computational problems. It operates by evaluating a set of candidate algorithms on a collection of representative *training instances* and selecting the one with the best empirical performance. However, running each algorithm on every training instance is computationally expensive, making scalability a central challenge. In practice, a common workaround is to evaluate algorithms on smaller proxy instances derived from the original inputs. However, this practice has remained largely ad hoc and lacked theoretical grounding. We provide the first theoretical foundations for this practice by formalizing the notion of *size generalization*: predicting an algorithm's performance on a large instance by evaluating it on a smaller, representative instance, subsampled from the original instance. We provide size generalization guarantees for three widely used clustering algorithms (single-linkage, $k$-means++, and Gonzalez's $k$-centers heuristic) and two canonical max-cut algorithms (Goemans-Williamson and Greedy). We characterize the subsample size sufficient to ensure that performance on the subsample reflects performance on the full instance, and our experiments support these findings.

## 1 Introduction

Combinatorial partitioning problems such as clustering and max-cut are fundamental in machine learning [21, 22, 39, 40], finance [26], and biology [38, 74]. These problems are typically NP-hard, which has prompted the development of many practical approximation algorithms and heuristics. However, selecting the most effective heuristic in practice remains a key challenge, and there is little theoretical guidance on the best method to use on a given problem instance or application domain. In clustering, for example, Ben-David [22] notes that algorithm selection is often performed "in a very ad hoc, if not completely random, manner," which is unfortunate "given [its] crucial effect" on the resulting clustering.

To address this challenge, *data-driven algorithm selection* has emerged as a powerful approach, offering systematic methods to identify high-performing algorithms for the given domain [28, 36, 63, 73, 84]. Gupta and Roughgarden [52] provided a formal PAC-learning framework for data-driven algorithm selection, where we are given candidate algorithms $\mathcal{A}_1, ..., \mathcal{A}_K$ and training problem instances $\mathcal{X}_1, ..., \mathcal{X}_N$ from the application domain. A *value function* value evaluates the algorithm's output quality—for example, for max-cut, this is the density of the returned cut whereas for clustering, this is the quality of the resultant clustering. The goal is then to select $\mathcal{A}^\star \in \{\mathcal{A}_1, ..., \mathcal{A}_k\}$ which

39th Conference on Neural Information Processing Systems (NeurIPS 2025).

maximizes the empirical performance:

$$\max_{j \in [K]} \sum_{i \in [N]} \mathsf{value}(\mathcal{A}_j(\mathcal{X}_i)). \tag{1}$$

When $N$ is sufficiently large, *generalization* bounds imply that $\mathcal{A}^\star$ approximately maximizes the expected value on future *unseen* problems from the same application domain [e.g., 12, 14, 18, 52].

However, a key *computational* challenge in solving Equation (1) is that it requires running each algorithm $\mathcal{A}_j$ on every training instance $\mathcal{X}_i$. Since the algorithms $\mathcal{A}_j$ themselves often have super-linear runtime, this process quickly becomes prohibitively expensive, especially when $N$ and the underlying training instances $\mathcal{X}_i$ are large. Indeed, a major open direction is to design computationally efficient data-driven algorithm selection methods [15, 24, 53].

To help address this bottleneck, we introduce a new notion of *size generalization* for algorithm selection:

**Question 1.1** (Size generalization). *Given an algorithm $\mathcal{A}$ and problem instance $\mathcal{X}$ of size $n$, can we quickly estimate* $\mathsf{value}(\mathcal{A}(\mathcal{X}))$ *using a small, representative problem instance of size $m \ll n$?*

Beyond enabling efficient data-driven algorithm selection, size generalization has broader implications, even in the absence of training instances. For example, when faced with a single large problem instance, Question 1.1 provides a natural framework for efficiently selecting a suitable algorithm by evaluating candidate algorithms' performance on a smaller, representative subsample.

**Motivation from prior empirical work.** In discrete optimization, a common heuristic for algorithm selection on large-scale problems is to evaluate candidate algorithms on smaller, representative instances [e.g., 31, 35, 46, 51, 56–59, 77–79, 81]. Despite its widespread use, this approach remains ad hoc, lacking theoretical guarantees. We provide the first theoretical foundations for this practice.

## 1.1 Our contributions

To move towards a general theory of size generalization, we answer Question 1.1 for two canonical combinatorial problems: clustering and max-cut. For clustering, we analyze three well-studied algorithms: single-linkage clustering, Gonzalez's $k$-centers heuristic, and $k$-means++. For max-cut, we analyze the classical Goemans-Williamson (GW) semi-definite programming (SDP) rounding scheme and the Greedy algorithm [47, 64]. Finally, in Section 4, we empirically validate our theoretical findings. An exciting direction for future work is to extend this framework to other combinatorial algorithm selection problems, such as integer or dynamic programming [13, 14, 52].

A key challenge in answering Question 1.1 lies in the sequential nature of combinatorial algorithms, which makes them highly sensitive to individual data points in clustering (Figures 2 and 9 in Appendix B.2) and specific nodes in max-cut (Figures 4 and 5 in Appendix B.2.2). This sensitivity makes it difficult to identify conditions under which an algorithm's performance is robust to subsampling.

**Clustering algorithm selection.** In Section 2, we study clustering algorithm selection in the widely-studied *semi-supervised* setting [12, 19, 30, 62, 85], where each clustering instance $\mathcal{X}$ has an underlying *ground truth* clustering and $\mathsf{value}(\cdot)$ measures accuracy with respect to this ground truth. While the ground truth is unknown, it can be accessed through expensive oracle queries, which model interactions with a domain expert. A key advantage of data-driven clustering algorithm selection is that ground-truth information is *only* required for the training instances in Equation (1). Prior research bounds the number of training instances sufficient to ensure that in expectation over new clustering instances from the application domain—where the ground truth is unavailable—the selected algorithm $\mathcal{A}^*$ provides the best approximation to the unknown ground truth [12, 13].

We formalize the goal of size generalization for clustering algorithm selection as follows.

**Goal 1.2** (Size generalization for clustering). *Estimate the accuracy of a candidate clustering algorithm on an instance $\mathcal{X}$ by running the algorithm on a subsampled instance $\mathcal{X}' \subset \mathcal{X}$ of size $|\mathcal{X}'| \ll |\mathcal{X}|$. The selection procedure should have low runtime and require few ground-truth queries.*

We present size generalization bounds for center-based clustering algorithms, including a variant of Gonzalez's $k$-centers heuristic and $k$-means++, as well as single-linkage clustering. For the

center-based algorithms, we show that under natural assumptions, the sample size sufficient to achieve Goal 1.2 is independent of $|\mathcal{X}|$. Meanwhile, the single-linkage algorithm is known to be unstable in theory [29], yet our experiments reveal that its performance can often be estimated from a subsample. To explain this robustness, we characterize when size generalization holds for single-linkage, clarifying the regimes in which it remains stable or becomes sensitive to deletions.

*Comparison to coresets.* Our clustering results complement research on coresets [e.g., 33, 42, 54], but differ fundamentally and are not directly comparable. Given a clustering instance $\mathcal{X}$ with the goal of finding $k$ centers that minimize a *known objective $h$* (such as the $k$-means objective), a coreset is a subset $\mathcal{X}_c \subset \mathcal{X}$ such that for any set $C$ of $k$ centers, $h(C; \mathcal{X}) \approx h(C; \mathcal{X}_c)$. Coresets enable efficient approximation algorithms for minimizing $h$. However, we aim to minimize *misclassification* with respect to an *unknown* ground truth clustering. Thus, coresets do not provide any guarantees in our setting. We make no assumptions on the ground truth clustering (e.g., that it minimizes some objective or is approximation-stable [10])—it is arbitrary and may even be adversarial with respect to $\mathcal{X}$—and we provide approximation guarantees *directly* with respect to the ground truth.

**Max-cut algorithm selection.** In Section 3, we study max-cut—a problem with practical applications in circuit design [20], physics [16], and data partitioning [72]—with the following goal:

**Goal 1.3** (Size generalization for max-cut). *Estimate the performance of a candidate max-cut algorithm on a graph $G = (V, E)$ using a vertex-induced subgraph $G[S]$ for $S \subset V$ with $|S| \ll |V|$.*

Selecting between max-cut algorithms is non-trivial because competing methods differ sharply in solution quality and runtime. The GW algorithm outperforms the Greedy algorithm in many instances, but often, the reverse is true. Even when both produce cuts of comparable weights, Greedy has a significantly faster runtime, since GW solves the max-cut SDP relaxation—a computationally-expensive task—before performing randomized hyperplane rounding. Data-driven algorithm selection helps determine when GW's higher computational cost is justified.

To analyze GW, we prove convergence bounds for the SDP objective values under $G$ and $G[S]$, independent of graph structure. As such, we strengthen prior work by Barak et al. [17], who provided GW SDP convergence guarantees only for random geometric graphs. We then show that the cut density produced by the GW algorithm on $G$ can be estimated using the subgraph $G[S]$. For Greedy, we build on prior work [64] to bound the rate at which the cut density of $G[S]$ converges to $G$.

## 1.2 Additional related work

Ashtiani and Ben-David [5] study a semi-supervised approach to learning a data representation for clustering tasks. In contrast, we study algorithm selection. Voevodski et al. [82] study how to recover a low-error clustering with respect to a ground-truth clustering using few distance queries, assuming the instance is "approximation stable" [10]—meaning any clustering that approximates the $k$-medians objective is close to the ground truth. We make no such assumptions.

Our Max-Cut results complement theoretical work on estimating Max-Cut density using random vertex-induced subgraphs or coresets, with the aim of developing an accurate polynomial-time approximation or achieving a fast sublinear-time approximation [1–3, 23, 43, 64, 71, 83]. We build on this research in our analysis of Greedy, but these results cannot be used to analyze the GW algorithm. Rather, we obtain our results by proving general guarantees for approximating the GW SDP using a subsample. Barak et al. [17] provide similar guarantees for a broad class of SDPs, which includes the GW SDP; however, their results for max-cut rely on strong structural assumptions on the graph. In contrast, our results for the GW SDP do not require any such graph structure assumptions. We discuss this further in Appendix B. Beyond these works, there has been related work on sketch-and-solve approximation algorithms for clustering via SDP relaxations, which is in a similar spirit to ours but in a different setting [32, 68].

## 1.3 General notation

We use $x \approx_\epsilon y$ for the relation $y - \epsilon \leq x \leq y + \epsilon$. We define $\Sigma_n$ to be the permutations of $[n] := \{1, ..., n\}$. Given event $E$, $\bar{E}$ is the complement, and $\mathbb{1}(E)$ is the indicator of $E$.

**Input:** Instance $\mathcal{X} = (x_1, ..., x_{mk}) \subset \mathbb{R}^d$, $k, m \in \mathbb{Z}_{\geq 0}$, $f : \mathbb{R} \times \mathcal{P}(\mathcal{X}) \to \mathbb{R}_{\geq 0}$
Set $C^1 = \{c_1\}$ with $c_1 \sim \text{Unif}(\mathcal{X})$ and $\ell = 1$    // $\ell$ is a counter for iterating over $\mathcal{X}$
**for** $i = 2, 3, \ldots, k$ **do**
$\quad$ Set $x = x_\ell$, $d_x = d_{\text{center}}(x; C^{i-1})$, $\ell = \ell + 1$    // $x$ is a candidate cluster center
$\quad$ **for** $j = 2, 3, \ldots, m$ **do**
$\quad\quad$ Set $y = x_\ell$, $d_y = d_{\text{center}}(y; C^{i-1})$, $\ell = \ell + 1$    // $y$ is a fresh sample
$\quad\quad$ **if** $f(d_y; \mathcal{X})/f(d_x; \mathcal{X}) > z \sim \text{Uniform}(0, 1)$ **then**
$\quad\quad\quad$ Set $x = y$ and $d_x = d_y$    // $x \leftarrow y$ is the new candidate center
$\quad$ Set $C^i = C^{i-1} \cup \{x\}$
**return** $C^k$



**Algorithm 1:** ApxSeeding$(\mathcal{X}, k, m, f)$



## 2 Size generalization for clustering algorithm selection

In this section, we provide size generalization guarantees for semi-supervised clustering algorithm selection. Section 2.1 covers $k$-centers and $k$-means++ and Section 2.2 covers single linkage.

We denote a finite clustering instance as $\mathcal{X} \subset \mathbb{R}^d$ and let $\mathcal{P}(\mathcal{X}) := \{S : S \subset \mathcal{X}\}$ denote the power set of $\mathcal{X}$. A collection $\{S_1, ..., S_k\}$ is a $k$-clustering of $\mathcal{X}$ if it $k$-partitions $\mathcal{X}$ into $k$ disjoint subsets. We adopt the following measure of clustering quality, common in semi-supervised settings.

**Definition 2.1** ([5, 10, 12]). *Let $\mathcal{C} = (C_1, ..., C_k)$ be a $k$-clustering of $\mathcal{S} \subset \mathcal{X}$ and $\mathcal{G} = (G_1, ..., G_k)$ be the ground-truth $k$-clustering of $\mathcal{X}$. The* cost *of clustering $\mathcal{C}$ with respect to $\mathcal{G}$ is* $\text{cost}_{\mathcal{G}}(\mathcal{C}) := 1/|\mathcal{S}| \cdot \min_{\sigma \in \Sigma_k} \sum_{x \in \mathcal{S}} \sum_{j \in [k]} \mathbb{1}(x \in C_{\sigma(j)} \wedge x \notin G_j)$. *The* accuracy *of $\mathcal{C}$ is* $1 - \text{cost}_{\mathcal{G}}(\mathcal{C})$.

In Equation (1), an algorithm's *value* in this semi-supervised setting is its clustering accuracy. We evaluate our methods by the number of queries to *distance* and *ground-truth oracles*. A distance oracle outputs the distance $d(u, v)$ between queries $u, v \in \mathcal{X}$. A ground-truth oracle, given $x \in \mathcal{X}$ and a target number of clusters $k$, returns an index $j \in [k]$ in constant time. The *ground truth clustering* $G_1, \ldots, G_k$ is the partitioning of $\mathcal{X}$ such that for all $x \in G_j$, the ground-truth oracle returns $j$. This oracle could represent a domain expert, as in prior work [6, 76, 80], and is relevant in many real-world applications, such as medicine [70], where labeling a full dataset is costly. Further, it can be implemented using a *same-cluster query oracle*, which has been extensively studied [6, 65, 76].

### 2.1 $k$-Means++ and $k$-Centers clustering

We now provide size generalization bounds for *center-based clustering algorithms*. These algorithms return centers $C := \{c_1, ..., c_k\} \subset \mathbb{R}^d$ and partition $\mathcal{X}$ into $S_1, ..., S_k$ where each cluster is defined as $S_i = \{x : i = \arg\min d(x, c_j)\}$ (with arbitrary tie-breaking). We define $d_{\text{center}}(x; C) := \min_{c_i} d(x, c_i)$. Two well-known center-based clustering objectives are $k$-means, minimizing $\sum d_{\text{center}}(x; C)^2$, and $k$-centers, minimizing $\max d_{\text{center}}(x; C)$. Classical approximation algorithms for these problems are $k$-means++ [4] and Gonzalez's heuristic [1985], respectively. Both are special cases of a general algorithm which we call Seeding. In Seeding, the first center is selected uniformly at random. Given $i - 1$ centers $C^{i-1}$, the next center $c_i$ is sampled with probability $\propto f(d_{\text{center}}(c_i; C^{i-1}); \mathcal{X})$, where $f$ dictates the selection criterion. Under $k$-means++, $f(z; \mathcal{X}) = z^2$ (following the one-step version common in theoretical analyses [4, 7]), and under Gonzalez's heuristic, which selects the farthest point from the existing centers, $f(z; \mathcal{X}) = \mathbb{1}(z = \arg\max d_{\text{center}}(x; C^{i-1}))$. Seeding terminates after $k$ rounds, requiring $\mathcal{O}(|\mathcal{X}|k)$ distance oracle queries. See Algorithm 3 in Appendix E.1 for the pseudo-code of Seeding$(\mathcal{X}, k, f)$.

To estimate the accuracy of Seeding with $m$ samples, we use ApxSeeding (Algorithm 1), an MCMC approach requiring only $\mathcal{O}(mk^2)$ distance oracle queries. The acceptance probabilities are controlled by $f$, ensuring the selected centers approximate Seeding. ApxSeeding generalizes a method by Bachem et al. [7]—designed for $k$-means—to accommodate any function $f$. Also, unlike Bachem et al., we provide *accuracy* guarantees with respect to an arbitrary ground truth.

Theorem 2.2 specifies the required sample size $m$ in ApxSeeding to achieve an $\epsilon$-approximation to the accuracy of Seeding. Our bound depends linearly on $\zeta_{k,f}(\mathcal{X})$, a parameter that quantifies the smoothness of the distribution over selected centers when sampling according to $f$ in Seeding.

Ideally, if this distribution is nearly uniform, $\zeta_{k,f}(\mathcal{X})$ is close to 1. However, if the distribution is highly skewed towards selecting certain points, then $\zeta_{k,f}(\mathcal{X})$ may be as large as $n$.

**Theorem 2.2.** *Let $\mathcal{X} \subset \mathbb{R}^d$, $\epsilon, \epsilon' > 0$, $\delta \in (0,1)$, and $k \in \mathbb{Z}_{>0}$. Define the sample complexity $m = \mathcal{O}(\zeta_{k,f}(\mathcal{X}) \log(k/\epsilon))$ where $\zeta_{k,f}(\mathcal{X})$ quantifies the sampling distribution's smoothness:*

$$\zeta_{k,f}(\mathcal{X}) := \max_{Q \subset \mathcal{X} : |Q| \leq k} \max_{x \in \mathcal{X}} \left( nf(d_{\mathsf{center}}(x;Q);\mathcal{X}) \cdot \sum_{y \in \mathcal{X}} f(d_{\mathsf{center}}(y;Q);\mathcal{X})^{-1} \right).$$

*Let $S$ and $S'$ be the partitions of $\mathcal{X}$ induced by $\mathsf{Seeding}(\mathcal{X}, k, f)$ and $\mathsf{ApxSeeding}(\mathcal{X}_{mk}, k, m, f)$ where $\mathcal{X}_{mk}$ is a sample of $mk$ points drawn uniformly with replacement from $\mathcal{X}$. For any ground-truth clustering $\mathcal{G}$ of $\mathcal{X}$, $\mathbb{E}\left[\mathsf{cost}_{\mathcal{G}}(S')\right] \approx_\epsilon \mathbb{E}\left[\mathsf{cost}_{\mathcal{G}}(S)\right]$. Moreover, given $S'$, $\mathsf{cost}_{\mathcal{G}}(S')$ can be estimated to additive error $\epsilon'$ with probability $1 - \delta$ using $\mathcal{O}(k\epsilon'^{-2}\log(\delta^{-1}))$ ground-truth queries.*

*Proof sketch.* Let $p(c_i|C^{i-1})$ and $p'(c_i|C^{i-1})$ denote the probabilities that Seeding and ApxSeeding, respectively, select $c_i$ as the $i^{th}$ center, given the first $i - 1$ centers $C^{i-1}$. The proof's key technical insight leverages MCMC convergence rates to show that $p(c_i|C^{i-1})$ and $p'(c_i|C^{i-1})$ are close in total variation distance and thus the output distributions of ApxSeeding and Seeding are similar. $\square$

**Application to Gonzalez's $k$-centers heuristic.** Directly applying our framework to Gonzalez's heuristic presents an obstacle: it *deterministically* selects the farthest point from its nearest center at each step, resulting in $\zeta_{k,f}(\mathcal{X}) = n$ and high sample complexity. We therefore analyze a smoothed variant, $\mathsf{SoftmaxCenters}(\mathcal{X}, k, \beta) = \mathsf{Seeding}(\mathcal{X}, k, f_{\mathsf{SoftmaxCenters}})$, where $f_{\mathsf{SoftmaxCenters}}(z; \mathcal{X}) = \exp(\beta z/R)$ and $R = \max_{x,y} d(x, y)$. This formulation introduces a tradeoff with $\beta$: larger values align the algorithm closely with Gonzalez's heuristic, yielding a strong $k$-centers approximation (Theorem 2.3). Conversely, smaller $\beta$ promotes more uniform center selection, improving the mixing rate of MCMC in ApxSeeding and reducing sample complexity (Theorem 2.4).

To address the first aspect of this tradeoff, we show that with an appropriately chosen $\beta$, SoftmaxCenters provides a constant-factor approximation to the $k$-centers objective on instances with *well-balanced $k$-centers solutions* [8, 37, 61]. Formally, a clustering instance $\mathcal{X}$ is $(\mu_\ell, \mu_u)$-*well-balanced* with respect to a partition $S_1, \ldots, S_k$ if for all $i \in [k]$, $\mu_\ell|\mathcal{X}| \leq |S_i| \leq \mu_u|\mathcal{X}|$.

**Theorem 2.3.** *Let $k \in \mathbb{Z}_{\geq 0}$, $\gamma > 0$ and $\delta \in (0,1)$. Let $S_{\mathsf{OPT}}$ be the partition of $\mathcal{X}$ induced by the optimal $k$-centers solution $C_{\mathsf{OPT}}$, and suppose $\mathcal{X}$ is $(\mu_\ell, \mu_u)$-well-balanced with respect to $S_{\mathsf{OPT}}$. Let $C$ be the centers obtained by $\mathsf{SoftmaxCenters}$ with $\beta = R\gamma^{-1}\log\left(k^2\mu_u\mu_\ell^{-1}\delta^{-1}\right)$. With probability $1 - \delta$, $\max_{x \in \mathcal{X}} d_{\mathsf{center}}(x;C) \leq 4\max_{x \in \mathcal{X}} d_{\mathsf{center}}(x;C_{\mathsf{OPT}}) + \gamma$.*

Furthermore, Theorem 2.4 shows that $\zeta_{k,f}$ can be bounded solely in terms of $\beta$.

**Theorem 2.4.** *For any $\beta > 0$, $\zeta_{k,\mathsf{SoftmaxCenters}}(\mathcal{X}) \leq \exp(2\beta)$.*

Setting $\beta$ as per Theorem 2.3 ensures the bound in Theorem 2.4—and thus the sample size sufficient for size generalization—is independent of $n$ and $d$. Moreover, our experiments (Section 4) show that choosing smaller values of $\beta$ still yields comparable approximations to the $k$-centers objective.

**Application to $k$-means++.** Bachem et al. [7] identify conditions on $\mathcal{X}$ to guarantee that $\zeta_{k,\mathsf{kmeans++}}(\mathcal{X}) := \zeta_{k,x \mapsto x^2}(\mathcal{X})$ is independent of $|\mathcal{X}|$ and $d$: if $\mathcal{X}$ is from a distribution that satisfies mild non-degeneracy assumptions and has support contained in a ball of radius $R$, then $\zeta_{k,\mathsf{kmeans++}}(\mathcal{X})$ scales linearly with $R^2$ and $k$. See Theorem D.1 in Appendix D.1 for details.

## 2.2 Single-linkage clustering

We next study the far more brittle single-linkage (SL) algorithm. While SL is known to be unstable [11, 29], our experiments (Section 4) reveal that its accuracy can be estimated on a subsample—albeit larger than that of the center-based algorithms. To better understand this phenomenon, we pinpoint structural properties of $\mathcal{X}$ that facilitate size generalization when present and hinder it when absent.

SL (Algorithm 2) can be viewed as a version of Kruskall's algorithm for computing minimum spanning trees, treating $\mathcal{X}$ as a complete graph with edge weights $\{d(x, y)\}_{x,y \in \mathcal{X}}$. We define the *inter-cluster distance* $d(A, B) := \min_{x \in A, y \in B} d(x, y)$ (see Figure 9a in Appendix E.2.1). Initially, each point $x \in \mathcal{X}$ forms its own cluster. At each iteration, the algorithm increases a distance threshold

**Input:** Clustering instance $\mathcal{X} \subset \mathbb{R}^d$ and $k \in \mathbb{Z}_{\geq 0}$
Initialize $\mathcal{C}^0 := \{C_1^0 = \{x_1\}, ..., C_n^0 = \{x_n\}\}$, $i = 1$
**while** $\left|\mathcal{C}^{i-1}\right| > 1$ **do**
    Set $d_i := \min_{A,B \in \mathcal{C}^{i-1}} d(A, B)$ and $\mathcal{C}^i = \mathcal{C}^{i-1}$
    **while** *there exist $A, B \in \mathcal{C}^i$ such that $d(A, B) = d_i$* **do**
        Choose $A, B \in \mathcal{C}^i$ such that $d(A, B) = d_i$ and set $\mathcal{C}^i = \mathcal{C}^i \setminus \{A, B\} \cup \{A \cup B\}$
    **if** $\left|\mathcal{C}^i\right| = k$ **then**
        $\mathcal{C} = \mathcal{C}^i$                              `// This is the algorithm's output`
    Set $i = i + 1$     `// Continued until` $\left|\mathcal{C}^i\right| = 1$ `to define variables in the analysis`
**return** $\mathcal{C}$

**Algorithm 2:** $\mathsf{SL}(\mathcal{X}, k)$

$d_i$ and merges points into the same cluster if there is a path between them in the graph with all edge weights at most $d_i$. The algorithm returns $k$ clusters corresponding to the $k$ connected components that Kruskall's algorithm obtains. The next definition and lemma formalize this process.

**Definition 2.5.** *The* min-max distance *between $x, y \in \mathcal{X}$ is $d_{\mathsf{mm}}(x, y; \mathcal{X}) = \min_p \max_i d(p_i, p_{i+1})$, where the $\min$ is taken across all simple paths $p = (p_1 = x, ..., p_t = y)$ from $x$ to $y$ in the complete graph over $\mathcal{X}$. For a subset $S \subset \mathcal{X}$, we also define $d_{\mathsf{mm}}(S; \mathcal{X}) := \max_{x,y \in S} d_{\mathsf{mm}}(x, y; \mathcal{X})$.*

**Lemma 2.6.** *In $\mathsf{SL}(\mathcal{X}, k)$, $x, y \in \mathcal{X}$ belong to the same cluster after iteration $\ell$ if and only if $d_{\mathsf{mm}}(x, y; \mathcal{X}) \leq d_\ell$.*

*Proof sketch.* The algorithm begins with each point as an isolated node and iteratively adds the lowest-weight edges between connected components. Nodes are connected after step $\ell$ if and only if there is a path connecting them where all edge weights are at most $d_\ell$. $\qquad\square$

Suppose we run $\mathsf{SL}$ on $m$ points $\mathcal{X}_m$ sampled uniformly without replacement from $\mathcal{X}$, so clusters merge based on $d_{\mathsf{mm}}(x, y; \mathcal{X}_m)$ (rather than $d_{\mathsf{mm}}(x, y; \mathcal{X})$). Since subsampling reduces the number of paths, it can only increase min-max distance: $d_{\mathsf{mm}}(x, y; \mathcal{X}_m) \geq d_{\mathsf{mm}}(x, y; \mathcal{X})$. If $d_{\mathsf{mm}}(x, y; \mathcal{X}_m)$ remains similar to $d_{\mathsf{mm}}(x, y; \mathcal{X})$ for most $x, y \in \mathcal{X}$, then $\mathsf{SL}(\mathcal{X}_m, k)$ and $\mathsf{SL}(\mathcal{X}, k)$ should return similar clusterings. We derive a sample complexity bound ensuring that this condition holds which depends on $\zeta_{k,\mathsf{SL}} \in (0, n]$, defined as follows for clusters $\{C_1, \ldots, C_k\} = \mathsf{SL}(\mathcal{X}, k)$:

$$\zeta_{k,\mathsf{SL}}(\mathcal{X}) := n \left\lceil \frac{\min_{i,j \in [k]} d_{\mathsf{mm}}(C_i \cup C_j; \mathcal{X}) - \max_{t \in [k]} d_{\mathsf{mm}}(C_t; \mathcal{X})}{\max_{t \in [k]} d_{\mathsf{mm}}(C_t; \mathcal{X})} \right\rceil^{-1}.$$

This quantity measures cluster separation by min-max distance, reflecting $\mathsf{SL}$'s stability with respect to random deletions from $\mathcal{X}$. Higher $\zeta_{k,\mathsf{SL}}(\mathcal{X})$ indicates greater sensitivity to subsampling, while a lower value suggests robustness. Appendix D.2.1 provides a detailed discussion and proves $\zeta_{k,\mathsf{SL}} \in (0, n]$.

**Theorem 2.7.** *Let $\mathcal{G} = \{G_1, ..., G_k\}$ be a ground-truth clustering of $\mathcal{X}$, $\mathcal{C} = \{C_1, ..., C_k\} = \mathsf{SL}(\mathcal{X}, k)$ be the clustering obtained from the full dataset, and $\mathcal{C}' = \{C_1', ..., C_k'\} = \mathsf{SL}(\mathcal{X}_m, k)$ be the clustering obtained from a random subsample $\mathcal{X}_m$ of size $m$. For*

$$m = \tilde{\mathcal{O}}\left(\left(\frac{k}{\epsilon^2} + \frac{n}{\min_{i \in [k]} |C_i|} + \zeta_{k,\mathsf{SL}}(\mathcal{X})\right) \log \frac{k}{\delta}\right),$$

*we have that $\mathbb{P}\left\{\mathsf{cost}_{\mathcal{G}}\left(\mathcal{C}'\right) \approx_\epsilon \mathsf{cost}_{\mathcal{G}}\left(\mathcal{C}\right)\right\} \geq 1 - \delta$. Computing $\mathcal{C}'$ requires $\mathcal{O}(m^2)$ calls to the distance oracle, while computing $\mathsf{cost}_{\mathcal{G}}\left(\mathcal{C}'\right)$ requires only $m$ queries to the ground-truth oracle.*

*Proof sketch.* Let $H$ be the event that $\mathcal{C}'$ is *consistent* with $\mathcal{C}$ on $\mathcal{X}_m$, i.e., there is a permutation $\sigma \in \Sigma_k$ such that for every $i \in [k]$, $C_i' \subset C_{\sigma(i)}$. The main challenge is to quantify how deviations from consistency affect the bottleneck distance. If $H$ does not occur, we show that for some $S \subset \mathcal{X}_m$, $d_{\mathsf{mm}}(S; \mathcal{X}_m) \ll d_{\mathsf{mm}}(S; \mathcal{X})$, indicating many consequential points must have been deleted. To formalize this, we lower bound the number of deletions required for this discrepancy to occur and thereby lower bound $\mathbb{P}[H]$. Conditioned on $H$, Hoeffding's bound implies the theorem statement. $\quad\square$

When $\zeta_{k,\mathsf{SL}}(\mathcal{X})$ and $1/\min_{i \in [k]} |C_i|$ are small relative to $n$, Theorem 2.7 guarantees that a small subsample suffices to approximate the accuracy of SL on the full dataset. In Appendix D.2, we show that the dependence on $\zeta_{k,\mathsf{SL}}(\mathcal{X})$ and $\min_{i \in [k]} |C_i|$ in Theorem 2.7 is necessary, so our results are tight. Appendix D.2 also presents empirical studies of $\zeta_{k,\mathsf{SL}}$ on natural data-generating distributions.

# 3 Max-cut

In this section, we present size generalization bounds for max-cut. Section 3.1 and Section 3.2 discuss the GW and Greedy algorithms, respectively. We denote a weighted graph as $G = (V, E, \boldsymbol{w})$, where $w_{ij}$ is the weight of edge $(i, j) \in E$. The vertex set is $V = [n]$. If $G$ is unweighted, we denote it as $G = (V, E)$. We use $A_G$ to denote the adjacency matrix, $D_G$ for the diagonal degree matrix, and $L_G := D_G - A_G$ for the Laplacian. The subgraph of $G$ induced by $S \subseteq V$ is denoted $G[S]$. A cut in $G$ is represented by $\boldsymbol{z} \in \{-1, 1\}^n$, and the max-cut problem is: $\operatorname{argmax}_{\boldsymbol{z} \in \{-1,1\}^n} \mathsf{weight}_G(\boldsymbol{z})$, where $\mathsf{weight}_G(\boldsymbol{z}) := \frac{1}{2} \sum_{(i,j) \in E} w_{ij}(1 - z_i z_j)$ is the cut weight and $\mathsf{weight}_G(\boldsymbol{z})/n^2$ is its density, a well-established metric in prior max-cut literature [17, 64].

## 3.1 The Goemans-Williamson (GW) algorithm

The GW algorithm yields a .878-approximation to the max-cut problem by solving the following semidefinite programming (SDP) relaxation, yielding a solution $X = \mathsf{SDPSolve}(G) \in \mathbb{R}^{n \times n}$ to:

$$\max_{X \succeq 0} \ 1/4 \cdot L_G \cdot X \quad \text{subject to} \quad (\boldsymbol{e}_i \boldsymbol{e}_i^T) \cdot X = 1 \quad \forall i = 1, \ldots, n. \tag{2}$$

GW then converts $X$ into a cut using randomized rounding, denoted $\mathsf{GWRound}(X) \in \{-1, 1\}^n$. To do so, it generates a standard normal random vector $\boldsymbol{u} \in \mathbb{R}^n$ and computes the Cholesky factorization $X = VV^T$, with $V \in \mathbb{R}^{n \times n}$. It then defines the cut $\boldsymbol{z} = \mathsf{GWRound}(X)$ by computing the dot product between the $i^{th}$ column $\boldsymbol{v}_i$ of $V$ and $\boldsymbol{u}$: $z_i = \mathrm{sign}(\boldsymbol{v}_i^T \boldsymbol{u})$. We denote the complete algorithm as $\mathsf{GW}(G) = \mathsf{GWRound}(\mathsf{SDPSolve}(G)) \in \{-1, 1\}^n$ (see Algorithm 4 in Appendix E.3).

Theorem 3.1 provides a size generalization bound for the SDP relaxation objective (2), proving that the optimal SDP objective value for a random subgraph $G[S_t]$ with $\mathbb{E}[|S_t|] = t$ converges to that of $G$ as $t \to n$, without imposing any assumptions on $G$. Building on this result, Theorem 3.3 shows that the cut density produced by the GW algorithm on $G$ can be estimated using just the subgraph $G[S_t]$.

**Size generalization for GW SDP objective value.** We begin with a size generalization bound for the SDP relaxation. Let $\mathsf{SDP}(G)$ denote the objective value of the optimal solution to Equation (2). We show that for a randomly sampled subset of vertices $S_t$ with $\mathbb{E}[|S_t|] = t$, the objective value $\mathsf{SDP}(G[S_t])$ converges to $\mathsf{SDP}(G)$ as $t$ increases. Our result holds for any graph $G$, unlike prior work by Barak et al. [17], who proved GW SDP convergence only for random geometric graphs.

**Theorem 3.1.** *Given $G = (V, E, \boldsymbol{w})$, let $S_t$ be a set of vertices with each node sampled from $V$ independently with probability $\frac{t}{n}$. Let $W = \sum_{(i,j) \in E} w_{ij}$. Then*

$$\left| \frac{1}{t^2} \mathbb{E}_{S_t}[\mathsf{SDP}(G[S_t])] - \frac{1}{n^2} \mathsf{SDP}(G) \right| \leq \frac{n-t}{n^2 t} \left( \mathsf{SDP}(G) - \frac{W}{2} \right).$$

*Proof sketch.* We prove this result by separately deriving upper and lower bounds on $\mathbb{E}[\mathsf{SDP}(G[S_t])]$. The primary challenge is deriving the upper bound. To do so, we analyze the dual of Equation (2):

$$\min_y \ \sum_{i=1}^n y_i \quad \text{subject to} \quad \sum_{i=1}^n y_i (\boldsymbol{e}_i \boldsymbol{e}_i^T) \succeq \frac{1}{4} L_G. \tag{3}$$

where $\boldsymbol{e_i} \in \mathbb{R}^n$ denotes the all-zero vector with 1 on index $i$. By strong duality, the optimal primal and dual values coincide. To produce an upper bound on $\mathbb{E}[\mathsf{SDP}(G[S_t])]$ that depends on the full graph $G$, let $\boldsymbol{y}^*$ be the optimal dual solution for $G$. Trimming $\boldsymbol{y}^*$ to $t$ dimensions yields a feasible—but loose—solution to the dual SDP defined on $G[S_t]$. To tighten this solution, we use the key observation that the portion of $\boldsymbol{y}^*$ corresponding to nodes in $[n] \setminus S_t$ is superfluous for $G[S_t]$. Lemma E.12 in Appendix E.3 shows that removing this component by setting $\bar{y}_i = y_i^* - \sum_{k \in [N] \setminus S_t} w_{ik}$ yields a feasible dual solution providing a tighter upper bound. Hence, $\sum_{i=1}^t \bar{y}_i$ upper bounds $\mathsf{SDP}(G[S_t])$.

Next, to obtain a lower bound that depends on the full graph $G$, let $X^*$ be the optimal primal SDP solution induced by $G$. Without loss of generality, assume that $S_t$ consists of the first $t$ nodes in $G$. Trimming $X^*$ to its $t$-th principal minor $X^*[t]$ yields a feasible solution to the primal SDP induced by $G[S_t]$. Hence, $L_{G[S_t]} \cdot X^*[t]$ is a lower bound for $\mathsf{SDP}(G[S_t])$. $\qquad\square$

As $t \to n$, the error term goes to 0. Notably, this result requires no assumptions on graph structure. The term $\mathsf{SDP}(G) - W/2$ (or $\mathsf{SDP}(G) - |E|/2$ for unweighted graphs) adapts to different graph structures. In unweighted sparse graphs, $\mathsf{SDP}(G) - |E|/2$ is at most $|E|/2$, benefiting from sparsity. However, the bound is still vanishing for unweighted dense graphs: $\mathsf{SDP}(G)$ is approximately $|E|/2$, so $\mathsf{SDP}(G) - |E|/2$ approaches 0. Meanwhile, in a graph with highly uneven weights, a random subgraph is unlikely to represent the full graph well, and $\mathsf{SDP}(G) - W/2$ can be as large as $W/2$.

In Theorem E.18 in Appendix E.3, we include a version of this result that holds with probability over $S_t$. We apply McDiarmid's Inequality to establish that with high probability, $\mathsf{SDP}(G[S_t])$ is close to its expectation and, therefore, close to the normalized SDP value induced by $G$.

**Size generalization bounds for GW.** We now present our size generalization bound for the GW algorithm. Given that the SDP objective values for the subsampled graph $G[S_t]$ and the full graph $G$ converge, one might expect the GW cut values to converge at the same rate. However, Lemma 3.2 demonstrates that this is not necessarily the case: even if two distinct optimal SDP solutions have the same objective value, the resulting distributions of GW cut values after randomized rounding can differ. This arises due to the non-Lipschitz behavior of the mapping between the SDP solution and the GW cut value. See Figure 5 for an illustrative example, and Appendix E.3 for a proof.

**Lemma 3.2.** *For any $n \geq 2$, there exists a graph $G_n$ on $n$ vertices with distinct optimal solutions $X_n \neq Y_n$ to Equation* (2) *and a constant $C > 0$ such that $L_{G_n} \cdot X_n = L_{G_n} \cdot Y_n$ but $|\mathbb{E}[\mathsf{weight}_{G_n}(\mathsf{GWRound}(X_n))] - \mathbb{E}[\mathsf{weight}_{G_n}(\mathsf{GWRound}(Y_n))]| \geq Cn$.*

Nonetheless, we show that we can still use Theorem 3.1 to upper and lower bound the GW cut value on the full graph as a function of $\mathsf{SDP}(G[S_t])$, albeit with a multiplicative and additive error.

**Theorem 3.3.** *Given $G = (V, E, \boldsymbol{w})$, let $S_t$ be a set of vertices with each node sampled from $V$ independently with probability $\frac{t}{n}$. Then, for $\epsilon_{SDP} = \frac{n-t}{n^2 t}\left(\mathsf{SDP}(G) - \frac{W}{2}\right)$, we have*

$$\frac{1}{n^2}\mathsf{weight}_G\left(\mathsf{GW}(G)\right) \in \left[\frac{0.878}{t^2}\mathop{\mathbb{E}}_{S_t}[\mathsf{SDP}(G[S_t])] - 0.878\epsilon_{SDP}, \frac{1}{t^2}\mathop{\mathbb{E}}_{S_t}[\mathsf{SDP}(G[S_t])]\right].$$

*Proof sketch.* By definition, $\mathsf{weight}_G(\mathsf{GW}(G)) \leq \mathsf{SDP}(G)$, and moreover, $\mathsf{weight}_G(\mathsf{GW}(G)) \geq 0.878 \cdot \mathsf{SDP}(G)$ [47]. Combining this with Theorem 3.1, which bounds the error of estimating the SDP optimal value, we obtain the theorem statement. $\qquad\square$

To interpret this result, we know that $\frac{1}{n^2}\mathsf{weight}_G(\mathsf{GW}(G)) \in \left[\frac{0.878}{n^2}\mathsf{SDP}(G), \frac{1}{n^2}\mathsf{SDP}(G)\right]$. Theorem 3.3 shows that $\frac{1}{n^2} \cdot \mathsf{SDP}(G)$ can be replaced by $\frac{1}{t^2} \cdot \mathbb{E}_{S_t}[\mathsf{SDP}(G[S_t])]$—which is much faster to compute—with error $\epsilon_{\mathsf{SDP}}$ that vanishes as $t \to n$. Thus, the GW cut value on $G$ can be estimated using the subgraph, providing an efficient way to estimate GW's performance on large graphs.

## 3.2 The greedy algorithm

We now present size generalization bounds for the Greedy 2-approximation algorithm. Despite its worse approximation guarantee, Greedy frequently performs comparably to—or sometimes better than—GW [55, 66], while requiring only linear runtime $O(|E|)$. Greedy iterates over the nodes in random order, denoted by the permutation $\sigma \in \Sigma_{|V|}$, and sequentially assigns each node to the side of the cut that maximizes the weight of the current partial cut (see Algorithm 5 in Appendix E.3). The next theorem shows we can approximate Greedy's performance with only $O(1/\epsilon^2)$ samples.

**Theorem 3.4.** *Given an unweighted graph $G = (V, E)$, let $S_t$ be $t$ vertices sampled from $V$ uniformly without replacement. For any $\epsilon \in [0, 1]$ and $t \geq \frac{1}{\epsilon^2}$,*

$$\left|\frac{1}{n^2}\mathbb{E}[\mathsf{weight}_G(\mathsf{Greedy}(G))] - \frac{1}{t^2}\mathbb{E}[\mathsf{weight}_{G[S_t]}(\mathsf{Greedy}(G[S_t]))]\right| \leq O\left(\epsilon + \frac{\log(t)}{\sqrt{n}}\right).$$

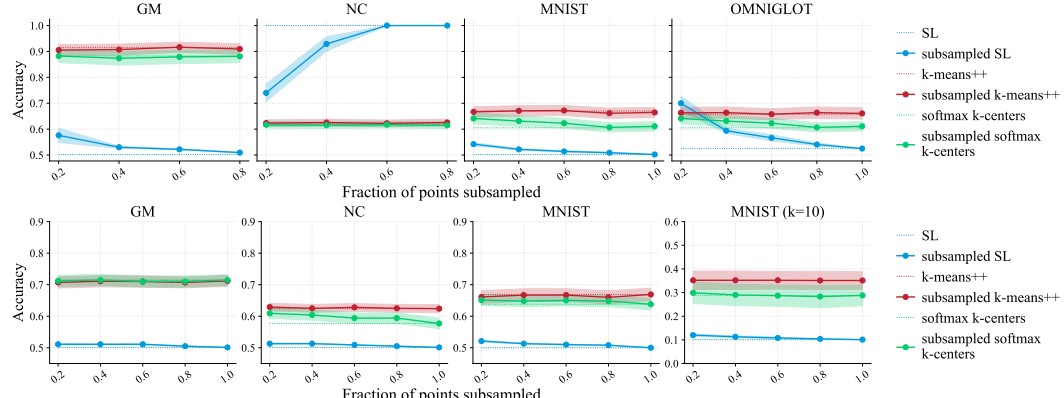

(a) Accuracy of clustering computed by SL, $k$-means++, and softmax $k$-centers on full data and on a subsample. Top row: n = 500 for MNIST, GM, and NC; n = 40 for Omniglot (due to smaller instance size). Bottom row: n = 2000 for MNIST, GM, and NC; the final panel uses n = 4000 and k = 10 for MNIST.

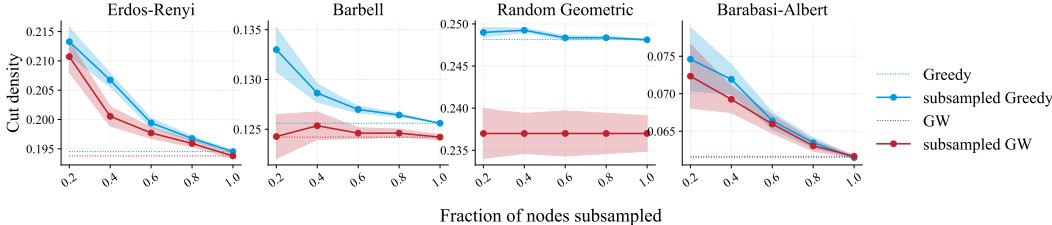

(b) Density of cuts computed by GW and Greedy on full graphs and on a subsample.

Figure 1: In 1, the proxy algorithms' accuracies on the subsample approach those of the original algorithms on the full instance as the sample size grows. Figure 1a shows this for clustering algorithms, and Figure 1b for max-cut algorithms. Shadows denote two standard errors about the average.

*Proof sketch.* At step $t$, we define the *partial cut* on the subset of nodes $\{\sigma[1], ..., \sigma[t]\}$ as $\boldsymbol{z}^t \in \{-1, 0, 1\}^{|V|}$, where $z_i^t \in \{-1, 1\}$ indicates node $i$ has been assigned and $z_i^t = 0$ indicates it has not. To evaluate the quality of a partial cut, we use the concept of a "fictitious" cut $\hat{\boldsymbol{z}}^t$, introduced by Mathieu and Schudy [64]. Intuitively, $\hat{\boldsymbol{z}}^t$ serves as an extrapolation of $\boldsymbol{z}^t$ into a complete (fractional) cut. It is defined by estimating how each unassigned vertex would have been assigned if it had been selected at time $\tau \leq t$, and averaging this estimate over all possible times $\tau$. This construct allows us to compare partial and full cuts by bringing them to a comparative scale. Since Greedy adds nodes sequentially, the theorem statement is equivalent to bounding the difference between the densities of a partial cut and the full cut. Analyzing the fictitious and partial cuts as martingales allows us to bound the difference between the partial and full cuts using the fictitious cut. □

## 4 Experiments

In this section, we present experiments validating that the best algorithm on a max-cut or clustering instance can be effectively inferred from a subsample.

**Clustering.** We present experiments on four datasets. (1) The MNIST generator [13]: handwritten digits with images labeled corresponding to them. (2) GM: points from a 2-dimensional isotropic Gaussian mixture model sampled from $\mathcal{N}((0,0)^\top, .5\boldsymbol{I})$ or $\mathcal{N}((1,1)^\top, .5\boldsymbol{I})$ with probability 1/2 and labeled by the Gaussian from which it was sampled. (3) The Omniglot generator [13]: handwritten characters from various languages, labeled by the language. (4) The noisy circles (NC) generator [69]: points on two concentric circles (with noise level 0.05, distance factor 0.2), labeled by the circle to which they belong. All instances contain $n$ points evenly divided into $k = 2$ clusters except the last panel of the bottom row, where $k = 10$. On the top row, for MNIST, GM, and NC, $n = 500$. For Omniglot, $n = 40$ (instances from this dataset are inherently smaller). On the bottom row, $n = 2000$ for all except the last panel, where we use $n = 4000$.

Figure 1a compares the accuracy of the proxy algorithms from Section 2 with the original algorithms run on the full dataset (averaged across $10^3$ trials), plotted as a function of the number of randomly sampled points, $m$. As $m$ increases, the proxy algorithms' accuracies converge to those of the original algorithms. Appendix C.2.1 includes additional experiments on softmax $k$-centers. We show that its $k$-centers objective value approximates or improves over Gonzalez's algorithm, even for small $\beta$.

**Max-cut.**    We test four random graph families with with $n = 50$ vertices: Erdös-Réyni ($p = 0.7$), Barbell with 5 random inter-clique edges, random geometric (radius 0.9), and Barabási-Albert ($m = 5$). Figure 1b plots the mean cut density over 150 trials for full and subsampled graphs versus the sampled fraction. As the sample size grows, the Greedy and GW densities on the subsamples converge to the full-graph values. For the random geometric and barbell graphs, even a small subsample reveals the better-performing algorithm, enabling low-cost algorithm selection, while on Barabási-Albert and Erdös-Réyni, the two curves coincide, implying that the faster Greedy will achieve comparable results. These results not only corroborate our theory but also demonstrate an even stronger convergence behavior. Appendix C.1 includes additional experiments on the SDP objective value convergence and demonstrates the percentage speed-up we gain from the subsampling scheme.

## 5    Conclusion

We introduced the notion of size generalization and established rigorous bounds for classical algorithms for two canonical partitioning problems: clustering and max-cut. Our analysis identifies sufficient conditions under which an algorithm's performance on a large instance can be estimated using a small, representative instance, addressing a key computational bottleneck in data-driven algorithm selection.

We hope that our work provides useful techniques for future work on size generalization in data-drive algorithm selection. In particular, we believe it would be interesting to extend our results to broader classes of optimization problems, such as integer programming and additional SDP-based algorithms (e.g., randomized rounding for correlation clustering). Another exciting direction for future work is to formulate a unified theoretical toolkit for size generalization—one that prescribes, given a new combinatorial algorithm, which structural properties guarantee that its performance on large instances can be extrapolated from small ones.

## Acknowledgments

We thank the reviewers for their anonymous feedback. The authors would like to thank Moses Charikar for insightful discussions during the early stages of this work. This work was supported in part by National Science Foundation (NSF) award CCF-2338226 and a Schmidt Sciences AI2050 fellowship.

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

# Appendix

## Table of Contents

# A  Code Repository

We release our code for the experiments at `https://github.com/Yingxi-Li/Size-Generalization`.

# B  Key Challenges and Additional Related Work

In this section of the Appendix, we expand on the discussion from the introduction (Section 1). We first elaborate on additional related work in the context of our paper. Then, we provide figures illustrating the key challenges we mention in Section 1.

## B.1  Additional related work

**Algorithm selection.**  Our work follows a line of research on algorithm selection. Practitioners typically utilize the following framework: (1) sample a training set of many problem instances from their application domain, (2) compute the performance of several candidate algorithms on each sampled instance, and (3) select the algorithm with the best average performance across the training set. This approach has led to breakthroughs in many fields, including SAT-solving [84], combinatorial auctions [63], and others [28, 36, 73]. On the theoretical side, [52] analyzed this framework from a PAC-learning perspective and applied it to several problems, including knapsack and max-independent-set. This PAC-learning framework has since been extended to other problems (for example, see [9] and the references therein). In this paper, we focus on step (2) of this framework in the context of clustering and study: when can we efficiently estimate the performance of a candidate clustering algorithm with respect to a ground truth?

**Size generalization: gap between theory and practice.**  There is a large body of research on using machine learning techniques such as graph neural networks and reinforcement learning for combinatorial optimization [e.g., 35, 46, 59, 79]. Typically, training the learning algorithm on small combinatorial problems is significantly more efficient than training on large problems. Many of these papers assume that the combinatorial instances are coming from a generating distribution [e.g., 35, 46, 79], such as a specific distribution over graphs. Others employ heuristics to shrink large instances [e.g., 59]. These approaches are not built on theoretical guarantees that ensure an algorithm's performance on the small instances is similar to its performance on the large instances. Our goal is to bridge this gap.

**Subsampling for max-cut and general CSP**  Subsampling for the max-cut objective has been studied in [49] in terms of property testing, which is the study of making decisions on data with only access to part of the input [48]. Property testing encompasses a more general category of study, and the notion of size generalization we study in this work is a specific variant of it.

A flourishing line of work since the 90s aims to develop PTAS that estimate the optimal objective value of constraint satisfaction problems (CSPs) [e.g., 1, 3, 43, 44, 64]. In a similar spirit, the long-standing area of sublinear-time approximation algorithms aim to approximate the objective value of max-cut and other CSP in sub-linear time. It dates back to 1998, when a constant time approximation scheme that approximates the max-cut objective for dense graphs within an error of $\varepsilon n^2$ was developed [49]. Many later works follow [e.g., 23, 71]. For example, Bhaskara et al. show that core-sets of size $O(n^{1-\delta})$ can approximate max-cut in graphs with an average degree of $n^\delta$. Despite having weaker approximation guarantees compared to the GW algorithm, these algorithms can have much faster runtimes.

The goal of size generalization, however, differs fundamentally from the goal of the polynomial or sublinear time approximation algorithms. Rather than estimating the optimal cut value, we directly estimate the empirical performance of specific heuristics. This enables principled algorithm selection among the best-performing methods, instead of defaulting to the fastest algorithm with looser guarantees. That said, our framework could naturally be used to compare these sampling-based PTAS algorithms alongside greedy and Goemans-Williamson, efficiently determining when a PTAS's high computational cost is justified.

Perhaps most related to our work, Barak et al. [17] study a similar subsampling procedure for general CSPs, which includes the GW SDP (both with and without the triangle inequality). For $\Delta$-dense

CSPs, i.e. CSPs where every variable appears in at least $\Delta$ constraints, they bound the accuracy of the subsampled objective within $\epsilon$-error. In contrast to [17], our result focuses only on the GW SDP rather than general CSPs. Furthermore, we do not make any density assumptions for our GW SDP result, such as those made in [17] and in many other works related to approximating max-cut values [e.g., 43, 75].

**Sublinear-time algorithms for clustering** The large body of sublinear-time clustering research [e.g., 34, 41, 67] also pursues a different goal from our paper: approximating the optimal clustering cost under specific cost models (e.g., k-means or k-center), often with structural assumptions (bounded dimension, cluster separation) and via uniform sampling or coreset constructions. We discuss our cost model and why head-to-head comparison with coresets is impossible further in lines 79-87.

**Coresets for clustering** Coresets for clustering have been extremely well studied in the clustering literature [e.g., 33, 42, 54]. Coresets are designed to allow efficient optimization of a *specific, known* objective function, such as the $k$-means objective. Below, we provide a detailed explanation of why the objective-preservation guarantee offered by coresets do not yield meaningful results in our setting.

For illustrative purposes, consider the $k$-means objective and suppose we are given a large original dataset $\mathcal{X} \subset \mathbb{R}^d$. In the problem of coresets for $k$-means, the task is to construct a small (possibly weighted) subset $\mathcal{X}' \subset \mathcal{X}$ (this set is called the *coreset*) such that the $k$-means objective is well-preserved for *any* choice of $k$ centroids $c_1, ..., c_k$. That is:

$$h(c_1, ..., c_k; \mathcal{X}) \approx h(c_1, ..., c_k; \mathcal{X}'), \text{ for any } c_1, ..., c_k \subset \mathbb{R}^d,$$

where

$$h(c_1, ..., c_k; \mathcal{X}) = \frac{1}{|\mathcal{X}|} \sum_{x \in \mathcal{X}} \min_{i \in [k]} d(c_i, x)$$

$$h(c_1, ..., c_k; \mathcal{X}') = \frac{1}{|\mathcal{X}'|} \sum_{x \in \mathcal{X}'} \min_{i \in [k]} d(c_i, x).$$

Such a guarantee on $\mathcal{X}'$ would ensure that solving the $k$-means problem on $\mathcal{X}'$ yields an approximate solution to the $k$-means problem on $\mathcal{X}$. Indeed, if $|\mathcal{X}'|$ is sufficiently small relative to $|\mathcal{X}|$, then this approach yields an efficient approximate solution to the $k$-means problem on $\mathcal{X}$.

*However*, notice that in the above description of coresets, the coreset $\mathcal{X}'$ is *only guaranteed* to approximate the solution to the $k$-means objective. On the other hand, the ground truth clustering may be *completely arbitrary* and may align *poorly* with the $k$-means objective. Consequently, the coreset guarantee does *not yield any approximation* guarantee for the ground truth labeling of $\mathcal{X}$.

Thus, the key difference between our work and the prior literature on coresets is that coresets assume that the target objective is a *known, closed-form* objective function (such as the $k$-means objective function). On the other hand, our task is much harder: we want to design a small set $\mathcal{X}'$ such that running a clustering algorithm on $\mathcal{X}'$ is enough to estimate the algorithm's accuracy with respect to an *arbitrary ground-truth labeling* of $\mathcal{X}$. Since the ground truth labeling is not known a priori, traditional guarantees from the coresets literature do not apply.

### B.2 Illustrations of key challenges

### B.2.1 Key challenges of clustering

Figure 2 shows an example of a dataset where the performance of $k$-means++ is extremely sensitive to the deletion or inclusion of a single point. This illustrates that on worst-case datasets (e.g., with outliers or highly influential points) with worst-case ground truth clusterings, we cannot expect that size generalization holds.

Similarly, Figure 3 shows that on datasets without outliers, it may be possible to construct ground truth clusterings that are highly tailored to the performance of a particular clustering algorithm (such as Single Linkage) on the dataset. The figure illustrates that this type of adversarial ground truth clustering is a key challenge towards obtaining size generalization guarantees for clustering algorithms.

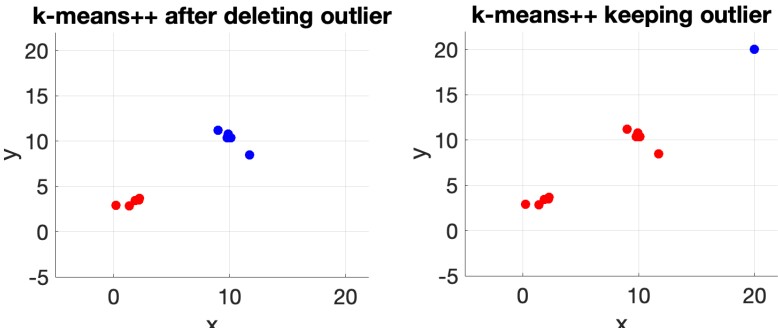

Figure 2: Sensitivity of $k$-means++ with respect to a single point. The example shows that the algorithm's accuracy can be extremely sensitive to the presence or absence of a single point, in this case, the outlier at $(20, 20)$. Depending on how the ground truth is defined, deleting the outlier can either boost or drop the accuracy by up to 50%.

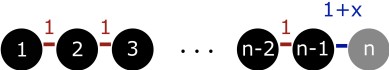

Figure 3: Example of a ground truth where size generalization by subsampling fails for single linkage. The shading indicates the ground truth. The first $n-1$ points are unit distance apart, while the last two points are $1 + x > 1$ distance apart. Single linkage has 0 cost on the full dataset. After randomly deleting a single point, the largest gap between consecutive points is equally likely to occur between any consecutive points. So, the expected cost on the subsample is $\geq \frac{1}{n} \sum_{i=2}^{n-1} \frac{(n-1-i)}{n} \overset{n\to\infty}{\to} 1/2$.

### B.2.2 Key challenges of max-cut

Figure 4 shows an example where the performance of Goemans-Williamson is sensitive to the deletion of even one node. The figure displays samples of GW values on the full Petersen graph and the Petersen graph with only one node deletion. As shown in the plot, a single node deletion completely changes the distribution of GW outputs. This can also go the other way for different examples, where a node deletion might cause worse performance with greater variance.

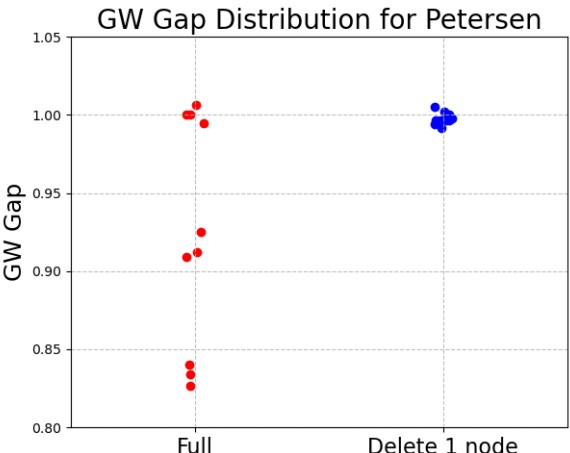

Figure 4: Sensitivity of Goemans-Williamson (GW) with respect to the deletion of one node. GW gap is calculated by $\frac{\text{GW}(G)}{\text{SDP}(G)}$. This figure illustrates the GW gap distribution on the full Peterson network and a Peterson network with one node deleted. Notice that the GW gap differs drastically when we delete 1 node from a graph.

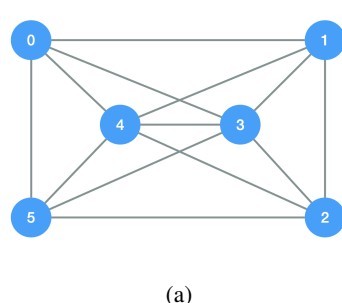

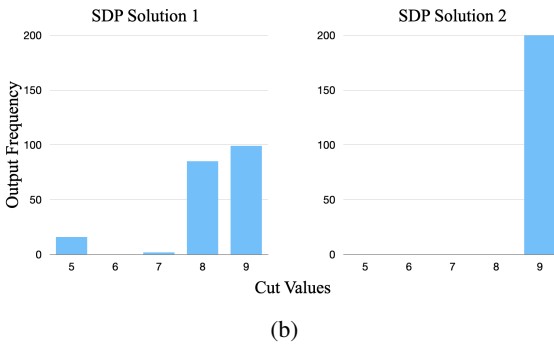

(a)                                     (b)

Figure 5: An example graph with different optimal SDP solutions leading to different GW output distribution. Figure 5a is an example graph with many distinct SDP optimal solutions. By definition, they all have the same GW SDP objective value. Figure 5b is the distribution of cut values returned by GW over two distinct optimal GW SDP solutions for graph in 5a, with 200 random hyperplanes sampled for each solution. Two optimal solutions result in very distinct GW output distribution.

As pointed out in Lemma 3.2, different SDP optimal solutions may have the same objective function values. However, their robustness to Goeman-Williamson rounding could be very different. For example, consider the graph in Figure 5a. The optimal cut in the graph is $\{0, 2, 4\}$ and $\{1, 3, 5\}$, and the optimal cut value (assuming unweighted) is 9. However, if we solve the SDP relaxation on the same graph, both SDP solutions follows are optimal:

$$X^* = \begin{bmatrix} 1.000 & -0.784 & 1.000 & -0.216 & -0.216 & -0.784 \\ -0.784 & 1.000 & -0.784 & -0.216 & -0.216 & 1.000 \\ 1.000 & -0.784 & 1.000 & -0.216 & -0.216 & -0.784 \\ -0.216 & -0.216 & -0.216 & 1.000 & -0.137 & -0.216 \\ -0.216 & -0.216 & -0.216 & -0.137 & 1.000 & -0.216 \\ -0.784 & 1.000 & -0.784 & -0.216 & -0.216 & 1.000 \end{bmatrix}$$

and

$$\hat{X} = \begin{bmatrix} 1 & -1 & 1 & -1 & 1 & -1 \\ -1 & 1 & -1 & 1 & -1 & 1 \\ 1 & -1 & 1 & -1 & 1 & -1 \\ -1 & 1 & -1 & 1 & -1 & 1 \\ 1 & -1 & 1 & -1 & 1 & -1 \\ -1 & 1 & -1 & 1 & -1 & 1 \end{bmatrix}$$

Both have an optimal value of 9. In fact, any solution in the convex combination of the two solutions $\{X | \alpha \hat{X} + (1 - \alpha) X^*, \alpha \in [0, 1]\}$ would be an optimal SDP solution. However, those solutions have distinct performance on GW. For example, GW on $\hat{X}$ will return optimal rounding with probability 1 over the draw of the random hyperplane, whereas GW on $X^*$ will not be as consistent, see Figure 5b.

## C    Additional Experiment

### C.1    Additional Max-Cut Results

We showcase additional max-cut experimental results in Figure 6, where instead of plotting the subsampled GW cut density, we plot the subsampled SDP objective and 0.878 times the subsampled SDP objective. In all cases, the cut density from the Greedy algorithm on subsampled graphs converges to that of the full graph, aligning with our theoretical results. Moreover, the cut density returned by GW on the full graph lies between $\frac{0.878}{t^2} \cdot \mathsf{SDP}(G[S_t])$ and $\frac{1}{t^2} \mathsf{SDP}(G[S_t])$, even for small subgraphs $G[S_t]$. Thus, by computing $\mathsf{SDP}(G[S_t])$ and running Greedy on a subgraph, we can infer that Greedy and GW yield similar cut densities on these graphs.

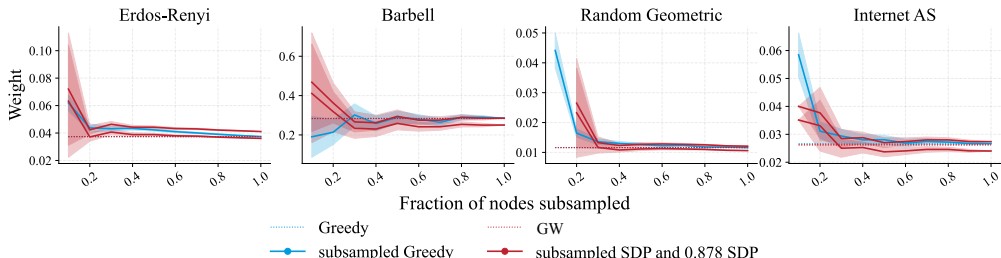

Figure 6: The Greedy algorithm's accuracy on the subsample approaches that of the algorithm on the full instance. The subsampled SDP objective value and 0.878 of the subsampled SDP objective value provide a more and more accurate prediction of the GW cut density as the sample size grows.

In Table 1, we demonstrate that even modest levels of subsampling substantially accelerate algorithm selection for the combinatorial problems under study. For example, on max-cut instances with $n = 500$ nodes, we report average solve times (in seconds) across subsample fractions $0.2, 0.4, 0.6, 0.8$, and $1.0$. We find that (1) an 80% subsample reduces GW's runtime by over 60% (from 961 s to 370 s), (2) a 60% subsample achieves roughly a 90% reduction (to 72 s), and (3) even the relatively fast Greedy algorithm exhibits noticeable speedups.

Table 1: Runtime (s) vs. Fraction of nodes sampled

| Method | 0.2 | 0.4 | 0.6 | 0.8 | 1.0 |
|--------|--------|---------|---------|----------|----------|
| GW | 0.5938 | 12.3565 | 72.2823 | 370.3707 | 961.4030 |
| Greedy | 0.0187 | 0.0834 | 0.1575 | 0.3231 | 0.5781 |

## C.2 Additional Discussion of Clustering Results

In this section, we provide additional discussion of our main results.

### C.2.1 Empirical performance of SoftmaxCenters

Surprisingly, our empirical results (Section 4) show that for several datasets, SoftmaxCenters achieves a *lower* objective value than Gonzalez's heuristic, even for $\beta = O(1)$. Intuitively, this is because Gonzalez's heuristic is pessimistic: it assumes that at each step, the best way to reduce the objective value is to put a center at the farthest point. In practice, a smoother heuristic may yield a lower objective value (a phenomenon also observed by García-Díaz et al. [45]).

As discussed in Section 4, Figure 7 compares the $k$-centers objective value of Gonzalez's algorithm and our softmax $k$-centers algorithm (averaged over $10^4$ repetitions) as a function of $\beta$ for a randomly sampled instance. The objective value attained by softmax algorithm closely approximates or improves over Gonzalez's algorithm even for small values of $\beta$. Figure 8 compares the $k$-centers objective value of Gonzalez's algorithm softmax (averaged over $10^5$ repetitions) as a function of $n$ for fixed $\beta = 1$ on randomly sampled GM, MNIST, and NC instances of $n$ points. Even with *fixed $\beta$*, the objective value of softmax continues to closely track that of Gonzalez's algorithm as $n$ increases.

## C.3 An adaptive subsampling scheme for algorithm selection

We demonstrate that random uniform subsampling provides an empirically stable approach to algorithm selection, even without access to the data-dependent quantities required by theoretical bounds. Instead of relying on these quantities, we propose an adaptive subsample algorithm selection substitute: run the target algorithms on randomly selected subsets of increasing sizes, compare their objective values or rankings across consecutive subset sizes, and terminate once the performance curve stabilizes.

To validate this approach, we conducted experiments on the Max-Cut problem across four graph families: random geometric, Erdős–Rényi, barbell, and Barabási–Albert graphs. For each instance,

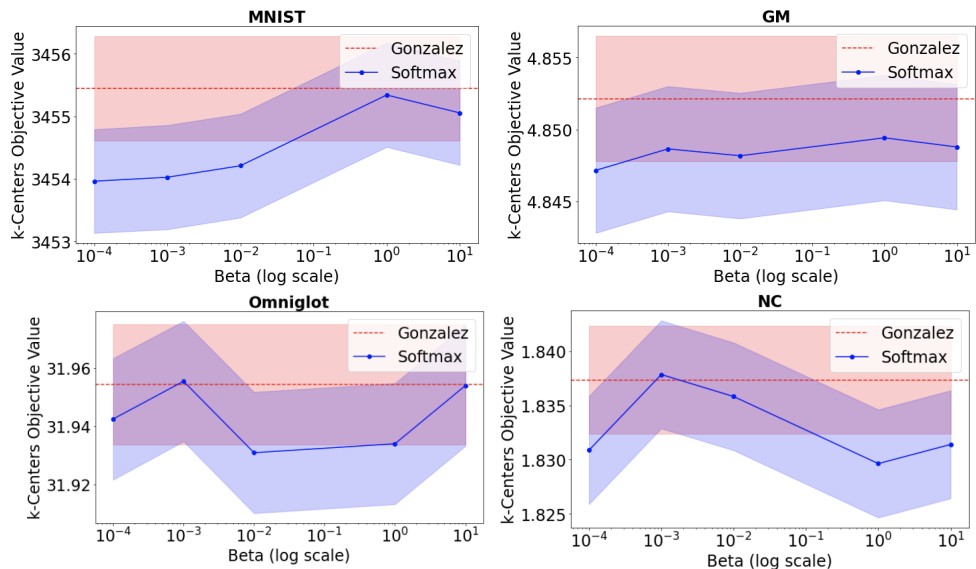

Figure 7: Objective value of softmax algorithm and of Gonzalez's algorithm vs. $\beta$. The plot shows 95% confidence intervals for the algorithms' $k$-centers objective values on randomly sampled instances. The softmax algorithm matches or improves over the objective value achieved by Gonzalez's algorithm, even for small $\beta$.

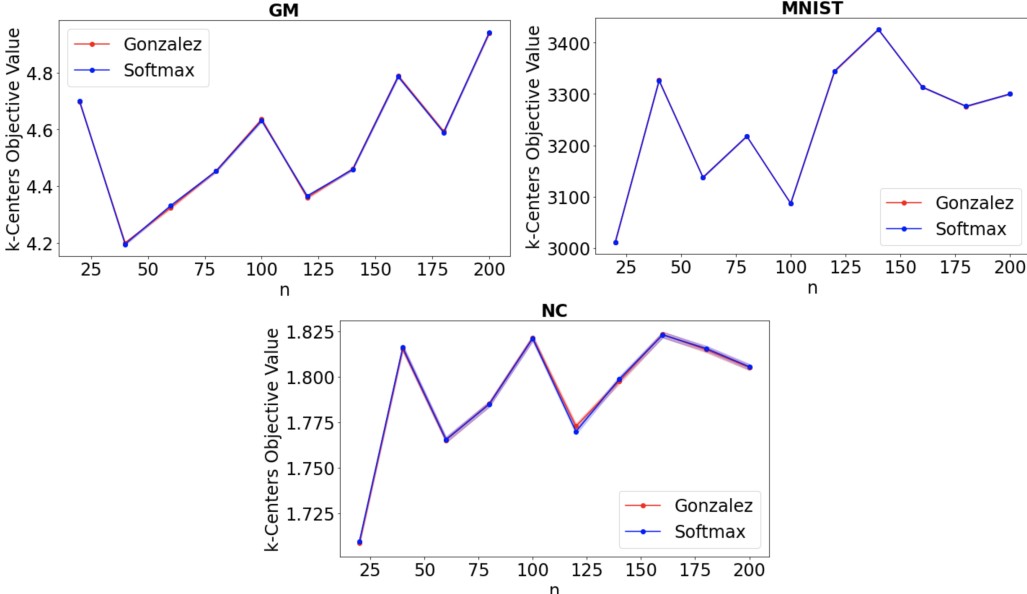

Figure 8: Objective of softmax algorithm and of Gonzalez's algorithm vs. $n$ for $\beta = 1$. The plot shows 95% confidence intervals for the algorithms' $k$-centers objective values on randomly sampled instances of size $n$. The relative performance of softmax tracks the performance of Gonzalez's algorithm extremely well as $n$ grows despite keeping the temperature $\beta$ fixed. (Omniglot was omitted in this figure because the instances are small.)

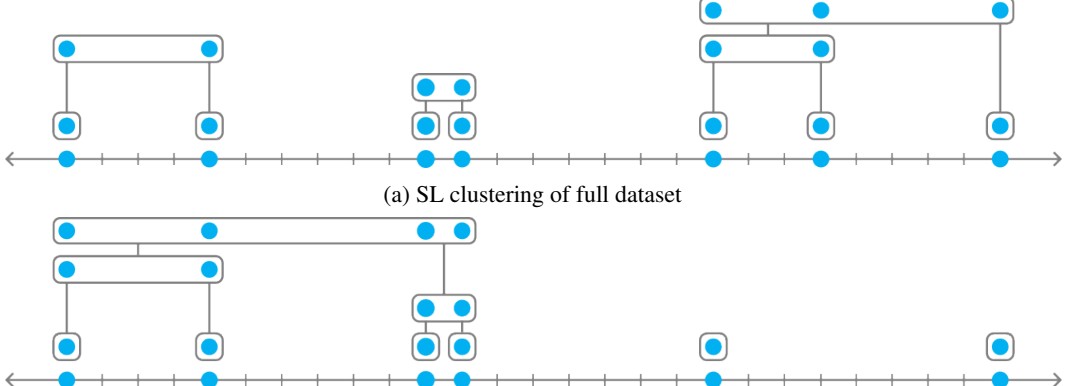

(a) SL clustering of full dataset

(b) SL clustering with one missing point

Figure 9: Figure 9a shows the SL clustering of $\mathcal{X}$, with three clusters $C_1$, $C_2$, and $C_3$ (left to right). In this example, $\zeta_{3,\mathsf{SL}}(\mathcal{X}) = n$, as $d_{\mathsf{mm}}(C_1 \cup C_2; \mathcal{X}) = 6$ and $d_{\mathsf{mm}}(C_3; \mathcal{X}) = 5$. There is a single point whose deletion dramatically changes the structure of the clustering in Figure 9b.

we sampled subsets of $2, 4, 8, \ldots, 64$ nodes from a total of $100$, solved each using both the Greedy and Goemans–Williamson (GW) algorithms, and averaged the objective values over $10$ trials. As shown in Table 2, in all cases, the relative ranking between Greedy and GW stabilized with subsets of size between 8 and 16.

Table 2: Empirical results of an adaptive subsample algorithm selection method, showcasing the stability of uniform subsampling.

| Graph type | Algorithm | 0.02 | 0.04 | 0.08 | 0.16 | 0.32 | 0.64 | Full |
|---|---|---|---|---|---|---|---|---|
| Random geometric | Greedy | 1.0 | 3.6 | 15.5 | 61.8 | 244.2 | 972.7 | 2381.8 |
| | GW | 1.0 | 3.6 | 15.2 | 60.0 | 239.6 | 997.3 | 2376.9 |
| | Best | both | both | Greedy | Greedy | Greedy | GW | Greedy |
| Erdős–Rényi | Greedy | 0.0 | 1.0 | 4.0 | 8.4 | 40.9 | 104.6 | 323.1 |
| | GW | 0.0 | 1.0 | 5.0 | 9.0 | 43.7 | 140.6 | 339.8 |
| | Best | both | both | GW | GW | GW | GW | GW |
| Barbell | Greedy | 0.4 | 2.1 | 7.1 | 29.6 | 126.3 | 511.5 | 1254.7 |
| | GW | 0.4 | 2.1 | 6.5 | 28.2 | 122.9 | 506.5 | 1249.5 |
| | Best | both | both | Greedy | Greedy | Greedy | Greedy | Greedy |
| Barabási–Albert | Greedy | 0.2 | 0.3 | 3.2 | 11.5 | 34.3 | 130.1 | 307.6 |
| | GW | 0.2 | 0.3 | 3.3 | 12.9 | 36.8 | 141.2 | 328.2 |
| | Best | both | both | GW | GW | GW | GW | GW |

# D  Additional Theoretical Discussion

## D.1  Sample complexity bound for MCMC k-means++

Here, for completeness, we state the value of $\zeta_{k,\mathsf{k\text{-}means++}}$ derived in [7].

**Theorem D.1** (Theorems 4 and 5 by Bachem et al. [7])**.** *Let $F$ be a distribution over $\mathbb{R}^d$ with expectation $\tilde{\mu} \in \mathbb{R}^d$ and variance $\sigma^2 := \int_x d(x, \tilde{\mu})^2 F(x)dx$ and support almost-surely bounded by an $R$-radius sphere. Suppose there is a $d'$-dimensional, $r$-radius sphere $B$ such that $d' > 2$, $F(B) > 0$, and $\forall x, y \in B$, $F(x) < cF(y)$ for some $c \geq 1$ (i.e., $F$ is non-degenerate). For $n$ sufficiently large, with high probability over $\mathcal{X} \sim F^n$, $\zeta_{k,\mathsf{k\text{-}means++}}(\mathcal{X}) = \mathcal{O}\left(R^2 k^{\min\{1, 4/d'\}}/(c\sigma^2 F(B)r^2)\right)$.*

## D.2  Additional discussion of single linkage results from Section 2.2

In this section, we include additional discussion of our single linkage results.

### D.2.1  Interpretation of $\zeta_{k,\mathsf{SL}}$

In this section, we describe why we interpret $\zeta_{k,\mathsf{SL}}(\mathcal{X})$ as a natural measure of the stability of $\mathsf{SL}$ on the dataset $\mathcal{X}$ with respect to random deletions. First, we emphasize that $\zeta_{k,\mathsf{SL}}$ is a property of how *robust* single linkage is on the dataset. It is *not* a measure of how accurately $\mathsf{SL}$ performs with respect to the ground truth and is defined completely independently of the ground truth.

First, we bound the range of $\zeta_{k,\mathsf{SL}}$.

**Lemma D.2.** $\zeta_{k,\mathsf{SL}} \in (0, n]$.

*Proof.* Note that $C_i, C_j, C_t \in \mathcal{C}$ are clusters in the $k$-clustering of $\mathcal{X}$ while $C_i \cup C_j$ intersects two clusters ($C_i$ and $C_j$) contained in the $k$-clustering of $\mathcal{X}$. So, Lemma 2.6 ensures that $\min_{i,j \in [k]} d_{\mathsf{mm}}(C_i \cup C_j; \mathcal{X}) - \max_{t \in [k]} d_{\mathsf{mm}}(C_t; \mathcal{X}) > 0$. $\qquad\square$

Suppose $\zeta_{k,\mathsf{SL}}(\mathcal{X}) > \alpha \cdot n$ for some $\alpha \in (0, 1]$. By rearranging the expression for $\zeta_{k,\mathsf{SL}}(\mathcal{X})$, we see that $\min_{i,j \in [k]} d_{\mathsf{mm}}(C_i \cup C_j; \mathcal{X}) > \frac{1}{\alpha} \cdot \max_{t \in [k]} d_{\mathsf{mm}}(C_t; \mathcal{X})$. That is, the distance threshold required for merging $C_i \cup C_j$ into a single cluster is not too large relative to the distance threshold required to merge $C_t$ into a single cluster. This is illustrated in Figure 9a with $i = 1$, $j = 2$, and $t = 3$. When we subsample from $\mathcal{X}$ to build $\mathcal{X}_m$, there is some probability that we distort the min-max distance so that $d_{\mathsf{mm}}(C_t; \mathcal{X}_m) > d_{\mathsf{mm}}(C_i \cup C_j; \mathcal{X}_m)$, causing $\mathsf{SL}(\mathcal{X}_m, k)$ to output a clustering that *contains* $C_i \cup C_j$ but does *not contain* $C_t$, as in Figure 9b[1]. If this occurs, we cannot expect size generalization to hold. When $\alpha$ is larger, the difference between $d_{\mathsf{mm}}(C_t; \mathcal{X})$ and $d_{\mathsf{mm}}(C_i \cup C_j; \mathcal{X}_m)$ is smaller, and thus there is a higher probability $d_{\mathsf{mm}}(C_t; \mathcal{X}_m) > d_{\mathsf{mm}}(C_i \cup C_j; \mathcal{X}_m)$. To control the probability of this bad event, we must take a larger value of $m$ in order to obtain size generalization with high probability.

We emphasize that $\zeta_{k,\mathsf{SL}}$ measures the stability of $\mathsf{SL}$ on $\mathcal{X}$ with respect to point deletion, but it is *fundamentally unrelated* to the cost of $\mathsf{SL}$ on $\mathcal{X}$ with respect to a ground truth (which is what we ultimately aim to approximate). Any assumption that $\zeta_{k,\mathsf{SL}}$ is small relative to $n$ *does not constitute any assumptions* that $\mathsf{SL}$ achieves good accuracy on $\mathcal{X}$. Theorem 2.7 states our main result for size generalization of single linkage clustering.

### D.2.2  Tightness of single-linkage analysis

In this section, we show that our dependence on the cluster size $\min_{i \in [k]} |C_i|$ and $\zeta_{k,\mathsf{SL}}$ in Theorem 2.7 is necessary. We also discuss further empirical analysis of $\zeta_{k,SL}$.

**Dependence on cluster size** $\min_{i \in [k]} |C_i|$.  Suppose our subsample entirely misses one of the clusters in $\mathcal{C} = \mathsf{SL}(\mathcal{X}, k)$, (i.e., $\mathcal{X}_m \cap C_i = \emptyset$). Then $\mathsf{SL}(\mathcal{X}_m, k)$ will cluster $\mathcal{X}_m$ into $k$ clusters, while $\mathsf{SL}(\mathcal{X}, k)$ partitions $\mathcal{X}_m$ into at most $k - 1$ clusters. Consequently, we can design ground truth on which clustering costs of $\mathcal{C}$ and $\mathcal{C}'$ vary dramatically.

**Lemma D.3.** *For any odd $n \geq 110$ and $\gamma > 0$, there exists a 2-clustering instance on $n$ nodes with a ground truth clustering $\mathcal{G}$ such that $\min_{i \in [k]} |C_i| = 1$, $\zeta_{k,\mathsf{SL}}(\mathcal{X}) \leq \gamma$ and with probability .27 for $m = n - 1$, $|\mathsf{cost}_\mathcal{G}(\mathsf{SL}(\mathcal{X}_m, 2)) - \mathsf{cost}_\mathcal{G}(\mathsf{SL}(\mathcal{X}, 2))| \geq 1/4$.*

**Dependence on $\zeta_{k,\mathsf{SL}}$.**  The dependence on $\zeta_{k,\mathsf{SL}}(\mathcal{X})$ is also necessary and naturally captures the instability of $\mathsf{SL}$ on a given dataset $\mathcal{X}$, as depicted in Figure 9. Lemma D.4 formalizes a sense in which the dependence on the relative merge distance ratio $\zeta_{k,\mathsf{SL}}$ is unavoidable.

**Lemma D.4.** *For any $n \geq 51$ with $n = 1$ mod 3, there exists a 2-clustering instance $\mathcal{X}$ on $n$ points such that $\zeta_{k,\mathsf{SL}} = n$; $\mathcal{C} = \mathsf{SL}(\mathcal{X}, 2)$ satisfies $\min_{C \in \mathcal{C}} |C| \geq \frac{n}{6}$; and for $m = n - 1$, with probability .23, $|\mathsf{cost}_\mathcal{G}(\mathsf{SL}(\mathcal{X}_m, 2)) - \mathsf{cost}_\mathcal{G}(\mathsf{SL}(\mathcal{X}, 2))| \geq 1/12$.*

**Empirical study of $\zeta_{k,\mathsf{SL}}$.**  We observe that for some natural distributions, such as when clusters are drawn from isotropic Gaussian mixture models, we can expect $\zeta_{k,\mathsf{SL}}$ to scale with the variance of the Gaussians. Intuitively, when the variance is smaller, the clusters will be more separated, so $\zeta_{k,\mathsf{SL}}$ will be smaller.

---

[1]The worst-case scenario in Figure 9b where a small number of "bridge" points connect two sub-clusters has been noted as a failure case for single-linkage style algorithms in previous work [e.g., 29, Section 3.1].

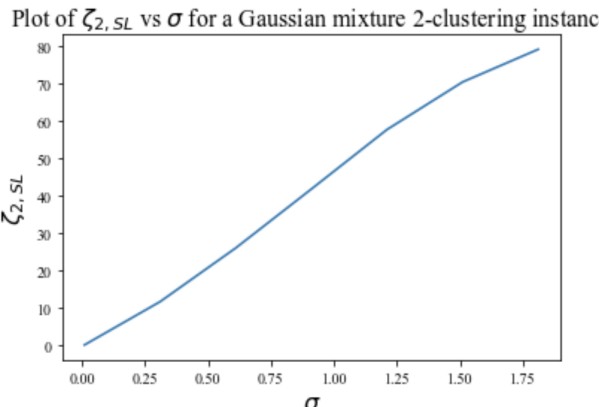

Figure 10: $\zeta_{k,\mathsf{SL}}$ vs $\sigma$ for isotropic Gaussian mixture model. The $y$-axis shows the average value of $\zeta_{k,\mathsf{SL}}(\mathcal{X})$ averaged over 30 draws of $\mathcal{X}$ for a given choice of $\sigma$.

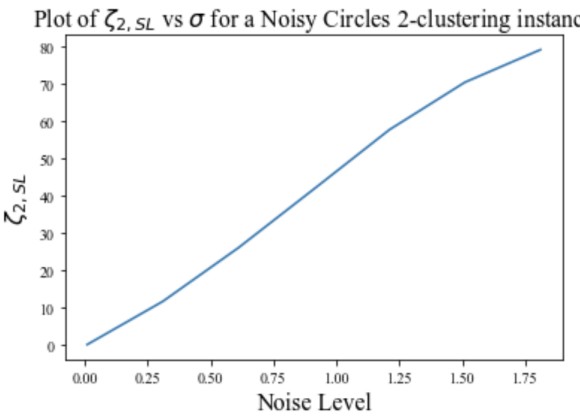

Figure 11: $\zeta_{k,\mathsf{SL}}$ vs $\sigma$ for noisy circle datasets. The $y$-axis shows the average value of $\zeta_{k,\mathsf{SL}}(\mathcal{X})$ averaged over 30 draws of $\mathcal{X}$ for a given choice of $\sigma$.

We ran experiments to visualize $\zeta_{k,SL}(\mathcal{X})$ versus appropriate notions of noise in the dataset for two natural toy data generators. First, we considered isotropic Gaussian mixture models in $\mathbb{R}^2$ consisting of two clusters centered at $(-2, -2)$ and $(2, 2)$ respectively with variance $\sigma^2 I$ (Figure 10). Second, we considered 2-clustering instances drawn from scikitlearn's noisy circles dataset with a distance factor of .5 and [69] with various noise levels (Figure 11). In both cases, $n = 300$ points were drawn. Moreover, in both cases, we see that, as expected, $\zeta_{k,\mathsf{SL}}$ grows with the noise in the datasets.

# E Omitted Proofs

In Section E.1, we include proofs pertaining to our analysis of center-based clustering methods ($k$-means++ and $k$-centers). In Section E.2, we include omitted proofs pertaining to our analysis of single linkage clustering. In Section E.3 and Section E.4, we include omitted proofs of our analysis of the max-cut GW and greedy algorithms from Section 3.1 and Section 3.2.

## E.1 Omitted Proofs from Section 2.1

In this section, we provide omitted proofs relating to our results on $k$-centers and $k$-means++ clustering. Throughout this section, we denote the total variation (TV) distance between two (discrete) distributions $p, q : \Omega \to [0,1]$ is $\|p - q\|_{TV} := \frac{1}{2} \sum_{\omega \in \Omega} |p(\omega) - q(\omega)|$. If $p(x, y)$ is the joint distribution of random variables $X$ and $Y$, $p_X(x)$ and $p_Y(y)$ denote the marginal distributions, and $p_{X|Y}(x|y)$ or $p_{Y|X}(y|x)$ denote the conditional distributions of $X|Y$ or $Y|X$, respectively.

First, we specify the pseudocode of Seeding algorithm, Algorithm 3.

**Input:** Instance $\mathcal{X} \subset \mathbb{R}^d$, $k \in \mathbb{Z}_{\geq 0}$, $f : \mathbb{R} \times \mathcal{P}(\mathcal{X}) \to \mathbb{R}_{\geq 0}$
Set $C^1 = \{c_1\}$ with $c_1 \sim \text{Unif}(\mathcal{X})$
**for** $i = 2, 3, \ldots, k$ **do**
  Select $c_i = y \in \mathcal{X}$ with probability proportional to $f(d_{\text{center}}(y; C^{i-1}); \mathcal{X})$
  Set $C^i = C^{i-1} \cup \{x\}$
**Return:** $C^k$

**Algorithm 3:** $\text{Seeding}(\mathcal{X}, k, f)$

Let $p, q$ be arbitrary, discrete joint distributions over random variables $X$ and $Y$ with conditionals and marginals denoted as follows: $p_{X,Y}(x, y) = p_X(x) \cdot p_{Y|X}(y|x)$ and $q_{X,Y}(x, y) = q_X(x) \cdot q_{Y|X}(y|x)$. If $\|p_X - q_X\|_{TV} \leq \epsilon_1$ and $\|p_{Y|X} - q_{Y|X}\|_{TV} \leq \epsilon_2$, then $\|p_{X,Y} - q_{X,Y}\|_{TV} \leq \epsilon_1 + \epsilon_2$.

*Proof.* For any $x, y$ we have

$$
\begin{aligned}
|p_{X,Y}(x, y) - q_{X,Y}(x, y)| &= \left| p_X(x) \cdot p_{Y|X}(y|x) - q_X(x) \cdot q_{Y|X}(y|x) \right| \\
&= \left| p_X(x) \cdot p_{Y|X}(y|x) - (q_X(x) - p_X(x)) \cdot q_{Y|X}(y|x) - p_X(x) \cdot q_{Y|X}(y|x) \right| \\
&= \left| p_X(x) \cdot (p_{Y|X}(y|x) - q_{Y|X}(y|x)) + q_{Y|X}(y|x) \cdot (q_X(x) - p_X(x)) \right| \\
&\leq p_X(x) \cdot \left| p_{Y|X}(y|x) - q_{Y|X}(y|x) \right| + q_{Y|X}(y|x) \cdot |q_X(x) - p_X(x)|
\end{aligned}
$$

Hence,

$$
\begin{aligned}
\|p_{X,Y} - q_{X,Y}\|_{TV} &= \frac{1}{2} \sum_{x,y} |p_{X,Y}(x, y) - q_{X,Y}(x, y)| \\
&\leq \sum_x p_X(x) \frac{1}{2} \sum_y \left| p_{Y|X}(y|x) - q_{Y|X}(y|x) \right| \\
&\quad + \frac{1}{2} \sum_x |q_X(x) - p_X(x)| \sum_y q_{Y|X}(y|x) \\
&\leq \epsilon_2 + \epsilon_1.
\end{aligned}
$$

$\square$

**Theorem 2.2.** *Let* $\mathcal{X} \subset \mathbb{R}^d$, $\epsilon, \epsilon' > 0, \delta \in (0, 1)$, *and* $k \in \mathbb{Z}_{>0}$. *Define the sample complexity* $m = \mathcal{O}(\zeta_{k,f}(\mathcal{X}) \log(k/\epsilon))$ *where* $\zeta_{k,f}(\mathcal{X})$ *quantifies the sampling distribution's smoothness:*

$$
\zeta_{k,f}(\mathcal{X}) := \max_{Q \subset \mathcal{X} : |Q| \leq k} \max_{x \in \mathcal{X}} \left( n f(d_{\text{center}}(x; Q); \mathcal{X}) \cdot \sum_{y \in \mathcal{X}} f(d_{\text{center}}(y; Q); \mathcal{X})^{-1} \right).
$$

*Let* $S$ *and* $S'$ *be the partitions of* $\mathcal{X}$ *induced by* $\text{Seeding}(\mathcal{X}, k, f)$ *and* $\text{ApxSeeding}(\mathcal{X}_{mk}, k, m, f)$ *where* $\mathcal{X}_{mk}$ *is a sample of* $mk$ *points drawn uniformly with replacement from* $\mathcal{X}$. *For any ground-truth clustering* $\mathcal{G}$ *of* $\mathcal{X}$, $\mathbb{E}[\text{cost}_{\mathcal{G}}(S')] \approx_\epsilon \mathbb{E}[\text{cost}_{\mathcal{G}}(S)]$. *Moreover, given* $S'$, $\text{cost}_{\mathcal{G}}(S')$ *can be estimated to additive error* $\epsilon'$ *with probability* $1 - \delta$ *using* $\mathcal{O}(k\epsilon'^{-2} \log(\delta^{-1}))$ *ground-truth queries.*

*Proof.* Let $p(C)$ be the probability of sampling a set of $k$ centers $C$ in $\mathsf{ApxSeeding}(\mathcal{X}_{mk}, k, m, f)$ and $p'(C)$ be the probability of sampling a set of $k$ centers $C$ in $\mathsf{Seeding}(\mathcal{X}, k, f)$. Likewise, let $p(c_i|C^{i-1})$ and $p'(c_i|C^{i-1})$ be the probability of sampling $c_i$ as the $i$-th center given $C^{i-1}$ was selected up to the $(i-1)$-th step in $\mathsf{ApxSeeding}(\mathcal{X}_{mk}, k, m, f)$ and $\mathsf{Seeding}(\mathcal{X}, k, f)$ respectively. We have that

$$p(C) = \frac{1}{n} \prod_{i=2}^{k} p(c_i|C^{i-1}), \text{ and } p'(C) = \frac{1}{n} \prod_{i=2}^{k} p'(c_i|C^{i-1}).$$

By Corollary 1 of [27], $\mathsf{ApxSeeding}$ (Algorithm 1) with $m = O(\zeta_{k,f}(\mathcal{X}) \log(k/\epsilon))$ is such that for all $C^{i-1} \subset \mathcal{X}$ with $|C^{i-1}| \leq k - 1$,

$$\left\| p(\cdot|C^{i-1}) - p'(\cdot|C^{i-1}) \right\|_{TV} \leq \frac{\epsilon}{k-1}.$$

By chaining Fact E.1 over $i = 2, ..., k$, it follows that $\|p - p'\|_{TV} \leq \epsilon$. Now, let $p(Z|c_1)$ and $p'(Z|c_1)$ denote the probability of selecting centers $Z = \{z_2, ..., z_k\}$ in iterations $i = 2, ..., k$, conditioned on having selected $c_1$ in the first iteration of $\mathsf{Seeding}$ and $\mathsf{ApxSeeding}$ respectively. Then,

$$p'(Z|c_1) - |(p(Z|c_1) - p'(Z|c_1))| \leq p(Z|c_1) \leq p'(Z|c_1) + |(p(Z|c_1) - p'(Z|c_1))|.$$

Let $\mathcal{X}^{k-1}$ denote the set of all subsets of size $k-1$ in $\mathcal{X}$. In the following, for any set of centers $Z = z_1, ..., z_t$ we will use $\mathsf{cost}_{\mathcal{G}}(Z)$ to denote the cost of the clustering (Voronoi partition) induced by the centers $Z$. Using the fact that the proportion of mislabeled points is always between 0 and 1, we then have that

$$\mathbb{E}\left[\mathsf{cost}_{\mathcal{G}}(S)\right] = \sum_{c_1 \in \mathcal{X}} \frac{1}{n} \sum_{Z \in \mathcal{X}^{k-1}} \mathsf{cost}_{\mathcal{G}}(c_1 \cup Z) \, p(Z|c_1)$$

$$\leq \sum_{c_1 \in \mathcal{X}} \frac{1}{n} \sum_{Z \in \mathcal{X}^{k-1}} \mathsf{cost}_{\mathcal{G}}(c_1 \cup Z) \, p'(Z|c_1) + \frac{1}{n} \sum_{c_1 \in \mathcal{X}} \sum_{Z \in \mathcal{X}^{k-1}} \mathsf{cost}_{\mathcal{G}}(c_1 \cup Z) \, |(p(Z|c_1) - p'(Z|c_1))|$$

$$\leq \mathbb{E}\left[\mathsf{cost}_G(S')\right] + \frac{1}{n} \sum_{c_1 \in \mathcal{X}} \sum_{Z \in \mathcal{X}^{k-1}} |(p(Z|c_1) - p'(Z|c_1))| \leq \mathbb{E}\left[\mathsf{cost}_{\mathcal{G}}(S')\right] + \|p - p'\|_{TV}$$

$$\leq \mathbb{E}\left[\mathsf{cost}_{\mathcal{G}}(S')\right] + \epsilon.$$

By a symmetric argument, we obtain a lower bound

$$\mathbb{E}\left[\mathsf{cost}_{\mathcal{G}}(S)\right]$$

$$= \sum_{c_1 \in \mathcal{X}} \frac{1}{n} \sum_{Z \in \mathcal{X}^{k-1}} \mathsf{cost}_{\mathcal{G}}(c_1 \cup Z) \, p(Z|c_1)$$

$$\geq \sum_{c_1 \in \mathcal{X}} \frac{1}{n} \sum_{Z \in \mathcal{X}^{k-1}} \mathsf{cost}_{\mathcal{G}}(c_1 \cup Z) \, p'(Z|c_1) - \frac{1}{n} \sum_{c_1 \in \mathcal{X}} \sum_{Z \in \mathcal{X}^{k-1}} \mathsf{cost}_G(c_1 \cup Z) \, |(p(Z|c_1) - p'(Z|c_1))|$$

$$\geq \mathbb{E}\left[\mathsf{cost}_{\mathcal{G}}(S')\right] - \frac{1}{n} \sum_{c_1 \in \mathcal{X}} \sum_{Z \in \mathcal{X}^{k-1}} |(p(Z|c_1) - p'(Z|c_1))| \geq \mathbb{E}\left[\mathsf{cost}_{\mathcal{G}}(S')\right] - \|p - p'\|_{TV}$$

$$\geq \mathbb{E}\left[\mathsf{cost}_{\mathcal{G}}(S')\right] - \epsilon.$$

Next, we can compute $\hat{c}_{\mathcal{G}}(S')$ as follows. Let $\tau : \mathcal{X} \mapsto [k]$ be the mapping corresponding to the ground truth clustering (i.e., $\tau(x) = i$ if and only if $x \in G_i$.) Let $\mathcal{X}' = \{x_1, ..., x_\ell\}$ be a *fresh* random sample of $\ell$ elements selected without replacement from $\mathcal{X}$, and let

$$\hat{c}_{\mathcal{G}}(S') := \min_{\sigma \in \Sigma_k} \frac{1}{\ell} \sum_{x \in \mathcal{X}'} \sum_{i \in [k]} \mathbb{1}\left(x \in S'_{\sigma(i)} \wedge x \notin G_i\right).$$

Consider any $\sigma \in \Sigma_k$. Then,

$$\mathbb{E}\left[\frac{1}{\ell} \sum_{x \in \mathcal{X}'} \sum_{i \in [k]} \mathbb{1}\left(x \in S'_{\sigma(i)} \wedge x \notin G_i\right)\right] = \frac{1}{n} \sum_{x \in \mathcal{X}} \sum_{i \in [k]} \mathbb{1}\left(x \in S'_{\sigma(i)} \wedge x \notin G_i\right).$$

Hoeffding's inequality guarantees that with probability $\delta/k!$,

$$\frac{1}{\ell}\sum_{x\in\mathcal{X}'}\sum_{i\in[k]}\mathbb{1}\left(x\in S'_{\sigma(i)}\wedge x\notin G_i\right)\approx_\epsilon\frac{1}{n}\sum_{x\in\mathcal{X}}\sum_{i\in[k]}\mathbb{1}\left(x\in S'_{\sigma(i)}\wedge x\notin G_i\right)$$

for $l=\mathcal{O}(k\epsilon^{-2}\log(k\delta^{-1}))$. The theorem follows by union bound over all $k!$ permutations $\sigma\in\Sigma_k$. $\qquad\square$

**Theorem 2.3.** *Let $k\in\mathbb{Z}_{\geq 0}$, $\gamma>0$ and $\delta\in(0,1)$. Let $S_{\mathsf{OPT}}$ be the partition of $\mathcal{X}$ induced by the optimal $k$-centers solution $C_{\mathsf{OPT}}$, and suppose $\mathcal{X}$ is $(\mu_\ell,\mu_u)$-well-balanced with respect to $S_{\mathsf{OPT}}$. Let $C$ be the centers obtained by* SoftmaxCenters *with $\beta=R\gamma^{-1}\log\left(k^2\mu_u\mu_\ell^{-1}\delta^{-1}\right)$. With probability $1-\delta$, $\max_{x\in\mathcal{X}}d_{\mathsf{center}}(x;C)\leq 4\max_{x\in\mathcal{X}}d_{\mathsf{center}}(x;C_{\mathsf{OPT}})+\gamma$.*

*Proof.* Let $S_{\mathsf{OPT}}=\{S_1,...,S_k\}$ be the optimal partition according to the $k$-centers objective and let OPT denote the optimal $k$-centers objective value. Let $S_{\sigma(i)}\in S_{\mathsf{OPT}}$ be such that $c_i\in S_{\sigma(i)}$ where $c_i\in C$, and let $C^{i-1}\subset C$ denote the centers chosen in the first $i-1$ iterations of SoftmaxCenters. We use $S_{\iota(i)}\in S_{\mathsf{OPT}}$ to denote the partition such that $c_i\in S_{\iota(i)}$ where $c_i\in C$. We use $s_i\in S_i$ to denote the optimal $k$ centers, and use $C^{(i-1)}\subset C$ to denote the centers chosen in the first $i-1$ iterations of SoftmaxCenters. We will show that for any $i\in\{2,...,k-1\}$, either $m_i:=\max_{x\in\mathcal{X}}d(x;C^{(i-1)})\leq 4\mathsf{OPT}+\gamma$, or else with good probability, $c_i$ belongs to a different partition $S_{\iota(i)}$ than any of $c_1,...,c_{i-1}$ (i.e., $\iota(i)\neq\iota(j)$ for any $j<i$).

To this end, suppose $m_i>4\mathsf{OPT}+\gamma$, then let $B:=\{x:x\in S_{\iota(j)}\text{ for some }j<i\}$. For notational convenience, let $\beta'=\gamma^{-1}\log(k^2\mu_u\mu_\ell^{-1}\delta^{-1})$. We have

$$\mathbb{P}\{c_i\in B\}=\frac{\sum_{x\in B}\exp(\beta'd(x;C^{(i-1)}))}{\sum_{x\in\mathcal{X}}\exp(\beta'd(x;C^{(i-1)}))}.$$

For every $j\in[k]$, we know that $c_j,s_{\iota(j)}\in S_{\iota(j)}$ (each point goes to the same partition as its closest center.) By the triangle inequality, it follows that for any $x\in S_{\iota(j)}$,

$$d(x;C^{(i-1)})\leq d(x,c_j)+d(c_j,s_{\iota(j)})\leq 2\mathsf{OPT}.$$

Consequently, using the fact that $g(z)=z/(z+c)$ is an increasing function of $z$ for $c>0$, we have

$$\mathbb{P}\{c_i\in B\}\leq\frac{|B|\exp(2\beta'\mathsf{OPT})}{|B|\exp(2\beta'\mathsf{OPT})+\sum_{x\in\mathcal{X}\setminus B}\exp(\beta'd(x;C^{(i-1)}))}$$

Now, let $\ell\in[k]$ be the index of the cluster in which $\exists y\in S_\ell$ with

$$d(y;C^{(i-1)})=m_i>4\mathsf{OPT}+\gamma.\tag{4}$$

Such an $\ell$ is guaranteed to exist, due to the assumption that $m_i>4\mathsf{OPT}+\gamma$. Now, for $x\in S_\ell$ and $j<i$,

$$d(x;c_j)+d(x,y)\geq d(y;c_j)\geq 4\mathsf{OPT}+\gamma,$$
$$d(x,y)\leq 2\mathsf{OPT},$$

and hence $d(x,c_j)\geq 2\mathsf{OPT}+\gamma$. Now, note that $S_\ell\cap B=\emptyset$, because by the definition of $B$, whenever $S_\ell\cap B\neq\emptyset$ we must have $S_\ell\cap B=S_\ell$ and consequently $d(y;C^{(i-1)})\leq d(y,c^j)+d(c^j,s_{\iota(j)})\leq 2\mathsf{OPT}$ (contradicting (4)). So, it follows that

$$\mathbb{P}\{c_i\in B\}\leq\frac{|B|\exp(2\beta'\mathsf{OPT})}{|B|\exp(2\beta'\mathsf{OPT})+|S_\ell|\exp(\beta'(4\mathsf{OPT}+\gamma))}=\frac{1}{1+\frac{|S_\ell|}{|B|}\exp(\beta'(4\mathsf{OPT}+\gamma)-2\mathsf{OPT})}$$

$$\leq\frac{|B|}{|S_\ell|}\exp(-\beta'\gamma)\leq k\mu_u/\mu_\ell\exp(-\beta'\gamma)=\frac{\delta}{k}.$$

Now, by union bounding over $i=2,...,k$ we see that with probability at least $1-\delta$, either $m_i\leq 4\mathsf{OPT}+\gamma$ for some $i<[k]$; or, $\iota(i)\neq\iota(j)$ for any $j<i$.

In the former case, the approximation guarantee is satisfied. In the latter case, note that if every $c_i$ belongs to a distinct optimal cluster $S_{\iota(i)}$, then every $x\in S_{\iota(i)}$ for some $i\in[k]$. Consequently, by the triangle inequality, we have that for any $x\in S_{\iota(i)}$

$$d(x;C)\leq d(x;c_i)\leq d(x,s_{\iota(i)})+d(c_i;s_{\iota(i)})\leq 2\mathsf{OPT}<4\mathsf{OPT}+\gamma.$$

$\qquad\square$

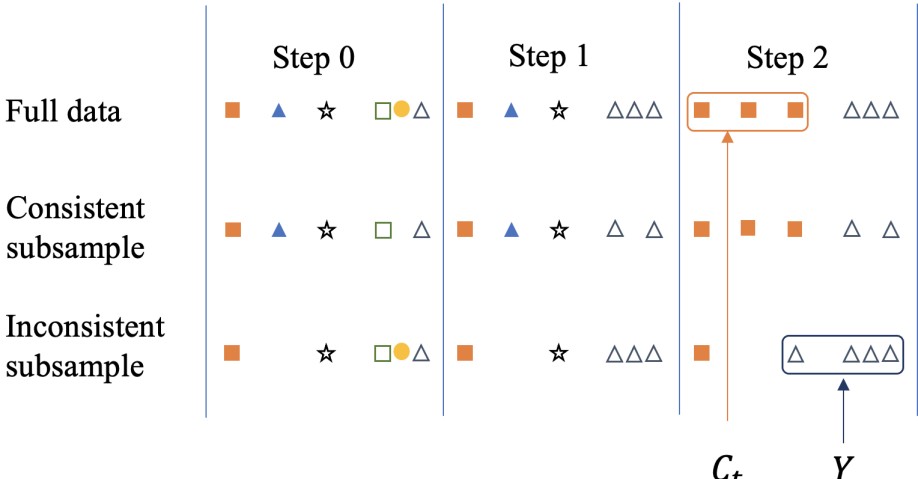

Figure 12: Examples of subsamples which induce a consistent and inconsistent clustering. The figure shows an example dataset $\mathcal{X}$ in one dimension along with the clustering $\mathcal{C}$ obtained by $\mathsf{SL}(\mathcal{X}, 2)$ and $\mathsf{SL}(\mathcal{X}_m, 2)$ on two different subsamples. Observe that second subsample produces an inconsistent clustering because the set $Y$ (1) has a nonempty intersection with both of the clusters from the full data and (2) gets merged into a single cluster at Step 4, before all points from $C_t$ have been merged together.

**Theorem 2.4.** *For any $\beta > 0$, $\zeta_{k,\mathsf{SoftmaxCenters}}(\mathcal{X}) \leq \exp(2\beta)$.*

*Proof.* For any $Q$, let $x_Q^\star = \operatorname{argmax}_{x \in \mathcal{X}} \exp(\beta d_{\mathsf{center}}(x; Q))$. Then, we can see that

$$\sum_{y \neq x_Q^\star} \exp(\beta d_{\mathsf{center}}(y; Q)/R) + \exp(\beta d_{\mathsf{center}}(x_Q^\star; Q)/R) \geq (n-1) + \exp(\beta d_{\mathsf{center}}(x_Q^*, Q)/R).$$

Meanwhile, $\exp(\beta d_{\mathsf{center}}(x_Q^*, Q)/R) \leq \exp(2\beta)$. Since $h(x) = \frac{x}{(n-1)+x}$ and $g(x) = \frac{xa}{(x-1)+a}$ are increasing in $x$ for $a > 1$, we find

$$\zeta_{k,\mathsf{SoftmaxCenters}}(\mathcal{X}) := n \cdot \max_{Q \subset \mathcal{X}: |Q| \leq k} \max_{x \in \mathcal{X}} \frac{\exp(\beta d_{\mathsf{center}}(x; Q))}{\sum_{y \in Q} \exp(\beta d_{\mathsf{center}}(y; Q))}$$

$$\leq n \cdot \frac{\exp(2\beta)}{(n-1) + \exp(2\beta)} \leq \exp(2\beta).$$

$\square$

**Lemma E.1.** *For any $n > \exp(\beta R) - 1$, there is a clustering instance $\mathcal{X} \subset [-R, R]$ of size $n$ such that $\zeta_{k,\mathsf{SoftmaxCenters}}(\mathcal{X}) \geq \exp(\beta)/2$.*

*Proof.* Consider $\mathcal{X} \subset [-R, R]$ such that $x_1 = R$ and $x_2, ..., x_n = 0$. By taking $Q = \{q_1 = 0, ..., q_k = 0\}$, we see that $\zeta_{k,\mathsf{SoftmaxCenters}}(\mathcal{X}) \geq \frac{n \exp(\beta)}{n-1+\exp(\beta)} \geq \frac{n \exp(\beta)}{2(n-1)} \geq \exp(\beta)/2$. $\square$

### E.2 Omitted Proofs from Section 2.2

In Section E.2.1, we present omitted proofs pertaining to size generalization of single linkage. In Section E.2.2, we present omitted proofs of our lower bounds.

### E.2.1 Size generalization of single linkage clustering

In this section, we provide omitted proofs relating to our results on single linkage clustering. We show that under natural assumptions on $\mathcal{X}$, running $\mathsf{SL}$ on $\mathcal{X}$ yields similar accuracy as $\mathsf{SL}$ on a uniform random subsample $\mathcal{X}_m$ of $\mathcal{X}$ of size $m$ (drawn with replacement) for $m$ sufficiently large.

We approach this analysis by showing that when $m$ is sufficiently large, the order in which clusters are merged in $\mathsf{SL}(\mathcal{X}, k)$ and $\mathsf{SL}(\mathcal{X}_m, k)$ is similar with high probability. Concretely, we show that for any subsets $S, T \subset \mathcal{X}_m$, the order in which $S$ and $T$ are merged is similar when we run $\mathsf{SL}(\mathcal{X}, k)$ and $\mathsf{SL}(\mathcal{X}_m, k)$. We use $d_i$ (respectively $d_{mi}$) to denote the merge distance at iteration $i$ when running $\mathsf{SL}(\mathcal{X}, k)$ (respectively $\mathsf{SL}(\mathcal{X}_m, k)$). Similarly, $\mathcal{C}^i$ (respectively $\mathcal{C}_m^i$) denotes the clustering at iteration $i$. We use $g_m(S)$ to denote the first iteration at which all points in $S$ are merged into a common cluster (we refer to this as the *merge index* of $S$). That is, $g_m : \mathcal{P}(\mathcal{X}_m) \to [n-1]$ is a mapping such that for any $S \subset \mathcal{X}_m$, $g_m(S) = t$, where $t$ is the first iteration in $\mathsf{SL}(\mathcal{X}_m, k)$ such that $S \subset C$ for some $C \in \mathcal{C}_m^t$. Correspondingly, we use $d_{g(S)}$ and $d_{m\,g_m(S)}$ denote the *merge distance* of $S \subset \mathcal{X}$ when running $\mathsf{SL}$ on $\mathcal{X}$ and $\mathcal{X}_m$ respectively. Table 3 summarizes the notation used in our analysis.

| Notation | Informal Description | Formal Definition |
|---|---|---|
| $g(S)$ | The first iteration of the outer loop in $\mathsf{SL}(\mathcal{X}, k)$ at which all points in $S$ are merged into a single cluster. | For $S \subset \mathcal{X}$, $g(S) = \min\{\ell \in \mathbb{Z}_{\geq 0} : \exists C \subset \mathcal{C}^\ell$ such that $S \subset C\}$ when running $\mathsf{SL}(\mathcal{X}, k)$. |
| $g_m(S)$ | The first iteration of the outer loop in $\mathsf{SL}(\mathcal{X}_m, k)$ at which all points in $S$ are merged into a single cluster. | For $S \subset \mathcal{X}_m$, $g_m(S) = \min\{\ell \in \mathbb{Z}_{\geq 0} : \exists C \subset \mathcal{C}^\ell$ such that $S \subset C\}$ when running $\mathsf{SL}(\mathcal{X}_m, k)$. |
| $d_\ell$ | The merge distance at iteration $\ell$ of the outer loop in $\mathsf{SL}(\mathcal{X}, k)$ | $\max_{i,j} \min_{x \in C_i^{(\ell-1)}, y \in C_j^{(\ell-1)}} d(x, y)$. when running $\mathsf{SL}(\mathcal{X}, k)$. |
| $d_{m_\ell}$ | The merge distance at iteration $\ell$ of the outer loop in $\mathsf{SL}(\mathcal{X}_m, k)$ | $\min_{i,j} \max_{x \in C_i^{(\ell-1)}, y \in C_j^{(\ell-1)}} d(x, y)$. when running $\mathsf{SL}(\mathcal{X}_m, k)$. |

Table 3: Single linkage clustering analysis notation. The table summarizes key notation used in our analysis of single linkage clustering

We first prove Lemma 2.6, which characterizes when two points will be merged into a common cluster in single linkage clustering.

**Lemma E.2.** *In $\mathsf{SL}(\mathcal{X}, k)$, $x, y \in \mathcal{X}$ belong to the same cluster after iteration $\ell$ if and only if $d_{\mathsf{mm}}(x, y; \mathcal{X}) \leq d_\ell$.*

*Proof.* For the forward direction, we will induct on the size of the cluster $C$. In the base case, if $|C| = 2$, $C$ must be the result of merging some two clusters $A = \{x_i\}, B = \{x_j\}$ such that $d(A, B) = d(x_i, x_j) \leq d_t \leq d_\ell$ at some iteration $t \leq \ell$. Therefore, $d_{\mathsf{mm}}(x_i, x_j; \mathcal{X}) \leq d(x_i, x_j) \leq d_\ell$.

Now, assume that the statement holds whenever $|C| < m$. Then $|C|$ must be the result of merging some two clusters $A, B$ such that $|A| < m-1$, $|B| < m-1$, and $d(A, B) \leq d_t \leq d_k$ for some iteration $t \leq \ell$. Let $x_i \in A$ and $x_j \in B$ be two arbitrary points. Since $d(A, B) \leq d_\ell$, there exist $u \in A, v \in B$ such that $d(u, v) \leq d_\ell$. By inductive hypothesis, $d_{\mathsf{mm}}(x_i, u; \mathcal{X}) \leq d_t \leq d_\ell$ and $d_{\mathsf{mm}}(x_j, v; \mathcal{X}) \leq d_t \leq d_\ell$. Consequently, there exist paths $p_1 \in \mathcal{P}_{x_i, u}, p_2 \in \mathcal{P}_{x_j, v}$ such that $p_1 \cup (u, v) \cup p_2$ is a path between $x_i$ and $x_j$ with maximum distance between successive nodes at most $d_\ell$. Therefore, $d_{\mathsf{mm}}(x_i, x_j; \mathcal{X}) \leq d_\ell$.

For the reverse direction, it is easy to see that after the merging step in iteration $k$, all nodes $u, v$ such that $d(u, v) \leq d_k$ must be in the same cluster. Consequently, if $d_{\mathsf{mm}}(x_i, x_j; \mathcal{X}) \leq d_k$, there exists a path $p \in \mathcal{P}_{i,j}$ where $p = (v_1 = x, v_2, ..., v_k, v_{k+1} = x_j)$ with $d(v_{i+1}, v_i) \leq d_k$ for all $i \in [k]$. Consequently, all vertices on $p$ must be in the same cluster, and hence $x_i, x_j \in C$ for some $C \in \mathcal{C}^k$. $\qquad\square$

The analogous statement clearly holds for $\mathcal{X}_m$ as well, as formalized by the following corollary.

**Corollary E.3.** *Under $\mathsf{SL}(\mathcal{X}_m, k)$, $x, y \in \mathcal{X}_m$ are in the same cluster after the $\ell$-th round of the outer while loop if and only if $d_{\mathsf{mm}}(x, y; \mathcal{X}_m) \leq d_{m_\ell}$.*

*Proof.* We repeat the identical argument used in Lemma 2.6 on $\mathcal{X}_m$. $\qquad\square$

Lemma 2.6 and Corollary E.3 characterize the criteria under which points in $\mathcal{X}$ or $\mathcal{X}_m$ will be merged together in single linkage clustering. We can use these characterizations to obtain the following corollary, which relates $d_{g(S)}$ and $d_{m_{g(S)}}$ (see Table 3) to $d_{\mathsf{mm}}(S; \mathcal{X})$ and $d_{\mathsf{mm}}(S; \mathcal{X}_m)$ respectively (recall Definition 2.5).

**Corollary E.4.** *For any clustering instance $\mathcal{X}$ and set $S \subseteq \mathcal{X}$, $d_{g(S)} = \max_{x,y \in S} d_{\mathsf{mm}}(S; \mathcal{X})$. Likewise, for any $S \subseteq \mathcal{X}_m$, $d_{m_{g_m(S)}} = d_{\mathsf{mm}}(S; \mathcal{X}_m)$.*

*Proof.* By definition, we know that on round $g(S)$, two sets $A$ and $B$ were merged where $S \subseteq A \cup B$ but $S \not\subseteq A$ and $S \not\subseteq B$. Moreover, $d_{g(S)} = \min_{x \in A, y \in B} d(x, y)$. We claim that for any $x \in S \cap A$ and $y \in S \cap B$, $d_{\mathsf{mm}}(x, y; \mathcal{X}) \geq d_{g(S)}$. For a contradiction, suppose there exists $x \in S \cap A$ and $y \in S \cap B$ such that $d_{\mathsf{mm}}(x, y; \mathcal{X}) < d_{g(S)}$. Then there exists a path $p$ between $x$ and $y$ such that for all elements $v_i, v_{i+1}$ on that path, $d(v_i, v_{i+1}) < d_{g(S)}$. However, this path must include elements $v_i, v_{i+1}$ such that $v_i \in A$ and $v_{i+1} \in B$, which contradicts the fact that $d_{g(S)} = \min_{x \in A, y \in B} d(x, y)$. Conversely, by Lemma 2.6, we know that for all $x, y \in S$, $d_{\mathsf{mm}}(x, y; \mathcal{X}) \leq d_{g(S)}$. Therefore, $d_{g(S)} = \max_{x,y \in S} d_{\mathsf{mm}}(x, y; \mathcal{X})$. The second statement follows by identical argument on $\mathcal{X}_m$. $\square$

Our next step is to use Corollary E.4 to understand when the clustering $\mathcal{C} = \mathsf{SL}(\mathcal{X}, k)$ and $\mathcal{C}' = \mathsf{SL}(\mathcal{X}_m, k)$ are "significantly" different. Concretely, our goal is to provide sufficient guarantees to ensure that $\mathcal{C}$ and $\mathcal{C}'$ are *consistent* with each other on $\mathcal{X}_m$, i.e., that $\exists \sigma \in \Sigma_k$ such that for each cluster $C_i' \in \mathcal{C}'$, $C_i' \subset C_{\sigma(i)}$. To aid in visualization, Figure 12 provides an example of a subsample $\mathcal{X}_m$ which induces a consistent clustering and an example of a subsample $\mathcal{X}_m$ which induces an inconsistent clustering.

Now, suppose, for example, that $\mathcal{C}'$ is *not consistent* with $\mathcal{C}$ and let $C_t \in \mathcal{C}$ be the last cluster from $\mathcal{C}$ to be merged into a single cluster when running single linkage on the subsample $\mathcal{X}_m$. Since $\mathcal{C}'$ and $\mathcal{C}$ are *not consistent*, there must be some cluster $Y \in \mathcal{C}'$ which contains points from *both* of two other clusters. This means that when running $\mathsf{SL}(\mathcal{X}, k)$, $d_{g(Y)} > d_{g(C)}$ for all $C \in \mathcal{C}$. However, recall that since $Y \in \mathcal{C}'$, when running $\mathsf{SL}(\mathcal{X}_m, k)$, there must exist some $C_t \in \mathcal{C}$ such that $d_{m_{g_m(Y)}} < d_{m_{g_m(C_t)}}$. Since $d_{g(Y)} > d_{g(C_t)}$ but $d_{m_{g_m(Y)}} < d_{m_{g_m(C_t)}}$, this means that when subsampling from $\mathcal{X}$ points, the subsample $\mathcal{X}_m$ must have distorted the min-max distances restricted to $C_t$ and $Y$. The following Lemma E.5 formalizes this observation.

**Lemma E.5.** *Suppose $T$ is merged into a single cluster under $\mathsf{SL}(\mathcal{X}_m, k)$ before $S$ is merged into a single cluster (i.e., $g_m(S) > g_m(T)$). Then there exists a pair of points $u, v \in S$ such that $d_{\mathsf{mm}}(u, v; \mathcal{X}_m) - d_{\mathsf{mm}}(u, v; \mathcal{X}) \geq d_{g(T)} - d_{g(S)}$.*

*Proof.* First, we argue that if $g(S) \geq g(T)$, then the statement is trivial. Indeed, if $g(S) \geq g(T)$, then for any $u, v \in \mathcal{X}$, $d_{\mathsf{mm}}(u, v; \mathcal{X}_m) - d_{\mathsf{mm}}(u, v; \mathcal{X}) \geq 0 \geq d_{g(T)} - d_{g(S)}$. The first inequality is because any path in $\mathcal{X}_m$ clearly exists in $\mathcal{X}$, and the second inequality is because $d_\ell$ is non-decreasing in $\ell$.

On the other hand, if $g(S) < g(T)$, then Corollary E.4 indicates that

$$\max_{x,y \in S} d_{\mathsf{mm}}(x, y; \mathcal{X}) = d_{g(S)} < d_{g(T)} = \max_{x,y \in T} d_{\mathsf{mm}}(x, y; \mathcal{X}).$$

Meanwhile, since $g_m(S) > g_m(T)$, there exists a pair of nodes $u, v \in S$ such that

$$d_{\mathsf{mm}}(u, v; \mathcal{X}_m) = \max_{x,y \in S} d_{\mathsf{mm}}(x, y; \mathcal{X}_m) > \max_{x,y \in T} d_{\mathsf{mm}}(x, y; \mathcal{X}_m) \geq \max_{x,y \in T} d_{\mathsf{mm}}(x, y; \mathcal{X}) = d_{g(T)}.$$

Finally, because $u, v \in S$, Corollary E.4 guarantees that $d_{\mathsf{mm}}(u, v; \mathcal{X}) \leq d_{g(S)}$, we can conclude that $d_{\mathsf{mm}}(u, v; \mathcal{X}_m) - d_{\mathsf{mm}}(u, v; \mathcal{X}) > d_{g(T)} - d_{g(S)}$. $\square$

Lemma E.5 illustrates that the order in which sets in $\mathcal{X}_m$ are merged in $\mathsf{SL}(\mathcal{X})$ and $\mathsf{SL}(\mathcal{X}_m)$ can differ *only* if some min-max distance is sufficiently distorted when points in $\mathcal{X}$ were deleted to construct $\mathcal{X}_m$. In the remainder of the analysis, we essentially seek to show that *large* distortions in min-max distance are unlikely because they require deleting many consecutive points along a path. To this end, we first prove an auxiliary lemma, which bounds the probability of deleting $\ell$ points when drawing a uniform subsample of size $m$ with replacement from $n$ points.

**Lemma E.6.** *Let $S$ be a set of $n$ points and $S'$ be a random subsample of $m$ points from $S$, drawn without replacement. Let $T = \{t_1, ..., t_\ell\} \subset S$. Then,*

$$\exp\left(-\frac{m\ell}{(n-\ell)}\right) \leq \mathbb{P}\{T \cap S' = \emptyset\} \leq \exp\left(-\frac{m\ell}{n}\right).$$

*Proof.* Each sample, independently, does not contain $t_i$ for $i \in [\ell]$ with probability $1 - \ell/n$. Thus,

$$\mathbb{P}\{T \cap S' = \emptyset\} = \left(1 - \frac{\ell}{n}\right)^m.$$

For the upper bound, we use the property that $(1 - x/m)^m \leq \exp(-x)$:

$$\left(1 - \frac{\ell}{n}\right)^m = \left(1 - \frac{m\ell/n}{m}\right)^m \leq \exp\left(-\frac{m\ell}{n}\right).$$

Meanwhile, for the lower bound, we use the property that $\exp(-x/(1-x)) \leq 1 - x$ for $x \in [0,1]$:

$$\left(1 - \frac{\ell}{n}\right)^m \geq \exp\left(-m\frac{\ell/n}{1 - \ell/n}\right) = \exp\left(\frac{-m\ell}{n-\ell}\right).$$

$\square$

We can utilize the bound in Lemma E.6 to bound the probability of distorting the min-max distance between a pair of points by more than an additive factor $\eta$.

**Lemma E.7.** *For $\eta > 0$ and $S \subset \mathcal{X}$,*

$$\mathbb{P}\{\exists u, v \in \mathcal{X}_m \cap S : d_{\mathsf{mm}}(u, v; \mathcal{X}_m) - d_{\mathsf{mm}}(u, v; \mathcal{X}) > \eta\} \leq n^3 \exp\left(-m\lceil \eta/d_{g(S)} \rceil/n\right).$$

*Proof.* Suppose $\exists u, v \in \mathcal{X}_m \cap S$ such that $d_{\mathsf{mm}}(u, v; \mathcal{X}_m) - d_{\mathsf{mm}}(u, v; \mathcal{X}) > \eta$. Then, $\exists p = (u = p_1, p_2, ..., p_t = v) \in \mathcal{P}_{u,v,\mathcal{X}} \setminus \mathcal{P}_{u,v,\mathcal{X}_m}$ with cost $\max_i d(p_i, p_{i-1}) = d_{\mathsf{mm}}(u, v; \mathcal{X})$. Since $t \leq n$ and $u, v \in S$, deleting any node $p_i \in p$ can increase the cost of this path by at most $d_{g(S)}$. Consequently, $\mathcal{X}_m$ must have deleted at least $s = \lceil \eta/d_{g(S)} \rceil$ *consecutive* points along $p$. There are at most $n$ distinct sets of $s$ consecutive points in $p$. Let $E_i$ be the event that $x_1, ..., x_s \notin \mathcal{X}_m$. Then, by Lemma E.6,

$$\mathbb{P}\{x_1, ..., x_s \notin \mathcal{X}_m\} \leq \exp\left(-\frac{ms}{n}\right).$$

The statement now follows by union bound over the $n$ distinct sets of $s$ consecutive points in $p$ and over the at most $n^2$ pairs of points $u, v \in S$. $\square$

Finally, we can apply Lemma E.5 and Lemma E.7 to bound the probability that $\mathcal{C}'$ and $\mathcal{C}$ are inconsistent.

**Lemma E.8.** *Let $\mathcal{C} = \{C_1, ..., C_k\} = \mathsf{SL}(\mathcal{X}, k)$ and $\mathcal{C}' = \{C'_1, ..., C'_k\} = \mathsf{SL}(\mathcal{X}_m, k)$. Let $\alpha_{ij} = d_{(g(C_i \cup C_j))}$ and $\alpha_i = d_{g(C_i)}$. Let $E$ be the event that there exists a $C'_i$ such that $C'_i \not\subset C_j$ for all $j \in [k]$ (i.e., that $\mathcal{C}'$ is inconsistent with $\mathcal{C}$; see Figure 12). Then,*

$$\mathbb{P}\{E\} \leq (kn)^3 \exp\left(-\frac{m}{n\zeta_{k,\mathsf{SL}}}\right)$$

*Proof.* Let $S_\ell = C_\ell \cap \mathcal{X}_m$ for $\ell \in [k]$. Without loss of generality, we can assume that $i = 1$. So, let $E_{a,b,j}$ be the event that $C'_1 \cap C_a \neq \emptyset \neq C'_1 \cap C_b$ and

$$g_m(C'_1) < g_m(S_j), \text{ and } g(S_j) < g(C'_1).$$

By Lemma E.5, this implies that there exists a $u, v \in S_j$ such that

$$d_{\mathsf{mm}}(u, v; \mathcal{X}_m) - d_{\mathsf{mm}}(u, v; \mathcal{X}) \geq d_{g(C'_1)} - d_{g(S_j)}.$$

Since $C_a, C_b \in \mathcal{C}$, we know that $g(C'_1) = g(C_1 \cup C_2)$. Meanwhile, $g(S_j) = \alpha_j$, so we have

$$d_{\mathsf{mm}}(u, v; \mathcal{X}_m) - d_{\mathsf{mm}}(u, v; \mathcal{X}) \geq \alpha_{ab} - \alpha_j.$$

By Lemma E.7, it follows that

$$\mathbb{P}\left\{E_{a,b,j}\right\} \le n^3 \max_{a,b,j\in[k]} \exp\left(-\frac{m}{n}\left\lceil\frac{(\alpha_{ab}-\alpha_j)}{\alpha_j}\right\rceil\right) = (kn)^3 \exp\left(-\frac{m}{n\zeta_{k,\mathsf{SL}}}\right).$$

Now, $E = \bigvee_{a,b,j} E_{a,b,j}$. By union bound over the at most $k^3$ configurations of $a,b,j$, the lemma follows. $\qquad\square$

Finally, we can condition on the event that $\mathcal{C}$ and $\mathcal{C}'$ are consistent to obtain our main result.

**Theorem 2.7.** *Let $\mathcal{G} = \{G_1,...,G_k\}$ be a ground-truth clustering of $\mathcal{X}$, $\mathcal{C} = \{C_1,...,C_k\} = \mathsf{SL}(\mathcal{X},k)$ be the clustering obtained from the full dataset, and $\mathcal{C}' = \{C_1',...,C_k'\} = \mathsf{SL}(\mathcal{X}_m,k)$ be the clustering obtained from a random subsample $\mathcal{X}_m$ of size $m$. For*

$$m = \tilde{\mathcal{O}}\left(\left(\frac{k}{\epsilon^2} + \frac{n}{\min_{i\in[k]}|C_i|} + \zeta_{k,\mathsf{SL}}(\mathcal{X})\right)\log\frac{k}{\delta}\right),$$

*we have that $\mathbb{P}\left\{\mathsf{cost}_\mathcal{G}\left(\mathcal{C}'\right) \approx_\epsilon \mathsf{cost}_\mathcal{G}\left(\mathcal{C}\right)\right\} \ge 1-\delta$. Computing $\mathcal{C}'$ requires $\mathcal{O}(m^2)$ calls to the distance oracle, while computing $\mathsf{cost}_\mathcal{G}\left(\mathcal{C}'\right)$ requires only $m$ queries to the ground-truth oracle.*

*Proof.* Let $E$ be as defined in Lemma E.8 and let $F$ be the event that $\mathcal{X}\cap C_i = \emptyset$ for some $i\in\mathcal{X}$. Let $H = E \vee F$. $\mathbb{P}\left\{H\right\} \le \mathbb{P}\left\{E\right\} + \mathbb{P}\left\{F\right\}$, where, by Lemma E.6,

$$\mathbb{P}\left\{F\right\} \le k\max_i \mathbb{P}\left\{\mathcal{X}\cap C_i = \emptyset\right\} \le k\exp\left(-\frac{m\cdot\min_{i\in[k]}|C_i|}{n}\right).$$

And by Lemma E.8, we have that

$$\mathbb{P}\left\{E\right\} \le (nk)^3 \exp\left(-\frac{m}{\zeta_{k,\mathsf{SL}}}\right).$$

We also have that

$$\mathbb{P}\left\{\mathsf{cost}_\mathcal{G}\left(\mathcal{C}'\right) \approx_\epsilon \mathsf{cost}_\mathcal{G}\left(\mathcal{C}\right)\right\} \ge \mathbb{P}\left\{\mathsf{cost}_\mathcal{G}\left(\mathcal{C}'\right) \approx_\epsilon \mathsf{cost}_\mathcal{G}\left(\mathcal{C}\right)|\bar{H}\right\}\mathbb{P}\left\{\bar{H}\right\}.$$

Now, let $\tau : \mathcal{X} \mapsto [k]$ be the mapping such that $\tau(x) = i$ if and only if $x\in G_i$. Consider any permutation $\sigma : [k] \to [k]$. If $\bar{H}$ occurs, then we know that every $C_i' \subset C_{j_i}$ for some $j_i \in [k]$, and we know that each $C_j \cap \mathcal{X}_m \ne \emptyset$ for each $j\in[k]$. Consequently, there exists a permutation $\rho : [k] \mapsto [k]$ such that $C_i' \subset C_{\rho(i)}$. Without loss of generality, we can assume that $\rho$ is the identity, i.e., that $x\in C_i \iff x\in C_i'$ for any $x\in\mathcal{X}_m$ (or else reorder the sets in $\mathcal{C}$ such that this holds). So, conditioning on $\bar{H}$, we have

$$\frac{1}{m}\sum_{x\in\mathcal{X}_m}\sum_{i\in[k]}\mathbb{1}\left(x\in C_{\sigma(i)}' \wedge x\notin G_i\right) = \frac{1}{m}\sum_{x\in\mathcal{X}}\sum_{i\in[k]}\mathbb{1}\left(x\in C_{\sigma(i)}' \wedge x\notin G_i \wedge x\in\mathcal{X}_m\right)$$

$$= \frac{1}{m}\sum_{x\in\mathcal{X}}\sum_{i\in[k]}\mathbb{1}\left(x\in C_{\sigma(i)} \wedge x\notin G_i \wedge x\in\mathcal{X}_m\right),$$

$$\mathbb{E}\left[\frac{1}{m}\sum_{x\in\mathcal{X}_m}\sum_{i\in[k]}\mathbb{1}\left(x\in C_{\sigma(i)}' \wedge x\notin G_i\right)\right] = \mathbb{E}\left[\frac{1}{m}\sum_{x\in\mathcal{X}}\sum_{i\in[k]}\mathbb{1}\left(x\in C_{\sigma(i)} \wedge x\notin G_i \wedge x\in\mathcal{X}_m\right)\right]$$

$$= \frac{1}{m}\frac{m}{n}\cdot\mathbb{E}\left[\sum_{x\in\mathcal{X}}\sum_{i\in[k]}\mathbb{1}\left(x\in C_{\sigma(i)} \wedge x\notin G_i\right)\right].$$

So, by Hoeffding's inequality and union bound over the permutations $\tau$,

$$\mathbb{P}\left\{|\mathsf{cost}_\mathcal{G}\left(\mathcal{C}'\right) - \mathsf{cost}_\mathcal{G}\left(\mathcal{C}\right)| \le \epsilon|\bar{H}\right\} \ge 1 - (k!)\exp\left(-2\epsilon^2 m\right) \ge 1 - k^k\exp\left(-2\epsilon^2 m\right).$$

Consequently, by taking $m \ge \mathcal{O}\left(k\epsilon^{-2}\log\left(k\delta^{-1}\right)\right)$, we can ensure that

$$\mathbb{P}\left\{\mathsf{cost}_\mathcal{G}\left(\mathcal{C}'\right) \approx_\epsilon \mathsf{cost}_\mathcal{G}\left(\mathcal{C}\right)|\bar{H}\right\} \ge 1 - \delta/2.$$

By taking $m \ge \mathcal{O}\left(\max\left(\zeta_{k,\mathsf{SL}}(\mathcal{X})\log\left(nk\delta^{-1}\right), \frac{n}{\min_{i\in[k]}|C_i|^2}\log\left(\delta^{-1}\right)\right)\right)$ we can also ensure $\mathbb{P}\left\{\bar{H}\right\} \ge 1 - \delta/2$. Hence, $\mathbb{P}\left\{\mathsf{cost}_\mathcal{G}\left(\mathcal{C}'\right) \approx_\epsilon \mathsf{cost}_\mathcal{G}\left(\mathcal{C}\right)\right\} \ge 1 - \delta$. $\qquad\square$

### E.2.2 Lower bounds for size generalization of single linkage clustering

We now turn our attention to proving the lower bound results in Section 2.2. In the following, we use the notation $B_\epsilon(z) := \{z \in \mathbb{R} : \|x - z\|_2 \leq \epsilon\}$ to denote an $\epsilon$-ball centered at $z$ and consider clustering with respect to the standard Euclidean metric $d(x, y) := \|x - y\|_2$.

**Lemma E.9.** *For any odd $n \geq 110$ and $\gamma > 0$, there exists a 2-clustering instance on $n$ nodes with a ground truth clustering $\mathcal{G}$ such that $\min_{i \in [k]} |C_i| = 1$, $\zeta_{k,\mathsf{SL}}(\mathcal{X}) \leq \gamma$ and with probability .27 for $m = n - 1$, $|\mathsf{cost}_\mathcal{G}(\mathsf{SL}(\mathcal{X}_m, 2)) - \mathsf{cost}_\mathcal{G}(\mathsf{SL}(\mathcal{X}, 2))| \geq 1/4$.*

*Proof.* Consider $0 < \alpha < 1/2$ and the 2-clustering instance $\mathcal{X} = \{x_1, ..., x_{n-1}, x_n\} \subset \mathbb{R}^2$ where $x_1, ..., x_{(n-1)/2} \in B_\alpha(0)$, $x_{(n-1)/2+1}, ..., x_{n-1} \in B_\alpha(3)$, and $x_n = 6$. Suppose $\mathcal{G} = \{G_1, G_2\}$ where $G_1 = \{x_1, ..., x_{n-1}\}$ and $G_2 = \{x_n\}$.

It is easy to see that $\mathsf{cost}_\mathcal{G}(\mathsf{SL}(\mathcal{X})) = 0$, since $x_n$ is sufficiently far from 0 and 3 to ensure that $x_n$ will be the last point to be merged with any other point. Consider $m = n - 1$. Whenever $x_n \notin \mathcal{X}_m$, $|\mathcal{X}_m \cap B_\alpha(0)| \geq n/4$, and $|\mathcal{X}_m \cap B_\alpha(3)| \geq n/4$ we have that $\mathsf{cost}_\mathcal{G}(\mathsf{SL}(\mathcal{X}_m)) \geq 1/4$, as the algorithm will separate the points in $B_\alpha(0)$ from those in $B_\alpha(3)$. We can lower bound the probability of this event as follows.

By Lemma E.6,
$$\mathbb{P}\{x_n \in \mathcal{X}_m\} \leq 1 - \exp(-1) \leq 0.63.$$

By a Chernoff bound,
$$\mathbb{P}\left\{|B_\alpha(3) \cap \mathcal{X}_m| \leq \frac{n}{4}\right\} = \mathbb{P}\left\{|B_\alpha(0) \cap \mathcal{X}_m| \leq \frac{n}{4}\right\}$$
$$= \mathbb{P}\left\{|B_\alpha(0) \cap \mathcal{X}_m| \leq \frac{(n-1)^2}{2n} \cdot \frac{2n^2}{4(n-1)^2}\right\}$$
$$\leq \mathbb{P}\left\{|B_\alpha(0) \cap \mathcal{X}_m| \leq \frac{(n-1)^2}{2n} \cdot 2/3\right\}$$
$$\leq \exp\left(-\frac{(n-1)^2}{36n}\right) \leq .05.$$

Thus, $\mathbb{P}\{\mathsf{cost}_\mathcal{G}(\mathsf{SL}(\mathcal{X}_m)) \geq 1/4\} \geq 1 - 0.73 = .27$. Moreover, note that because the failure probabilities analyzed above are *independent* of $\alpha$, we can ensure that $\zeta_{k,\mathsf{SL}} \leq \gamma$ without affecting any of the failure probabilities, which are independent. $\qquad\square$

**Lemma E.10.** *For any $n \geq 51$ with $n = 1 \bmod 3$, there exists a 2-clustering instance $\mathcal{X}$ on $n$ points such that $\zeta_{k,\mathsf{SL}} = n$; $\mathcal{C} = \mathsf{SL}(\mathcal{X}, 2)$ satisfies $\min_{C \in \mathcal{C}} |C| \geq \frac{n}{6}$; and for $m = n - 1$, with probability .23, $|\mathsf{cost}_\mathcal{G}(\mathsf{SL}(\mathcal{X}_m, 2)) - \mathsf{cost}_\mathcal{G}(\mathsf{SL}(\mathcal{X}, 2))| \geq 1/12$.*

*Proof.* Take $\alpha > 0$ and $2\alpha > \beta > \alpha$. Suppose $\mathcal{X}$ is composed of four sets $\mathcal{X} = L \cup \{b\} \cup M \cup R$, where the left points $x \in L$ satisfy $x = 0$. The middle points $x \in M$ satisfy $x = 2\alpha$, and the right points $x \in R$ satisfy $x = 2 \cdot \alpha + \beta$. $|L| = |M| = |R| = \frac{n-1}{3}$. The "bridge" point $b = \alpha$. Here,
$$\zeta_{k,\mathsf{SL}}(\mathcal{X}) = n\lceil(\beta - \alpha)/\alpha\rceil^{-1} = n.$$

Suppose the ground truth clustering $\mathcal{G}$ is defined as $G_1 = L \cup \{b\} \cup M$ and $G_2 = R$. Since $\beta > \alpha$, single linkage achieves a cost of 0 on $\mathcal{X}$. Meanwhile, suppose that $b \notin \mathcal{X}_m$. Then, because the minmax distance between any point in $L$ and $M$ is now $2\alpha > \beta$, single linkage run on $\mathcal{X}_m$ will create one cluster for $L$ and one cluster for $M \cup R$. Provided that $\mathcal{X}_m$ contains at least $\frac{n-1}{6}$ points in $M$, this implies that single linkage will have a cost greater than or equal to $1/12$ on $\mathcal{X}_m$. It now remains to bound the probability of these simultaneous events. Let $E_1$ be the event that $b \notin \mathcal{X}_m$, and let $E_2$ be the event that $|M \cap \mathcal{X}_m| \geq \frac{n-1}{6}$. We have
$$\mathbb{P}\{E_1 \wedge E_2\} = \mathbb{P}\{E_1\}\mathbb{P}\{E_2|E_1\} \geq \mathbb{P}\{E_1\}\mathbb{P}\{E_2\}.$$

This is because $E_1$ is the event of *not including* certain elements in $\mathcal{X}_m$ and $E_2$ is the probability of *including* certain elements in $\mathcal{X}_m$, so, $\mathbb{P}\{E_2|E_1\} \geq \mathbb{P}\{E_2\}$. Now, consider $m = n - 1$. By Lemma E.6,
$$\mathbb{P}\{E_1\} \geq \exp\left(-m\frac{1}{(n-1)}\right) = \exp(-1) \geq .36.$$

**Input:** $X$ is a feasible matrix for (2)
Initialize $\boldsymbol{u} \in \mathbb{R}^n$, sampled from the $n$-dimensional standard Gaussian distribution
Compute $VV^T = X$           // Compute the Cholesky factorization of $X$
**for** $i = \{1, 2, ..., n\}$ **do**
    |    $z_i = \text{sign}(\boldsymbol{v}_i^T \boldsymbol{u})$, where $\boldsymbol{v}_i$ is the $i$-th column of matrix $V$
**return** $\boldsymbol{z_u}$

**Algorithm 4:** Rounding Procedure GWRound($X$)

Meanwhile, to analyze $\mathbb{P}\{E_2\}$ we can use a Chernoff bound to see that whenever $n > 3$, we have $n > m > 2/3n$, and consequently,

$$
\begin{aligned}
\mathbb{P}\{E_2\} &= 1 - \mathbb{P}\left\{|\mathcal{X}_m \cap M| < \frac{n-1}{6}\right\} \\
&= 1 - \mathbb{P}\left\{|\mathcal{X}_m \cap M| < \frac{n}{2m} \cdot \frac{m(n-1)}{3n}\right\} \\
&= 1 - \mathbb{P}\left\{|\mathcal{X}_m \cap M| < \left(1 - \frac{2m-n}{2m}\right) \cdot \frac{m(n-1)}{3n}\right\} \\
&\geq 1 - \exp\left(-\frac{(2m-n)^2}{2(2m)^2}\frac{m}{6}\right) \\
&= 1 - \exp\left(-\frac{(n-2)^2}{8(n-1)^2}\frac{(n-1)}{6}\right) \\
&= 1 - \exp\left(-\frac{(n-2)^2}{48(n-1)}\right).
\end{aligned}
$$

where, in the fourth line, we substituted $\frac{m}{n} \cdot \frac{n-1}{3} \geq \frac{m}{n}\frac{n}{6} = \frac{m}{6}$. Then, whenever $n \geq 51$, we have that the argument in the exponential is at least 1. Hence,

$$
\mathbb{P}\{E_1 \wedge E_2\} \geq \exp(-1)\left(1 - \exp(-1)\right) > .23.
$$

So, the lemma follows by a union bound. $\qquad\square$

### E.3    Omitted Proofs from Section 3.1

This section provides omitted proofs relating to our results and discussions on the GW algorithm.

The following lemma shows that GW SDP attains strong duality, included here for completeness.

**Lemma E.11.** *The GW SDP relaxation of the max-cut problem (Equation* (2)*) and its dual problem (Equation* (3)*) attain strong duality.*

*Proof.* Notice that any PSD matrix $X \in \mathbb{R}^{n \times n}$ with 1 on the diagonal is feasible to Equation (2). Thus, we can find at least two distinct feasible solutions to Equation (2). Any convex combination of the two is a solution in the non-empty interior of the primal.

To find a feasible solution of the dual, notice that we can always choose $y_i = 2\sum_j w_{i,j} + \lambda$ where $\lambda > 0$ to make the matrix $S = \sum_{i=1}^n \left(e_i e_i^T\right) y_i - L$ a diagonally dominant matrix as shown below:

$$
S_{i,i} = 2\sum_j w_{i,j} + \lambda - \sum_j w_{i,j} = \sum_j w_{i,j} + \lambda \leq \sum_j |w_{i,j}|.
$$

Any diagonally dominant matrix is PSD. We choose distinct $\lambda_1$ and $\lambda_2$ to construct feasible $y_1$ and $y_2$. Any convex combination of $y_1$ and $y_2$ is a solution in the non-empty interior set of the dual problem. Because both primal and dual problems are feasible and have interiors, by Slater's condition, strong duality holds. $\qquad\square$

**Lemma E.12.** *Suppose we have a graph $G = (V, E, \boldsymbol{w})$. Let $S_t \subset V$. Let $\boldsymbol{y}^*$ be the optimal solution to the dual problem* (3) *induced by $G$. Let $\bar{y}_i = y_i^* - \sum_{k \in V \setminus S_t} w_{ik}$. Then $\bar{\boldsymbol{y}}$ is a feasible solution to the dual problem induced by $G[S_t]$.*

*Proof.* Let the set of node $V$ in $G$ be indexed with $S_t$ as its first $t$ elements. The optimal objective value of SDP does not depend on the node indexing, nor does the dual objective value (by strong duality).

Let $\boldsymbol{y}^*$ be the optimal dual solution. It satisfies the following dual constraint:

$$\sum_{i=1}^{n} \left( \boldsymbol{e_i} \boldsymbol{e_i}^T \right) y_i^* - \frac{1}{4} L_G \succeq 0,$$

where $\boldsymbol{e_i} \in \mathbb{R}^n$ denotes the all-zero vector with 1 on index $i$. Therefore, $\boldsymbol{e_i} \boldsymbol{e_i}^T$ is the all-zero matrix with 1 on the $(i, i)$ index. In other words, $\sum_{i=1}^{n} \left( \boldsymbol{e_i} \boldsymbol{e_i}^T \right) y_i^*$ is the diagonalization of the vector $\boldsymbol{y}^*$.

Let $L_G^t$ denote the $t$-th principle minor of $L_G$, i.e. the upper left $t$-by-$t$ sub-matrix of $L_G$. To distinguish between $\boldsymbol{e_i} \in \mathbb{R}^t$ and $\boldsymbol{e_i} \in \mathbb{R}^n$, for the rest of the proof, we use $\boldsymbol{e_{n,i}}$ to denote the the former $n$-dimensional vector and $\boldsymbol{e_{t,i}}$ to denote the latter $t$-dimensional vector. By Sylvester's criterion, because $\sum_{i=1}^{n} \left( \boldsymbol{e_{n,i}} \boldsymbol{e_{n,i}}^T \right) y_i^* - \frac{1}{4} L_G \succeq 0$, we have that $\sum_{i=1}^{t} \left( \boldsymbol{e_{t,i}} \boldsymbol{e_{t,i}}^T \right) y_i^* - \frac{1}{4} L_G^t \succeq 0$. Let $L_{G[S_t]}$ denote the Laplacian of $G[S_t]$. We thus have:

$$\sum_{i=1}^{t} \left( \boldsymbol{e_{t,i}} \boldsymbol{e_{t,i}}^T \right) y_i^* - \frac{1}{4} L_{G[S_t]} - \frac{1}{4} \left( L_G^t - L_{G[S_t]} \right) \succeq 0.$$

By the definition of Laplacian matrices, $L_G^t$ and $L_{G[S_t]}$ are both $t$-by-$t$ matrices with the same off-diagonal values, as shown below:

$$L_G^t = \begin{bmatrix} \ddots & & \vdots & & \\ \cdots & \sum_{k \in V} w_{ik} & w_{ij} & \cdots & \\ & w_{ij} & & \ddots & \\ & \vdots & & & \ddots \end{bmatrix}, \text{ and } \quad L_{G[S_t]} = \begin{bmatrix} \ddots & & \vdots & & \\ \cdots & \sum_{k \in S_t} w_{ik} & w_{ij} & \cdots & \\ & w_{ij} & & \ddots & \\ & \vdots & & & \ddots \end{bmatrix}.$$

Moreover, $\left( L_G^t - L_{G[S_t]} \right)$ is a diagonal matrix with $(i,i)$-th element $\sum_{k \in V} w_{ik} - \sum_{k \in S_t} w_{ik} = \sum_{k \in V \setminus S_t} w_{ik}$. Thus, we have

$$\sum_{i=1}^{t} \left( \boldsymbol{e_{t,i}} \boldsymbol{e_{t,i}}^T \right) y_i^* - \frac{1}{4} L_{G[S_t]} - \frac{1}{4} (L_G^t - L_{G[S_t]}) \succeq 0$$

$$\Leftrightarrow \sum_{i=1}^{t} \left( \boldsymbol{e_{t,i}} \boldsymbol{e_{t,i}}^T \right) y_i^* - \frac{1}{4} L_{G[S_t]} - \sum_{i=1}^{t} \left( \boldsymbol{e_{t,i}} \boldsymbol{e_{t,i}}^T \right) \frac{1}{4} \sum_{k \in V \setminus S_t} w_{ik} \succeq 0$$

$$\Leftrightarrow \sum_{i=1}^{t} \left( \boldsymbol{e_{t,i}} \boldsymbol{e_{t,i}}^T \right) \left( y_i^* - \frac{1}{4} \sum_{k \in V \setminus S_t} w_{ik} \right) - \frac{1}{4} L_{G[S_t]} \succeq 0.$$

Let $\bar{y}_i = y_i^* - \frac{1}{4} \sum_{k \in V \setminus S_t} w_{ik}$. We have just shown that $\bar{\boldsymbol{y}}$ satisfies the constraint of the GW SDP dual problem induced by $G[S_t]$. It is therefore a feasible solution to the GW SDP dual problem induced by $G[S_t]$. $\square$

**Lemma E.13.** *Given $G = (V, E, \boldsymbol{w})$, let $S_t$ be a set of vertices with each node sampled from $V$ independently with probability $\frac{t}{n}$. The expected value of $\mathsf{SDP}(G[S_t])$ can be upper-bounded by:*

$$\mathbb{E}_{S_t}[\mathsf{SDP}(G[S_t])] \leq \frac{t}{n} \mathsf{SDP}(G) - \frac{t(n-t)}{n^2} \cdot \frac{W}{2},$$

*with $W = \sum_{e \in E} w_e$, i.e. the sum of all edge weights in $G$.*

*Proof.* From Lemma E.12, we know that $\bar{\boldsymbol{y}}$ is a feasible SDP dual solution induced by $G[S_t]$. Let $\hat{\boldsymbol{y}}$ be the optimal solution to Equation (3). Then, we have that for any $S_t$:

$$\mathsf{SDP}(G[S_t]) = \sum_{i \in S_t} \hat{y}_i \leq \sum_{i \in S_t} \bar{y}_i.$$

Taking the expectation over the random sample $S_t$ on both sides, we have:

$$\mathbb{E}_{S_t}[\mathsf{SDP}(G[S_t])] = \mathbb{E}_{S_t}\left[\sum_{i \in S_t} \hat{y}_i\right] \leq \mathbb{E}_{S_t}\left[\sum_{i \in S_t}\left(y_i^* - \frac{1}{4}\sum_{k \in V \setminus S_t} w_{ik}\right)\right]$$

$$= \mathbb{E}_{S_t}\left[\sum_{i \in V} \mathbb{1}\{i \in S_t\}\left(y_i^* - \frac{1}{4}\sum_{k \in V \setminus S_t} w_{ik}\right)\right]$$

$$= \mathbb{E}_{S_t}\left[\sum_{i \in V} \mathbb{1}\{i \in S_t\}y_i^*\right] - \mathbb{E}_{S_t}\left[\sum_{i \in V}\left(\mathbb{1}\{i \in S_t\}\frac{1}{4}\sum_{k \in V \setminus S_t} w_{ik}\right)\right].$$

Because each node in $S_t$ is sampled independently with probability $t/n$, the probability that $i \in S_t, k \notin S_t$ is the product of probability that $i \in S_t$ and $k \notin S_t$, i.e. $\frac{t(n-t)}{n^2}$. We use this fact to finish upper bounding $\mathbb{E}_{S_t}[\mathsf{SDP}(G[S_t])]$:

$$\mathbb{E}_{S_t}[\mathsf{SDP}(G[S_t])] \leq \sum_{i \in V} \mathbb{P}[i \in S_t]y_i^* - \mathbb{E}_{S_t}\left[\sum_{i \in V}\left(\mathbb{1}\{i \in S_t\}\frac{1}{4}\sum_{k \in V} \mathbb{1}\{k \notin S_t\}w_{ik})\right)\right]$$

$$= \frac{t}{n}\sum_{i \in V} y_i^* - \frac{1}{4}\sum_{i \in V}\sum_{k \in V} \mathbb{P}[i \in S_t, k \notin S_t]w_{ik}$$

$$= \frac{t}{n}\mathsf{SDP}(G) - \frac{t(n-t)}{n^2}\cdot\frac{W}{2},$$

with $W = \sum_{e \in E} w_e$, i.e. the sum of all edge weights in $G$. $\qquad\square$

**Lemma E.14.** *Given $G = (V, E, \boldsymbol{w})$, let $S_t$ be a set of vertices with each node sampled from $V$ independently with probability $\frac{t}{n}$. The expected value of $\mathsf{SDP}(G[S_t])$ can be lower-bounded by:*

$$\mathbb{E}_{S_t}[\mathsf{SDP}(G[S_t])] \geq \frac{t^2}{n^2}\mathsf{SDP}(G).$$

*Proof.* Let the set of node $V$ in $G$ be indexed with $S_t$ as its first $t$ elements. The optimal objective value of SDP does not depend on the node indexing. Let $X^* \in \mathbb{R}^{n \times n}$ denote the optimal solution for SDP induced by $G$ and $\hat{X} \in \mathbb{R}^{|S_t| \times |S_t|}$ denote the optimal solution for SDP induced by $G[S_t]$. Let $X_{S_t}^*$ denote the $|S_t|$-th principle minor of $X^*$. By the Slater's condition, $X_{S_t}^*$ is PSD. It is therefore a feasible solution to the GW SDP induced by $G[S_t]$:

$$\mathbb{E}_{S_t}[\mathsf{SDP}(G[S_t])] = \mathbb{E}_{S_t}[L_{G[S_t]} \cdot X_{S_t}] \geq \mathbb{E}_{S_t}[L_{G[S_t]} \cdot X_{S_t}^*].$$

We re-write the objective value in terms of the Cholesky decomposition of $X^*$, which we denote by $\boldsymbol{v}_i^*$ for all $i \in V$. Note that $X_{ij}^* = \boldsymbol{v}_i^{*T}\boldsymbol{v}_j^*$:

$$\mathbb{E}_{S_t}[\mathsf{SDP}(G[S_t])] \geq \mathbb{E}_{S_t}[L_{G[S_t]} \cdot X_{S_t}^*]$$

$$= \mathbb{E}_{S_t}\left[\frac{1}{2}\sum_{i,j \in S_t} w_{ij}(1 - \boldsymbol{v}_i^{*T}\boldsymbol{v}_j^*)\right]$$

$$= \mathbb{E}_{S_t}\left[\frac{1}{2}\sum_{(i,j) \in E} \mathbb{1}\{i \in S_t, j \in S_t\}w_{ij}(1 - \boldsymbol{v}_i^{*T}\boldsymbol{v}_j^*)\right].$$

By independence of the sampling of nodes $i$ and $j$, we have:

$$\mathbb{E}_{S_t}[\mathsf{SDP}(G[S_t])] \geq \frac{1}{2} \sum_{(i,j) \in E} \mathbb{P}[i \in S_t, j \in S_t] w_{ij}(1 - \boldsymbol{v}_i^{*T} \boldsymbol{v}_j^*)]$$

$$= \mathbb{P}[i \in S_t, j \in S_t]\mathsf{SDP}(G) = \frac{t^2}{n^2}\mathsf{SDP}(G).$$

$\square$

**Theorem 3.1.** *Given $G = (V, E, \boldsymbol{w})$, let $S_t$ be a set of vertices with each node sampled from $V$ independently with probability $\frac{t}{n}$. Let $W = \sum_{(i,j) \in E} w_{ij}$. Then*

$$\left| \frac{1}{t^2} \mathbb{E}_{S_t}[\mathsf{SDP}(G[S_t])] - \frac{1}{n^2}\mathsf{SDP}(G) \right| \leq \frac{n-t}{n^2 t}\left( \mathsf{SDP}(G) - \frac{W}{2} \right).$$

*Proof.* Combining Lemma E.13 and Lemma E.14 and basic algebraic manipulation gives us this result. $\square$

Theorem 3.1 gives us a size generalization bound in expectation over the draw of $S_t$. We are also interested in whether it is possible to attain a bound with probability over one draw of $S_t$. To do so, we first introduce the McDiarmid's Inequality.

**Theorem E.15** (McDiarmid's Inequality). *Let $f : \chi_1 \times \chi_2 \times ... \times \chi_n \to \mathbb{R}$ satisfy:*

$$\sup_{x_i' \in \chi_i} |f(x_1, ..., x_i, ...x_n) - f(x_1, ..., x_i', ...x_n)| \leq c_i.$$

*Then consider independent r.v. $X_1, X_2, ..., X_n$ where $X_i \in \chi_i$ for all $i$. For $\epsilon > 0$,*

$$\mathbb{P}[|f(X_1, ..., X_n) - \mathbb{E}[f(X_1, ..., X_n)]| \geq \epsilon] \leq 2 \exp\left( -\frac{2\epsilon^2}{\sum_{i=1}^{n} c_i^2} \right).$$

We wish to apply McDiarmid's inequality to provide a high probability bound for how different $\mathsf{SDP}(G[S_t])$ can be from $\mathbb{E}[\mathsf{SDP}(G[S_t])]$. In order to do so, we consider $\mathsf{SDP}(\cdot)$ as a function of a list of indicator variables, say $y_1, ..., y_n \in \{0, 1\}^n$, indicating whether node $i$ is in subsample $S_t$. We need a result that bound the maximum change in objective value if we add a node $i \notin S_t$ to $S_t$ or delete a node $j \in S_t$ from $S_t$. The following lemma states that this maximum change in objective can at most be the degree of the node added or deleted.

**Lemma E.16.** *Suppose we have a graph $G = (V, E)$ with $n = |V|$. Let $S \subseteq V$ be a subset of nodes and $k \in S$. Let $S' = S \setminus \{k\}$. Then,*

$$|\mathsf{SDP}(G[S]) - \mathsf{SDP}(G[S'])| \leq \deg(k).$$

*By a symmetric argument, let $k \in V$ and $k \notin S$. Let $S' = S + \{k\}$. Then we also have*

$$|\mathsf{SDP}(G[S]) - \mathsf{SDP}(G[S'])| \leq \deg(k).$$

*Proof.* Note that by a symmetric argument, the change in the SDP objective value after adding a node is the same as the change in the SDP objective value after deleting a node (those are the same argument with the definition of $S$ and $S'$ flipped). Thus, it is sufficient to prove the first part of the statement assuming $S$ has node $k$ and $S'$ does not.

We first show that $\mathsf{SDP}(G[S]) \geq \mathsf{SDP}(G[S'])$. GW SDP is permutation invariant, so suppose $k$ is indexed last in the set of nodes $S$. Let $t = |S|$. Let $X \in \mathbb{R}^{t \times t}$ be the optimal SDP solution induced by $G[S]$ and $X' \in \mathbb{R}^{(t-1) \times (t-1)}$ the optimal solution induced by $G[S']$. Padding $X'$ with 1 on $X_{tt}$ and 0 off-diagonal on the $t$-th dimension results in a feasible solution for GW SDP induced by $G[S]$, because a block-diagonal matrix in which each block is PSD is also PSD (by Cramer's

rule). Therefore, we know that adding a node will only increase the objective value of SDP, i.e. $\text{SDP}(G[S]) \geq \text{SDP}(G[S'])$. Thus,

$$|\text{SDP}(G[S]) - \text{SDP}(G[S'])| = \text{SDP}(G[S]) - \text{SDP}(G[S']).$$

We notice that the $(t-1)$-th principal minor of $X$ is a feasible solution to the GW SDP problem induced by $G[S']$ (because it is PSD and has 1 on the diagonal). Because $X'$ is the optimal solution to the GW SDP objective induced by $G[S']$,

$$\text{SDP}(G[S']) = \sum_{(i,j) \in E_{S'}} \frac{1}{2} w_{ij}(1 - x'_{ij}) \geq \sum_{(i,j) \in E_{S'}} \frac{1}{2} w_{ij}(1 - x_{ij}).$$

Applying those facts, we can show that

$$|\text{SDP}(G[S]) - \text{SDP}(G[S'])| = \text{SDP}(G[S]) - \text{SDP}(G[S'])$$

$$= \sum_{(i,j) \in E_S} \frac{1}{2} w_{ij}(1 - x_{ij}) - \sum_{(i,j) \in E_{S'}} \frac{1}{2} w_{ij}(1 - x'_{ij})$$

$$\leq \sum_{(i,j) \in E_S} \frac{1}{2} w_{ij}(1 - x_{ij}) - \sum_{(i,j) \in E_{S'}} \frac{1}{2} w_{ij}(1 - x_{ij}).$$

Let $E_S$ be the set of edges in $G[S]$ and $E_{S'}$ be the set of edges in $G[S']$. Because the set $S'$ contains all nodes in $S$ except for the node $k$, we know that the edge set $E_{S'}$ contains all edges in $E_S$ except $\{(i,k) \in E_S : i \in V\}$, thus we can split the sum over $E_S$ to two parts, $E_{S'}$ and the set $\{(i,k) \in E_S : i \in V\}$:

$$\text{SDP}(G[S]) - \text{SDP}(G[S']) = \left( \sum_{(i,j) \in E_{S'}} \frac{1}{2} w_{ij}(1 - x_{ij}) + \sum_{(i,k) \in E_S} \frac{1}{2} w_{ik}(1 - x_{ik}) \right)$$

$$- \sum_{(i,j) \in E_{S'}} \frac{1}{2} w_{ij}(1 - x_{ij})$$

$$= \sum_{(i,k) \in E_S} \frac{1}{2} w_{ik}(1 - x_{ik})$$

$$\leq \sum_{(i,k) \in E_S} w_{ik} = \deg(k).$$

$\square$

With Lemma E.16 and Lemma 3.1, we can put together a size generalization result for SDP with probability over the draw of $S_t$.

**Theorem E.17.** *Given* $G = (V, E)$, *let* $S_t$ *be a set of vertices with each node sampled from* $V$ *independently with probability* $\frac{t}{n}$. *With probability* $1 - \delta$ *over the draw of* $S_t$,

$$\left| \frac{1}{n^2} \text{SDP}(G) - \frac{1}{t^2} \text{SDP}(G[S_t]) \right| \leq \frac{n-t}{n^2 t} \left( \text{SDP}(G) - \frac{|E|}{2} \right) + \sqrt{\frac{n^3}{t^4} \log \left( \frac{2}{\delta} \right)}.$$

*Proof.* By McDiarmid's Inequality, we have that

$$\mathbb{P}[|\text{SDP}(G[S_t]) - \mathbb{E}[\text{SDP}(G[S_t])]| \geq \epsilon] \leq 2 \exp \left( -\frac{2\epsilon^2}{\sum_{i=1}^{n} \deg(i)^2} \right).$$

We can upper bound the sum of weighted degrees over all nodes by

$$\sum_{i=1}^{n} \deg(i)^2 \le \sum_{i=1}^{n} (n-1)\deg(i) = (n-1)|E|.$$

Thus, we can also rewrite the upper bound as

$$\mathbb{P}\left[|\mathsf{SDP}(G[S_t]) - \mathbb{E}[\mathsf{SDP}(G[S_t])]| \ge \epsilon\right] \le 2\exp\left(-\frac{\epsilon^2}{(n-1)|E|}\right).$$

We now combine this and the expectation bound for SDP size generalization: with probability $1 - 2\exp\left(-\frac{\epsilon^2}{|E|(n-1)}\right)$,

$$\left|\frac{1}{n^2}\mathsf{SDP}(G) - \frac{1}{t^2}\mathsf{SDP}(G[S_t])\right| \le \left|\frac{1}{n^2}\mathsf{SDP}(G) - \frac{1}{t^2}\mathbb{E}[\mathsf{SDP}(G[S_t])]\right|$$

$$+ \left|\frac{1}{t^2}\mathbb{E}[\mathsf{SDP}(G[S_t])] - \frac{1}{t^2}\mathsf{SDP}(G[S_t])\right|$$

$$\le \frac{n-t}{n^2 t}(\mathsf{SDP}(G) - W) + \frac{\epsilon}{t^2}.$$

We set

$$\delta = 2\exp\left(-\frac{\epsilon^2}{(n-1)|E|}\right)$$

$$\Leftrightarrow \log\left(\frac{\delta}{2}\right) = -\frac{\epsilon^2}{(n-1)|E|}$$

$$\Leftrightarrow \epsilon^2 = (n-1)|E|\log\left(\frac{2}{\delta}\right) \le (n-1)^2 n \log\frac{2}{\delta}.$$

Thus, with probability $1 - \delta$,

$$\left|\frac{1}{n^2}\mathsf{SDP}(G) - \frac{1}{t^2}\mathsf{SDP}(G[S_t])\right| \le \frac{n-t}{n^2 t}(\mathsf{SDP}(G) - W) + \frac{\epsilon}{t^2}$$

$$\le \frac{n-t}{n^2 t}(\mathsf{SDP}(G) - W) + n^{3/2}\log\frac{2}{\delta}\cdot t^{-2}$$

$$= \frac{n-t}{n^2 t}(\mathsf{SDP}(G) - W) + \frac{n^{3/2}}{t^2}\sqrt{\log\left(\frac{2}{\delta}\right)}.$$

$\square$

Below, we generalize Theorem E.17 to apply to weighted graphs.

**Theorem E.18.** *Given $G = (V, E, \boldsymbol{w})$, let $S_t$ be a set of vertices with each node sampled from $V$ independently with probability $\frac{t}{n}$. Let $W = \sum_{e \in E} w_e$. With probability $1 - \delta$ over the draw of $S_t$,*

$$\left|\frac{1}{n^2}\mathsf{SDP}(G) - \frac{1}{t^2}\mathsf{SDP}(G[S_t])\right| \le \frac{n-t}{n^2 t}\left(\mathsf{SDP}(G) - \frac{|E|}{2}\right) + \frac{W}{t^2}\sqrt{\log\left(\frac{2}{\delta}\right)}.$$

*Proof.* We modify the previous proof slightly to get this result. In the weighted version, we upper bound the sum of the weighted degrees squared by

$$\sum_{i=1}^{n} \deg(i)^2 \le \max_i \deg(i) \sum_{i=1}^{n} \deg(i) = W\sum_{i=1}^{n}\deg(i) \le W^2.$$

This is tight because we can imagine allocating all edge weights to the neighboring edges of just one node.

The rest of the calculation follows that in the proof of Theorem E.17. $\square$

**Lemma E.19.** *For any $n \geq 2$, there exists a graph $G_n$ on $n$ vertices with distinct optimal solutions $X_n \neq Y_n$ to Equation (2) and a constant $C > 0$ such that $L_{G_n} \cdot X_n = L_{G_n} \cdot Y_n$ but $|\mathbb{E}[\mathsf{weight}_{G_n}(\mathsf{GWRound}(X_n))] - \mathbb{E}[\mathsf{weight}_{G_n}(\mathsf{GWRound}(Y_n))]| \geq Cn$.*

*Proof.* Let $G_n$ with $n = \{2, 4, 6, 8, ...\}$ be unweighted complete graphs with $n$ vertices. Let $X_n \in \mathbb{R}^{n \times n}$ be a $n \times n$ matrix with 1 on the diagonal and $-\frac{1}{n-1}$ off the diagonal. Let

$$Y_n = \begin{bmatrix} 1 \\ -1 \\ 1 \\ -1 \\ ... \end{bmatrix} \cdot \begin{bmatrix} 1 & -1 & 1 & -1 & ... \end{bmatrix} \in \mathbb{R}^{n \times n}.$$

We start by verifying that both $X_n$ and $Y_n$ are optimal solutions through finding their dual optimal certificates that satisfy the KKT condition. Let $\boldsymbol{z}_x = [n/4, ..., n/4]^T \in \mathbb{R}^n$. We check that this is a valid dual certificate for KKT condition:

- Dual feasibility: $\mathrm{diag}\boldsymbol{z}_x - \frac{1}{4}L_{G_n}$ is the all $\frac{1}{4}$ matrix in $\mathbb{R}^{n \times n}$, therefore also PSD, satisfying dual feasibility.

- Same objective value: $\sum_{i \in [n]} \boldsymbol{z}_{xi} = \frac{n^2}{4} = \frac{1}{4}L_{G_n} \cdot X_n$.

- Complementary slackness: $\mathrm{Tr}([\mathrm{diag}\boldsymbol{z}_x - L_{G_n}] X_n) = 0$.

- Primal feasibility: $X_n$ is PSD and has 1 on the diagonal.

Therefore, we have a valid dual certificate for the optimality of $X_n$–$X_n$ is the optimal solution for GW SDP induced by $G_n$.

We check the same thing for $Y_n$, Let $\boldsymbol{z}_y = [n/4, ..., n/4]^T \in \mathbb{R}^n$. Notice that since we are using the same dual certificate, we do not need to check for dual feasibility anymore. Because $L_{G_n} \cdot X_n = L_{G_n} \cdot Y_n$, we also won't need to check that the objective values match. We only check that:

- Complementary slackness: $\mathrm{Tr}([\mathrm{diag}\boldsymbol{z}_y - L_{G_n}] Y_n) = \sum_{i \in V} \boldsymbol{z}_{yi} - n(n-1) + 1 \cdot 2(n/2)(n/2 - 1) + (-1) \cdot (n/2)^2 = 0$.

- Primal feasibility: $Y_n$ is PSD and has all 1 on the diagonal.

Hence, $Y_n$ is also optimal.

However, we see that the expected value of GW using those two solutions differ by a gap that does not close as $n \to \infty$:

$$
\begin{aligned}
\mathbb{E}[\mathsf{GW}(Y)] - \mathbb{E}[\mathsf{GW}(X)] &= \frac{1}{\pi} \cdot \sum_{e \in E} \arccos(Y_e) - \arccos(X_e) \\
&= \frac{1}{\pi} \cdot \frac{1}{2} \cdot \frac{n^2}{2} \arccos(-1) - \frac{1}{\pi} \cdot \frac{n(n-1)}{2} \cdot \arccos\left(-\frac{1}{n-1}\right) \\
&= \frac{n^2}{4} - \frac{n(n-1)}{2\pi} \arccos\left(-\frac{1}{n-1}\right) \\
&\to \frac{\pi - 2}{4\pi} n \quad \text{as } n \to \infty
\end{aligned}
$$

Hence, choose $C = \frac{\pi-2}{4\pi}$, we have proven the statement. $\qquad\square$

**Theorem 3.3.** *Given $G = (V, E, \boldsymbol{w})$, let $S_t$ be a set of vertices with each node sampled from $V$ independently with probability $\frac{t}{n}$. Then, for $\epsilon_{SDP} = \frac{n-t}{n^2 t}\left(\mathsf{SDP}(G) - \frac{W}{2}\right)$, we have*

$$\frac{1}{n^2}\mathsf{weight}_G(\mathsf{GW}(G)) \in \left[\frac{0.878}{t^2} \mathop{\mathbb{E}}_{S_t}[\mathsf{SDP}(G[S_t])] - 0.878\epsilon_{SDP}, \frac{1}{t^2} \mathop{\mathbb{E}}_{S_t}[\mathsf{SDP}(G[S_t])]\right].$$

*Proof.* We start by lower bounding $\frac{1}{n^2}\text{weight}_G\left(\text{GW}(G)\right)$ using the fact that for any graph $G$, $0.878\text{SDP}(G) \leq \text{weight}_G\left(\text{GW}(G)\right)$:

$$\frac{1}{n^2}\text{weight}_G\left(\text{GW}(G)\right) \geq \frac{0.878}{n^2}\text{SDP}(G)$$

$$= \left(\frac{0.878}{n^2}\text{SDP}(G) + \frac{0.878}{t^2}\underset{S_t}{\mathbb{E}}[\text{SDP}(G[S_t])]\right) - \frac{0.878}{t^2}\underset{S_t}{\mathbb{E}}[\text{SDP}(G[S_t])]$$

$$= \frac{0.878}{t^2}\underset{S_t}{\mathbb{E}}[\text{SDP}(G[S_t])] - 0.878\epsilon_{\text{SDP}}.$$

We then upper bound $\frac{1}{n^2}\text{weight}_G\left(\text{GW}(G)\right)$ using the fact that for any graph $G$, $\text{weight}_G\left(\text{GW}(G)\right) \leq \text{SDP}(G)$:

$$\frac{1}{n^2}\text{weight}_G\left(\text{GW}(G)\right) \leq \frac{1}{n^2}\text{SDP}(G)$$

$$= \frac{1}{n^2}\text{SDP}(G) + \frac{1}{t^2}\underset{S_t}{\mathbb{E}}[\text{SDP}(G[S_t])] - \frac{1}{t^2}\underset{S_t}{\mathbb{E}}[\text{SDP}(G[S_t])]$$

$$= \frac{1}{t^2}\underset{S_t}{\mathbb{E}}[\text{SDP}(G[S_t])] + \left(\frac{1}{n^2}\text{SDP}(G) - \frac{1}{t^2}\underset{S_t}{\mathbb{E}}[\text{SDP}(G[S_t])]\right)$$

$$\leq \frac{1}{t^2}\underset{S_t}{\mathbb{E}}[\text{SDP}(G[S_t])]).$$

$\square$

## E.4    Omitted Proofs from Section 3.2

**Input:** Max-cut instance $G = (V, E)$; $|V| = n$, $\sigma$ sampled uniformly from $\Sigma_n$
**for** $t = \{1, 2, ..., n\}$ **do**
  | Place each node $\sigma[t]$ on the side that maximizes the number of crossing edge
  | Set $z_{\sigma[t]}$ be 1 or $-1$ according to node placement
**Return:** $z$

**Algorithm 5:** $\text{Greedy}(G)$

### E.4.1    Notations and preliminaries

Although there are many variants of the Greedy algorithm for max-cut ([25], [60], [64]), we focus on the implementation in Algorithm 5 as it is perhaps the most canonical and illustrative. For the greedy algorithm, we use a $2n^2$-dimensional vector $\boldsymbol{x}$ to describe a cut: let $i \in \{1, 2\}$ denote the 2 sides of the cut, $t$ denote the node placement time step, then

$$x_{iv}^t = \begin{cases} 1 & \text{if node } v \text{ is placed on the side } i \text{ at time step } t. \\ 0 & \text{if node } v \text{ is not placed on the side } i \text{ at time step } t \\ 0 & \text{if node } v \text{ has not been placed at time step } t. \end{cases}$$

Intuitively, we can view our max-cut problem as a problem of trying to maximize the number of crossing edges (or minimize the number of non-crossing edges) where a crossing edge is an edge $e = \{u, v\}$ such that $u$ is on side $i$ and $v$ is on side $j$. Let $a_{u_1,i_1,u_2,i_2} = 1/2$ if $(u1, u2) \in E$ and $i_1 = i_2$ and 0 otherwise. Then the objective function $z$ can be formulated as:

$$z(\boldsymbol{x}) = \sum_{\substack{1 \leq u_1,u_2 \leq n \\ i_1,i_2 \in \{1,2\}}} a_{u_1,i_1,u_2,i_2} x_{u_1,i_1} x_{u_2,i_2}.$$

Let $a'_{u_1,i_1,u_2,i_2} = 1/2$ if $(u1, u2) \in E$ and $i_1 \neq i_2$ and 0 otherwise. Another way to formulate the objective function is as follows:

$$w(\boldsymbol{x}) = \sum_{\substack{1 \leq u_1,u_2 \leq n \\ i_1,i_2 \in \{1,2\}}} a'_{u_1,i_1,u_2,i_2} x_{u_1,i_1} x_{u_2,i_2}.$$

It will be useful for later calculations to define $w(\boldsymbol{x}) = A'(\boldsymbol{x}, \boldsymbol{x})$, with $A'(\boldsymbol{x}, \boldsymbol{x})$ defined as:

$$A'(\boldsymbol{x}^{(1)}, \boldsymbol{x}^{(2)}) = \sum_{\substack{1 \le u_1, u_2 \le n \\ i_1, i_2 \in \{1,2\}}} a'_{u_1, i_1, u_2, i_2} x^{(1)}_{u_1, i_1} x^{(2)}_{u_2, i_2}.$$

Notice that $z(\boldsymbol{x})$ is the sum of weights of the non-crossing edges in cut $\boldsymbol{x}$, and if both sides of an edge are not placed, this edge will also be counted as non-crossing.

Let $b(\boldsymbol{x})$ of dimension 2n denote the partial derivative of $z(\boldsymbol{x})$ and $b'(\boldsymbol{x})$ denote the partial derivative of $w(\boldsymbol{x})$. Since $b_{u_1, i_1} = \sum_{u_2, i_2} a_{u_1, i_1, u_2, i_2} x_{u_2, i_2}$, we can see that $b_{u,i}$ denote the increase in $z(x_{u,i})$ if we were to increase $x_{u,i}$ from 0 to 1. Thus, we define the greedy step $g_{u,i}$ as follows:

$$g^t_{u,i} = \begin{cases} 1 & \text{if } i = \arg\min_i b_{u,i}(\boldsymbol{x}^{t-1}) \text{ or } \arg\max_i b'_{u,i}((\boldsymbol{x}^{t-1}) \\ 0 & \text{if } i = \arg\max_i b_{u,i}((\boldsymbol{x}^{t-1}) \text{ or } \arg\min_i b'_{u,i}((\boldsymbol{x}^{t-1}) \; . \\ 0 & \text{if i is not yet considered or has been considered} \end{cases}$$

And finally, we define the fictitious cut:

**Definition E.20.** *(Mathieu and Schudy [64]) The fictitious cut $\hat{x}^t_u$ of a partial cut $x^t_u$ is defined as*

$$\hat{x}^t_u = \begin{cases} x^t_u & \text{if u is placed before time step t} \\ \frac{1}{t} \sum_{\tau=0}^{t-1} g^\tau_v & \text{otherwise.} \end{cases}$$

Notice that $w(\cdot)$ is the same function as $\text{weight}_G(\cdot)$. We will be proving the size generalization result using $w(\cdot)$ as our objective, but [64] uses $z(\cdot)$ as the max-cut objective throughout their paper. Some of their results written in terms of $z(\cdot)$ are not directly transferrable to be a result written in terms of $w(\cdot)$. Thus, if we use those results from [64], we will explicitly provide proofs for them.

### E.4.2 Proof of Theorem 3.4

Proof of Theorem 3.4 can be divided into two parts. In the first part, we prove that the difference between fictitious cut at time step $t > t_0$ and $t_0$ is bounded. Then, we show that the normalized difference between a fictitious cut and the actual cut at time step $t$ is bounded.

The following lemma says that taking one time step, the change in fictitious cut shall be bounded by the difference in $b$-values, i.e., the partial derivative of the cut value.

**Lemma E.21** (Lemma 2.2 in [64]).

$$\mathbb{E}[z(\hat{\boldsymbol{x}}^t) - z(\hat{\boldsymbol{x}}^{t-1})] = \frac{4n^2}{t^2} + 2\frac{n}{t(n-t+1)} \mathbb{E}\left[\left|b(\hat{\boldsymbol{x}}^{t-1}) - b\left(\frac{n}{t}\boldsymbol{x}^{t-1}\right)\right|_1\right] \quad \forall t$$

**Lemma E.22** (Lemma 2.6 in [64]). $B^t_{vi} = \left|\frac{t}{n-t}\left(b_{vi}(\hat{\boldsymbol{x}}^t) - b_{vi}(\frac{n}{t}\boldsymbol{x}^t)\right)\right|$ *is a martingale with step size bounded by $\frac{4n}{n-t}$.*

Lemma E.22 does not exactly agree with the statement of Lemma 2.6 in [64], we have it under absolute value here because though [64] doesn't have it in their statement, they proved everything under an absolute value sign. Using it, we show the following lemma:

**Lemma E.23.** *For every t, we have $\mathbb{E}\left[\left|b(\hat{\boldsymbol{x}}^{t-1}) - b\left(\frac{n}{t}(\boldsymbol{x}^{t-1})\right|_1\right] = O(\sigma)$, where $\sigma = O\left(\frac{n^2}{\sqrt{t}}\right)$*

*Proof.* Because $B^t_{vi} = \left|\frac{t}{n-t}\left(b_{vi}(\hat{\boldsymbol{x}}^t) - b_{vi}(\frac{n}{t}\boldsymbol{x}^t)\right)\right|$ is a martingale step size bounded by $\frac{4n}{n-t}$, we know that $\frac{t}{n-t}\left|b(\hat{\boldsymbol{x}}^{t-1}) - b\left(\frac{n}{t}\boldsymbol{x}^{t-1}\right)\right|_1$ is a martingale with step size bounded by $\frac{4n^2}{n-t}$.

We apply Azuma-Hoeffding's inequality:

$$\mathbb{P}\left[\left\|b(\hat{\boldsymbol{x}}^{t-1}) - b\left(\frac{n}{t}\boldsymbol{x}^{t-1}\right)\right\|_1 \geq \lambda\right] = \mathbb{P}\left[\frac{t}{n-t}\left\|b(\hat{\boldsymbol{x}}^{t-1}) - b\left(\frac{n}{t}\boldsymbol{x}^{t-1}\right)\right\|_1 \geq \frac{t}{n-t}\lambda\right]$$

$$\leq 2exp\left(-\frac{\lambda^2 t^2/(n-t)^2}{32n^4 t/(n-t)^2}\right)$$

$$\leq 2exp\left(-\frac{\lambda^2 t}{32n^4}\right)$$

Let $\sigma = O\left(\frac{n^2}{\sqrt{t}}\right)$, then we shall have:

$$\mathbb{P}\left[\left\|b(\hat{\boldsymbol{x}}^{t-1}) - b\left(\frac{n}{t}\boldsymbol{x}^{t-1}\right)\right\|_1 \geq \lambda\right] \leq 2exp\left(-\frac{\lambda^2}{\sigma^2}\right)$$

$\square$

With Lemma E.21 and Lemma E.23, we are now ready to bound the difference between the fictitious cuts given two distinct time steps.

**Lemma E.24.** *For every $t \geq t_0$, we have $\mathbb{E}[z(\hat{\boldsymbol{x}}^t)] - \mathbb{E}[z(\hat{\boldsymbol{x}}^{t_0})] = O\left(\frac{n^2}{\sqrt{t_0}} + n^{1.5}(1 - \log(t))\right)$*

*Proof.*

$$\mathbb{E}[z(\hat{\boldsymbol{x}}^t)] - \mathbb{E}[z(\hat{\boldsymbol{x}}^{t_0})] = \sum_{\tau=t_0+1}^{t} \mathbb{E}[z(\hat{\boldsymbol{x}}^\tau) - z(\hat{\boldsymbol{x}}^{\tau-1})]$$

$$= \sum_{\tau=t_0+1}^{t} \frac{4n^2}{\tau^2} + 2\frac{n}{\tau(n-\tau+1)}\mathbb{E}\left[\left\|b(\hat{\boldsymbol{x}}^{\tau-1}) - b\left(\frac{n}{\tau}(\boldsymbol{x}^{\tau-1})\right)\right\|_1\right] \quad (5)$$

$$= \sum_{\tau=t_0+1}^{t} \frac{4n^2}{\tau^2} + 2\frac{n}{\tau(n-\tau+1)}O\left(\frac{n^2}{\sqrt{\tau}}\right)$$

The second equality uses Lemma E.21 and the third equality uses Lemma E.23. Now we look at the first term and the second term of (5) separately. The first term of (5) can be bounded by:

$$\sum_{\tau=t_0+1}^{t} \frac{4n^2}{\tau^2} \approx 4n^2\left(\int_{\tau=1}^{t}\frac{1}{\tau^2}d\tau - \int_{\tau=1}^{t_0}\frac{1}{\tau^2}d\tau\right) = 4n^2\left(\frac{1}{t_0} - \frac{1}{t}\right)$$

The second term of (5) can be bounded by:

$$\sum_{\tau=t_0+1}^{t} 2\frac{n}{\tau(n-\tau+1)}O\left(\frac{n^2}{\sqrt{\tau}}\right) = 2n^2 O\left(\sum_{\tau=t_0+1}^{t} \frac{n}{\tau^{1.5}(n-\tau+1)}\right)$$

$$= 2n^2 \cdot O\left(\sum_{\tau=t_0+1}^{n/2} \frac{n}{\tau^{1.5}(n-\tau+1)} + \sum_{\tau=n/2}^{t} \frac{n}{\tau^{1.5}(n-\tau+1)}\right)$$

$$\leq 2n^2 \cdot O\left(\sum_{\tau=t_0+1}^{n/2} \frac{1}{\tau^{1.5}} + \sum_{\tau=n/2}^{t} \frac{1}{\tau^{0.5}(n-\tau+1)}\right)$$

$$\leq 2n^2 \cdot O\left(\sum_{\tau=t_0+1}^{n/2} \frac{1}{\tau^{1.5}} + \frac{1}{\sqrt{n/2}}\sum_{\tau=n/2}^{t} \frac{1}{n-\tau+1}\right)$$

$$\approx 2n^2 \cdot O\left(\int_{\tau=t_0+1}^{n/2} \frac{1}{\tau^{1.5}}d\tau + \frac{1}{\sqrt{n/2}}\int_{\tau=n/2}^{t} \frac{1}{n-\tau+1}d\tau\right)$$

$$= 2n^2 \cdot O\left(\frac{1}{\sqrt{t_0+1}} - \frac{1}{\sqrt{n/2}} + \frac{1}{\sqrt{n/2}}(\log(t) - \log(n/2))\right)$$

$$= O\left(\frac{n^2}{\sqrt{t_0}} + n^{1.5}(\log(t) - 1)\right)$$

Therefore,

$$\mathbb{E}[z(\hat{\boldsymbol{x}}^t)] - \mathbb{E}[z(\hat{\boldsymbol{x}}^{t_0})] \leq O\left(\frac{n^2}{t_0} + \frac{n^2}{\sqrt{t_0}} - n^{1.5}(\log(t) - 1)\right)$$

$$= O\left(\frac{n^2}{\sqrt{t_0}} + n^{1.5}(\log(t) - 1)\right)$$

$\square$

We show that the same result holds for the other objective $w(\cdot)$:

**Lemma E.25.** *For every $t \geq t_0$, we have $\mathbb{E}[w(\hat{\boldsymbol{x}}^t)] - \mathbb{E}[w(\hat{\boldsymbol{x}}^{t_0})] = O\left(\frac{n^2}{\sqrt{t_0}} + n^{1.5}(\log(t) - 1)\right)$*

*Proof.* Notice that for any fictitious cut $\hat{\boldsymbol{x}}^t$, $w(\hat{\boldsymbol{x}}^t) + z(\hat{\boldsymbol{x}}^t) = |E|$. Therefore,

$$\mathbb{E}[z(\hat{\boldsymbol{x}}^t)] - \mathbb{E}[z(\hat{\boldsymbol{x}}^{t_0})] = \mathbb{E}[z(\hat{\boldsymbol{x}}^t) - z(\hat{\boldsymbol{x}}^{t_0})]$$
$$= \mathbb{E}[|E| - w(\hat{\boldsymbol{x}}^t) - (|E| - w(\hat{\boldsymbol{x}}^{t_0}))]$$
$$= \mathbb{E}[w(\hat{\boldsymbol{x}}^{t_0}) - w(\hat{\boldsymbol{x}}^t)]$$
$$= O\left(\frac{n^2}{\sqrt{t_0}} + n^{1.5}(\log(t) - 1)\right)$$

$\square$

Now we have a bound on the change of fictitious cut weight from one time step to another. We will now bound the normalized difference between a fictitious cut weight and the actual cut weight at a fixed time step.

We start by stating the following facts:

**Lemma E.26** (Lemma 3.4 and 3.5 in [64]). *For any $\sigma \geq 0$, let $C(\sigma) = \{X|\lambda > 0, \mathbb{P}[X \geq \sigma + \lambda] \leq e^{-\lambda^2/\sigma^2}\}$. Let $\sigma$ and $\alpha$ by positive constant and $X$ and $Y$ be random variables. Then:*

- *If $X \in C(\sigma)$, then $\alpha X \in C(\alpha \sigma)$*

- *If $X \in C(\sigma)$ and $Y \leq X$, then $Y \in C(\sigma)$*

- *The random variable with constant value $\sigma$ is in $C(\sigma)$*

- *If $X \in C(\sigma_x)$ and $Y \in C(\sigma_y)$, then $X + Y \in C(\sigma_x + \sigma_y)$*

Then, we show that $B'^t_{vi}$ is also a martingale.

**Lemma E.27** (Lemma 2.5 in [64]). *$Z^t_v = \frac{t}{n-t}(\hat{\boldsymbol{x}})t^t - (n/t)\boldsymbol{x}^t_v)$ is a martingale.*

**Lemma E.28.** *$B'^t_{vi} = \left| \frac{t}{n-t}\left(b'_{vi}(\hat{\boldsymbol{x}}^t - b'_{vi}(\frac{n}{t}\boldsymbol{x}^t))\right) \right|$ is a martingale with step size bounded by $\frac{4n}{n-t}$.*

*Proof.* Let $i_1, i_2 \in \{1, 2\}$. Because $b'_{u_1, i_1} = \sum_{u_2, i_2} a'_{u_1, i_1, u_2, i_2} x_{u_2, i_2}$, $B'^t_{vi}$ is a martingale following Lemma E.27 by linearity. We also bound the step size by linearity with $Z^t_u$, which would be at most $\frac{4n}{n-t}$. $\qquad\square$

**Lemma E.29** (Lemma 3.6 in [64] in terms of $w(x)$). *For a fixed cut $\boldsymbol{y}$*

$$\mathbb{P}\left[\left|w\left(\frac{n}{t}\boldsymbol{x}^t_{(\boldsymbol{y})}\right) - w\left(\hat{\boldsymbol{x}}^t_{(\boldsymbol{y})}\right)\right| \geq \sigma + \lambda\right] \leq e^{\frac{-\lambda^2}{\sigma^2}}$$

*where*

$$\sigma = \frac{n^2}{\sqrt{t}}$$

*Proof.* Let $F'_t = w\left(\frac{n}{t}\boldsymbol{x}^t_{(\boldsymbol{y})}\right) - w\left(\hat{\boldsymbol{x}}^t_{(\boldsymbol{y})}\right)$. Define $\bar{\boldsymbol{x}} = \frac{t}{n}\hat{\boldsymbol{x}}^t_{(\boldsymbol{y})}$. We can show that (drop t in the next chunk of calculation for simplicity)

$$F' = w\left(\frac{n}{t}\boldsymbol{x}_{(\boldsymbol{y})}\right) - w\left(\hat{\boldsymbol{x}}_{(\boldsymbol{y})}\right)$$

$$= \left(\frac{n}{t}\right)^2 \left(\sum_{u_1, i_1, u_2, i_2} a'_{u_1, i_1, u_2, i_2}(x_{u_1, i_1} x_{u_2, i_2} - \bar{x}_{u_1, i_1} \bar{x}_{u_2, i_2})\right)$$

$$= \left(\frac{n}{t}\right)^2 (n - t)A'(\frac{\boldsymbol{x} - \bar{\boldsymbol{x}}}{n - t}, (\boldsymbol{x} + \bar{\boldsymbol{x}})$$

Let $D'^t = A'(\frac{\boldsymbol{x}^t - \bar{\boldsymbol{x}}^t}{n-t}, \boldsymbol{x}^t + \bar{\boldsymbol{x}}^t) - A'(\frac{\boldsymbol{x}^{t-1} - \bar{\boldsymbol{x}}^{t-1}}{n-t}, \boldsymbol{x}^{t-1} + \bar{\boldsymbol{x}}^{t-1})$, then

$$A'(\frac{\boldsymbol{x}^t - \bar{\boldsymbol{x}}^t}{n - t}, \boldsymbol{x}^t + \bar{\boldsymbol{x}}^t) = \sum_{\tau=1}^{t}(D^\tau - \mathbb{E}[D^\tau | S^{t-1}]) + \sum_{\tau=1}^{t} \mathbb{E}[D^\tau | S^{t-1}]$$

By Lemma E.26, it is sufficient to show each of these two terms in $C\left(O\left(\frac{n^2}{\sqrt{t}}\right)\right)$.

The first term is a martingale with step $O(\frac{t}{n-t})$ because, at each time step, there could be at most a change of $t$ in objective value (in that case node $\sigma[t]$ has $t$ outgoing edges connecting to nodes that have been placed and all of them are placed so that all edges are added to the cut). We can apply Azuma-Hoeffding's inequality knowing the step size and in that case $\sigma = \frac{2t^3}{(n-t)^2}$. When both $t > \frac{n}{2}$ and $t < \frac{n}{2}$, we can show that $\frac{2t^3}{(n-t)^2}$ is dominated by $O\left(n^2\sqrt{\frac{n-t}{tn}}\right)$ and thus by $O\left(\frac{n^2}{\sqrt{t}}\right)$ simply because $\sqrt{\frac{n-t}{n}} \leq 1$.

The second term can be re-written as:

$$A'\left(\frac{\boldsymbol{x}^t - \bar{\boldsymbol{x}}^t}{n-t} - \frac{\boldsymbol{x}^{t-1} - \bar{\boldsymbol{x}}^{t-1}}{n-t+1}, \boldsymbol{x}^{t-1} + \bar{\boldsymbol{x}}^{t-1}\right) + A'\left(\frac{\boldsymbol{x}^{t-1} - \bar{\boldsymbol{x}}^{t-1}}{n-t+1}, \boldsymbol{x}^t + \bar{\boldsymbol{x}}^t + \boldsymbol{x}^{t-1} + \bar{\boldsymbol{x}}^{t-1}\right)$$

$$+ A'\left(\frac{\boldsymbol{x}^t - \bar{\boldsymbol{x}}^t}{n-t} - \frac{\boldsymbol{x}^{t-1} - \bar{\boldsymbol{x}}^{t-1}}{n-t+1}, \boldsymbol{x}^t + \bar{\boldsymbol{x}}^t + \boldsymbol{x}^{t-1} + \bar{\boldsymbol{x}}^{t-1}\right)$$

$$\tag{6}$$

Notice that if we expand the first term of (6), each element indexed by $u, i$ is a martingale difference $\frac{\boldsymbol{x}^t - \bar{\boldsymbol{x}}^t}{n-t} - \frac{\boldsymbol{x}^{t-1} - \bar{\boldsymbol{x}}^{t-1}}{n-t+1}$ times a constant (because we are conditioning on $S^{t-1}$), and martingale difference has an expected value of zero. The third term is bounded by $O(\frac{t}{n-t})$.

The second term of (6) we can rewrite as:

$$A'\left(\frac{\boldsymbol{x}^{t-1} - \bar{\boldsymbol{x}}^{t-1}}{n-t+1}, \boldsymbol{x}^t + \bar{\boldsymbol{x}}^t + \boldsymbol{x}^{t-1} + \bar{\boldsymbol{x}}^{t-1}\right)$$
$$= \frac{t}{n(n-t)}\left(\boldsymbol{x}^t + \bar{\boldsymbol{x}}^t + \boldsymbol{x}^{t-1} + \bar{\boldsymbol{x}}^{t-1}\right)\left(b'(\hat{x}^t) - b'(\frac{n}{t}x^t)\right)$$

By Lemma E.28, we know that $\left|b(\hat{\boldsymbol{x}}^{t-1}) - b\left(\frac{n}{t}(\boldsymbol{x}^{t-1})\right|_1 \in C(O(n^2/\sqrt{t}))$.

Now that every thing is in $C(O(n^2/\sqrt{t}))$, by Lemma E.26, we have that $\left|w\left(\frac{n}{t}\boldsymbol{x}^t_{(\boldsymbol{y})}\right) - w\left(\hat{\boldsymbol{x}}^t_{(\boldsymbol{y})}\right)\right| \in C(O(n^2/\sqrt{t}))$.

$\square$

**Lemma E.30.** *For a fixed $\boldsymbol{y}$,*

$$\mathbb{E}\left[\max_{y \in Y}\left|w\left(\frac{n}{t}\boldsymbol{x}^t_{(\boldsymbol{y})}\right) - w\left(\hat{\boldsymbol{x}}^t_{(\boldsymbol{y})}\right)\right|\right] = O(\sigma)$$

*where $\sigma = O(n^2/\sqrt{t})$*

*Proof.* Using Lemma E.29, we integrate $\mathbb{P}\left[\left|w\left(\frac{n}{t}\boldsymbol{x}^t_{(\boldsymbol{y})}\right) - w\left(\hat{\boldsymbol{x}}^t_{(\boldsymbol{y})}\right)\right| \geq \sigma\right]$ over $\lambda = 0$ to $\infty$, which gets us $O(\sigma)$. $\square$

**Theorem 3.4.** *Given an unweighted graph $G = (V, E)$, let $S_t$ be $t$ vertices sampled from $V$ uniformly without replacement. For any $\epsilon \in [0, 1]$ and $t \geq \frac{1}{\epsilon^2}$,*

$$\left|\frac{1}{n^2}\mathbb{E}[\text{weight}_G\left(\text{Greedy}(G)\right)] - \frac{1}{t^2}\mathbb{E}[\text{weight}_{G[S_t]}\left(\text{Greedy}(G[S_t]))\right]\right| \leq O\left(\epsilon + \frac{\log(t)}{\sqrt{n}}\right).$$

*Proof.*

$$\left|\frac{1}{n^2}\mathbb{E}[w(\boldsymbol{x}^n)] - \frac{1}{t^2}\mathbb{E}[w(\boldsymbol{x}^t)]\right| = \left|\mathbb{E}\left[\frac{1}{n^2}w(\boldsymbol{x}^n) - \frac{1}{t^2}w(\boldsymbol{x}^t)\right]\right|$$

$$= \left|\mathbb{E}\left[\frac{1}{n^2}w(\hat{\boldsymbol{x}}^n) - \frac{1}{t^2}w(\boldsymbol{x}^t)\right]\right|$$

$$\leq \left|\mathbb{E}\left[\frac{1}{n^2}w(\hat{\boldsymbol{x}}^n) - \frac{1}{n^2}w(\hat{\boldsymbol{x}}^t)\right]\right| + \left|\mathbb{E}\left[\frac{1}{n^2}w(\hat{\boldsymbol{x}}^t) - \frac{1}{t^2}w(\boldsymbol{x}^t)\right]\right|$$

$$= \frac{1}{n^2}\left(\left|\mathbb{E}\left[w(\hat{\boldsymbol{x}}^n) - w(\hat{\boldsymbol{x}}^t)\right]\right| + \left|\mathbb{E}\left[w(\hat{\boldsymbol{x}}^t) - w(\frac{n}{t}\boldsymbol{x}^t)\right]\right|\right)$$

By lemma E.25, we can bound the first term by

$$\left|\mathbb{E}\left[w(\hat{\boldsymbol{x}}^n) - w(\hat{\boldsymbol{x}}^t)\right]\right| \leq \mathbb{E}_\sigma\left[|w(\hat{\boldsymbol{x}}^n) - w(\hat{\boldsymbol{x}}^t)|\right]$$

$$\leq O\left(\frac{n^2}{\sqrt{t}} + n^{1.5}(\log(t) - 1)\right)$$

Let $\boldsymbol{y}^*$ be the cut found on the first $t$ element of $\sigma$ by algorithm 5. By lemma E.30, we can bound the second term by

$$\left| \mathbb{E}\left[ w(\hat{\boldsymbol{x}}^t) - w(\frac{n}{t}\boldsymbol{x}^t) \right] \right| = \mathbb{E}\left[ \left| w(\hat{\boldsymbol{x}}^t_{(\boldsymbol{y}^*)}) - w(\frac{n}{t}\boldsymbol{x}^t_{(\boldsymbol{y}^*)}) \right| \right] = O\left( \frac{n^2}{\sqrt{t}} \right)$$

Combining the results above gives us:

$$\left| \frac{1}{n^2}\mathbb{E}[w(\boldsymbol{x}^n)] - \frac{1}{t^2}\mathbb{E}[w(\boldsymbol{x}^t)] \right| \leq \frac{1}{n^2}O\left( \frac{n^2}{\sqrt{t}} + n^{1.5}(\log(t) - 1) \right)$$

$$= O\left( \frac{1}{\sqrt{t}} + \frac{1}{\sqrt{n}}(\log(t) - 1) \right)$$

$$= O\left( \frac{1}{\sqrt{t}} + \frac{\log(t)}{\sqrt{n}} \right)$$

Lastly, we solve for the sample complexity bound. Set $\epsilon \geq \frac{1}{\sqrt{t}}$, then when $t \geq \frac{1}{\epsilon^2}$, we get

$$\left| \frac{1}{n^2}\mathbb{E}[w(\boldsymbol{x}^n)] - \frac{1}{t^2}\mathbb{E}[w(\boldsymbol{x}^t)] \right| \leq O\left( \epsilon + \frac{\log(t)}{\sqrt{n}} \right)$$

$\square$

