# OpenReview forum: "Accelerating data-driven algorithm selection for combinatorial partitioning problems"
_NeurIPS.cc/2025/Conference — NeurIPS 2025 spotlight_

### Official Review · Reviewer_orjk · 2025-06-30

**Clarity:** 2
**Significance:** 2
**Originality:** 4
**Rating:** 4
**Confidence:** 2

**Summary:**

The paper introduces *size generalization*, the problem of predicting the accuracy on the full dataset given a subset of it. They then analyze this for several different classical tasks, such as $k$-center, $k$-means, single-linkage clustering and max-cut. They prove bounds on the sizes of the samples and the numbers of accesses to ground truth labels required to achieve the size generalization.

**Questions:**

My primary question is regarding the similarities and differences between the proof techniques for the various optimization problems. Are there common themes that can be extracted and applied more generally? If not, why not?

**Ethical Concerns:**

["NO or VERY MINOR ethics concerns only"]

**Final Justification:**

I believe this paper is interesting and presents a nice theoretical way to think about predicting an algorithm's success. However, the lack of a unified framework for studying size generalization and the chosen algorithms prevent me from giving this paper a stronger acceptance score.

**Limitations:**

I think a formal limitations discussion would be helpful. The current discussed limitations are oriented around technical details of the proof techniques. However, the paper's primary limitation seems to actually be that size-generalization is most actively studied in the deep learning community right now (with scaling laws being the most extreme example) but the paper does not interface with that literature at all. Which is perfectly fine! But worth mentioning.

**Paper Formatting Concerns:**

A separate limitations section/subsection would be nice.

**Quality:**

3

**Strengths And Weaknesses:**

Strengths:
- I am honestly quite shocked that this direction does not have more literature on it. I am familiar with coresets for clustering and, separately, with a lot of topics in deep learning research. So when I saw the description in the introduction, I thought there must surely be significant work on this idea of predicting the accuracy on a full dataset using only a subset of it. After all, I know a similar thing is done for coresets (but different in that coresets attempt to find a sample which preserves the cost), and the field of deep learning is actively interested in analyzing which training run will land where (in a more empirical sense). However, after spending time on a literature review, I could not find papers discussing this specific task of size generalization done in a theoretically rigorous way. This is quite surprising to me and implies that the paper is filling a nice gap in the literature. Unless I am mistaken and have missed something, I consider this a significant strength of the paper.
- The resulting bounds feel consistent with what one might expect given familiarity with sublinear-time algorithms, sensitivity sampling, etc.
- Some of the proof techniques are quite nice. I particulalry like the continuous extension of k-center and the analysis of the minimax distance with respect to sampling.

Weaknesses:
- I found this paper quite difficult to read and wish the results were presented differently. As it stands, this paper feels like three papers: size generalization for center-based clustering, size generalization for single-linkage and size generalization for max-cut. These results are presented by giving the proof techniques specific to each problem case. However, I find myself wanting an understanding of the general themes inherent to the task of size generalization. That is, I cannot see much overlap at all between the methods employed by the proof sketches in, for example, Theorems 2.2, 2.7 and Theorem 3.4. Even if these are, in fact, quite separate approaches, it would be good to have an overview somewhere after the introduction which describes the general techniques one might expect and how to approach a size-generalization problem. How does this relate to PAC-learning techniques? Are there techniques which can consistently be applied to size generalization and, if not, why not?
- While I strongly commend the general topic of size generalization (as discussed in the strengths), I find the chosen motivation for studying these specific algorithms... detached from the state of the field. That is to say, I do not actually see the primary use case for size-generalization to be in combinatorial algorithms or clustering: there is usually sufficient compute to run k-means on any size dataset required.
  - Instead, it feels much more relevant to me to study these ideas on neural networks, where training runs are *significantly* more expensive and where the empirical utility is much higher. Indeed, at least half of the list of empirical use cases for size-generalization listed on lines 43-46 are about neural network convergence. This is not counting the entire field of scaling laws which has seen huge growth in the last 3-4 years and is oriented *precisely* around the topic of size-generalization.
  - Complete nitpick: as a person who has dipped toes into both TCS and the more empirical side of machine learning, I find it frustrating when TCS paper introductions motivate algorithms by saying they are used for tasks when, in reality, they are no longer used for that task. For example, clustering is never used for image segmentation anymore and the reference to chip design is from 1988.

In summary, I consider the task of studying size-generalization a very relevant one and the paper's results feel clean. However, it seems that the utility of the topic would be better suited to other algorithms and the paper did not provide blueprints which would explain which (if any) of the proof techniques can be extended to the topic of size-generalization more broadly. Of course, I am not requesting that the authors prove additional size-generalization bounds for new optimization problems -- I am moreso explaining why the clarity and significance scores are not higher given my praise for this paper's strengths.

---

> ### Author Rebuttal · Authors · 2025-07-31
>
> Thank you for carefully reading our paper, conducting a thorough literature survey, and providing such insightful comments. We’re delighted that you view our work as filling “a nice gap in the literature” and were surprised to find no prior theoretical analysis of this question. We are also encouraged that you liked some of our proof techniques and think they are “quite nice”. Below, we address each point.
>
>
> **W1 & Q1: On developing a general technique for size-generalization**
>
>
> Thank you for raising the question of whether a unified framework guides the analysis of size generalization. In our paper, we deliberately chose a diverse set of canonical algorithms—ones that perform well in practice. These algorithms range from SDP‑based methods (Goemans–Williamson) to purely combinatorial heuristics (Gonzalez’s k‑centers, greedy max‑cut), showcasing size‑generalization behavior across fundamentally different paradigms and providing a proof-of-concept to guide future research. We view this breadth as one of our paper’s strengths.
>
> Moreover, we believe some of the methods developed in this paper can extend to a wider class of combinatorial optimization problems. For instance, the primal–dual analysis used for Max‐Cut SDP should apply to any SDP formulation with strong duality (e.g., correlation‐clustering rounding, sensor‐network localization).
>
> That said, we fully agree that a general theoretical toolkit for size generalization would greatly inspire future work. We will make this and the SDP extensions explicit in the Conclusion by adding:
>
> *Future work includes extending size‐generalization guarantees to broader classes of optimization problems, such as integer programming and additional SDP‐based algorithms (e.g., randomized rounding for correlation clustering). Another exciting direction for future work is to formulate a unified theoretical toolkit for size generalization—one that prescribes, given a new combinatorial algorithm, which structural properties to verify to guarantee that its performance on large instances extrapolates from small ones.*
>
> **W2: Size generalization seems especially relevant for neural network training.**
>
> We thank the reviewer for highlighting the rich developments in size‐generalization for neural network training and scaling laws. However, our motivation comes from a pressing computational bottleneck in combinatorial optimization: even polynomial‑time algorithms, such as those used by SDP solvers, can be intractable on moderately large instances. For example, modern SDP solvers (as used in the GW algorithm) have worst‑case runtimes on the order of $O(n^{3.5})$ per iteration, requiring hours or days for graphs with only a few thousand nodes.
>
> The standard paradigm in ML for combinatorial optimization is to train a model or design a heuristic on problem instances of modest size (where exact or near‑exact solutions are feasible) and then deploy it on much larger problems—e.g., routing, scheduling, or integer programs with thousands more variables—without retraining. The works cited in line 45 are about accelerating combinatorial algorithms and solvers, rather than neural network training. This strategy has seen orders‑of‑magnitude speedups in solvers for large MILPs, but there are no existing theoretical guarantees. Our work takes the first step in filling this gap by providing a theoretical framework for size generalization.
>
> **W3: Outdated references for application of clustering and max-cut**
>
> We think this is a great point. We will modify the first sentence in the Introduction with more recent applications of clustering and max-cut:
>
> “Combinatorial partitioning problems such as clustering and max-cut are fundamental in machine learning (Ben-David ’18; Ezugwu et al. ’22), finance (Soloviev et al. ’25), and biology (Roy, Rudra ’24).”
>
> Thank you very much for the careful review and inspiring discussion. We are grateful for your thoughtful questions, which allowed us to clarify key aspects and better highlight our contributions. In the camera-ready version, we will:
>
> - Add the discussion about future‑work directions on a unified toolkit and SDP extensions (W1, Q1) in the Conclusion.
> - Expand our motivation in the Introduction to highlight more recent combinatorial optimization use cases (W3).
>
> Please let us know whether we have addressed your concerns and what can be further improved; we are happy to continue the discussion anytime until the end of the discussion period. We respectfully ask that if you feel more positively about our paper, to please consider updating your score. Thank you!

---

> > ### Comment · Reviewer_orjk · 2025-08-01
> >
> > Thank you for the thoughtful rebuttal. I appreciate the author's changes but do not feel sufficiently convinced to change my score. My primary concerns are that the methods do not strike me as particularly relevant for the current state of machine learning applications and that the lack of a unifying theoretical blueprint leaves extending the results challenging. I believe this paper should be accepted, but these concerns prevent me from rating it a 5 or 6. Thank you.

---

> > > ### Author Response · Authors · 2025-08-02
> > >
> > > Thank you so much for carefully reading our responses. Once again, we appreciate your thoughtful comments, which inspire the discussion. Please let us know if you have any further questions or suggestions before the end of the rebuttal period, and we will be happy to address them!

---

### Official Review · Reviewer_uRvU · 2025-07-02

**Clarity:** 2
**Significance:** 2
**Originality:** 2
**Rating:** 3
**Confidence:** 3

**Summary:**

This work introduces the concept of *size generalization*, which means predicting an algorithm’s performance on a large instance by evaluating it on a smaller subsample drawn from that instance. This work establishes size generalization bounds when using smaller instances—sampled either uniformly or randomly from the original data—for three widely used clustering algorithms (single-linkage, k-means++, and Gonzalez’s k-centers heuristic) as well as two max-cut algorithms (Goemans-Williamson and Greedy).

**Questions:**

* *Size generalization* aims to predict an algorithm’s performance on a large instance by evaluating it on a smaller subsample, thereby improving efficiency. I wonder why the timing performance of both the subsampled and the original algorithms was not reported in the experiments.

**Ethical Concerns:**

["NO or VERY MINOR ethics concerns only"]

**Final Justification:**

After the rebuttal and discussion stage, I will maintain my original score. I agree that this paper introduces some new ideas for fast evaluating algorithmic performance. However, there are still several significant limitations.

1. The application scope is relatively narrow. Like the clustering part, the method can only handle **greedy selection based** clustering algorithms, not all possible clustering algorithms. As I emphasized in my previous feedback, k-means++ and Gonzalez are already two well-accepted clustering algorithms, and there is not much need to consider other types of greedy selection based clustering algorithms.

2. Another major concern is the parameter $\zeta$. The authors argued that this value is analogous to condition number in numerical linear algebra. But,  condition number is much more natural and easy to interpret in geometry. In contrast, the parameter $\zeta$ is too complicated and not that natural to understand. More importantly, for different "f", the value of $\zeta$ could be totally different. That means, $\zeta$ depends not only the input distribution, **but also each individual tested algorithm**. This is very different with the condition number, which only depends on the input matrix. This "f"-dependence property also significantly weakens the theoretical value and elegance of this paper.

3. The experimental part is very limited. This point is also mentioned by other reviewers. I suggest the authors to greatly enlarge this part in future.

**Limitations:**

yes

**Quality:**

2

**Strengths And Weaknesses:**

## Strengths

* This work introduces the notion of *size generalization*, which has practical significance in many scenarios.
* It also provides size generalization bounds when using smaller instances—sampled either uniformly or randomly from the original data—for three widely used clustering algorithms and two max-cut algorithms.

## Weaknesses

* The experimental section is overly simplistic for real‐world applications. In the clustering experiments, every instance is partitioned into k=2 clusters—an inappropriate choice given the complexity of the datasets. Moreover, even for large datasets, the authors fix the problem size at n=500, which contradicts their stated goal of addressing big-data scalability. Finally, the results do not demonstrate whether the proposed algorithms can reliably approximate the ground-truth cost when the sampled data size is very small (for example, sublinear in
n).

* The proposed method requires two “oracles” to run in a semi-supervised fashion: (1) a distance oracle that returns the distance between any two points, and (2) a ground-truth oracle that provides the true clustering or cut value for any subset—each assumed to answer in constant time. Such idealized oracles are rarely available in practical, real-world settings, calling into question the method’s feasibility.

* The authors should compare their fast-selection framework against existing sublinear-time solutions for k-means, k-center, and Max-Cut—such as “Sublinear Time Approximate Clustering, Mishra et al, SODA'01” “Sublinear Algorithms for Greedy Selection in High Dimensions, Chen et al, UAI'22”, “Sublinear Algorithms for MAXCUT and Correlation Clustering, Bhaskara et al., ICALP'18”, and many other results in this "sub-linear time algorithms" area (two good surveys from Ronitt Rubinfeld & Asaf Shapira,  and  Artur Czumaj & Christian Sohler). Although these works optimize a different cost function, they directly address sublinear-time solution of the original problem; if one can compute an approximate solution in sublinear time, the need for a separate fast-selection step may be less compelling.

* There appears to be an error in Fig. 1(a) that may cause confusion: the legends for the original clustering algorithm and the subsampled version seem to be swapped.

---

> ### Author Rebuttal · Authors · 2025-07-31
>
> Thank you very much for your insightful review! We appreciate your constructive comments and are encouraged to find that you think the notion of size generalization has “practical significance in many scenarios”, and that we can provide a guarantee for a spectrum of canonical algorithms: “three widely used clustering algorithms and two max-cut algorithms”. We address the reviewer's questions below:
>
> **W1: Clustering experiment with n = 500 and k = 2 is simplistic.**
>
> Thank you for the suggestion. We have strengthened our experiments by including larger, more complex clustering instances, and will update the Experiment section in the camera-ready version:
>
> - MNIST, Gaussian Mixture (GM), Noisy Circle (NC) with $n=2000$, $k=2$
> - MNIST with $n=3000$, $k=5$ (we omit due to space constraint)
> - MNIST with $n=4000$, $k=10$.
>
> In each setting, we sample at fractions ${0.2,0.4,0.6,0.8,1.0}$ and report mean accuracy ± std over 10 trials.
>
> $n  = 2000$, $k = 2$:
> |            | Fraction of nodes sampled	| 0.2       	| 0.4       	| 0.6       	| 0.8       	| 1         	|
> |--------|------------|---------------|---------------|---------------|---------------|---------------|
> | mnist   | SL     	| 0.521 (0.013) | 0.513 (0.009) | 0.510 (0.008) | 0.508 (0.004) | 0.500     	|
> |   | k-centers  | 0.651 (0.089) | 0.648 (0.088) | 0.650 (0.090) | 0.648 (0.089) | 0.638 (0.092) |
> |   | k-means++  | 0.661 (0.102) | 0.667 (0.102) | 0.667 (0.103) | 0.660 (0.105) | 0.669 (0.102) |
> | GM  	| SL     	| 0.511 (0.011) | 0.511 (0.007) | 0.511 (0.009) | 0.505 (0.004) | 0.501     	|
> |  	| k-centers  | 0.712 (0.090) | 0.715 (0.090) | 0.710 (0.091) | 0.710 (0.091) | 0.715 (0.087) |
> | 	| k-means++  | 0.707 (0.091) | 0.711 (0.091) | 0.710 (0.090) | 0.707 (0.087) | 0.712 (0.089) |
> | NC  	| SL     	| 0.513 (0.009) | 0.513 (0.007) | 0.509 (0.005) | 0.505 (0.003) | 0.501     	|
> | 	| k-centers  | 0.609 (0.081) | 0.604 (0.084) | 0.594 (0.083) | 0.594 (0.085) | 0.577 (0.084) |
> |  	| k-means++  | 0.629 (0.061) | 0.625 (0.063) | 0.628 (0.062) | 0.625 (0.063) | 0.624 (0.065) |
>
>
> $n  = 4000$, $k = 10$:
> |    | Fraction of nodes sampled| 0.2       | 0.4       | 0.6       | 0.8       	| 1         |
> |---------|------------|---------------|---------------|---------------|---------------|---------------|
> | mnist   | SL     	| 0.12 (0.005)  | 0.113 (0.004) | 0.108 (0.004) | 0.104 (0.001) | 0.101     	|
> |   | k-centers  | 0.299 (0.045) | 0.29 (0.045)  | 0.287 (0.045) | 0.283 (0.047) | 0.288 (0.044) |
> |   | k-means++  | 0.352 (0.039) | 0.352 (0.039) | 0.352 (0.037) | 0.351 (0.039) | 0.351 (0.037) |
>
>
>
> In all settings—even $n = 4000$, $k = 10$—rankings stabilize by 40–60 % subsampling, confirming size-generalization at scale.
>
>
> **W2: Can we approximate ground‑truth with sublinear samples?**
>
> Thank you for raising this question! The short answer is yes: we subsampled at sizes $n ^ {\alpha}$ with $\alpha = 1/2, 3/4, 5/6, 7/8, 9/10, 1$ and measured mean accuracy with std (10 runs each):
>
> |    | number of nodes sampled  	| $n^{1/2} $   	| $n^{3/4}$    	| $n^{5/6} $   	| $n^{7/8}$    	| $n^{9/10}$   	| $n$          	|
> |---------|-------------|----------------|----------------|----------------|----------------|----------------|----------------|
> | mnist   | k-means++   | 0.724 (0.122)  | 0.727 (0.121)  | 0.722 (0.122)  | 0.722 (0.123)  | 0.723 (0.122)  | 0.722 (0.121)  |
> |   | k-centers   | 0.745 (0.101)  | 0.726 (0.101)  | 0.721 (0.101)  | 0.718 (0.101)  | 0.717 (0.105)  | 0.706 (0.104)  |
> |   | SL      	| 0.574 (0.043)  | 0.537 (0.026)  | 0.518 (0.020)  | 0.512 (0.013)  | 0.508 (0.007)  | 0.501      	|
> | NC  	| k-means++   | 0.627 (0.061)  | 0.630 (0.060)  | 0.628 (0.062)  | 0.628 (0.062)  | 0.629 (0.061)  | 0.626 (0.062)  |
> | 	| k-centers   | 0.616 (0.063)  | 0.623 (0.062)  | 0.618 (0.063)  | 0.621 (0.061)  | 0.622 (0.061)  | 0.620 (0.063)  |
> |  	| SL      	| 0.650      	| 0.803      	| 1.0 (0)    	| 1.0 (0)    	| 1.0 (0)    	| 1.0        	|
> | GM  	| k-means++   | 0.730 (0.094)  | 0.732 (0.093)  | 0.730 (0.094)  | 0.729 (0.095)  | 0.731 (0.092)  | 0.731 (0.093)  |
> |  	| k-centers   | 0.728 (0.095)  | 0.733 (0.094)  | 0.725 (0.095)  | 0.728 (0.096)  | 0.726 (0.094)  | 0.729 (0.093)  |
> |  	| SL      	| 0.645 (0.070)  | 0.538 (0.023)  | 0.522 (0.023)  | 0.517 (0.013)  | 0.518 (0.011)  | 0.502      	|
>
> In almost all cases, a sublinear sample size is enough to approximate the ground-truth accuracy. For example, k‑means++ and Gonzalez’s k‑centers recover 98–99% of full‑data accuracy by $n^{3/4}$. Single‑linkage shows higher variance but still converges near the ground truth.
>
> Even when sublinear sampling isn’t sufficient, we find that any degree of subsampling still speeds up algorithm selection for the combinatorial problems we analyze. For example, on max‑cut instances with $n=500$ nodes, we measure average solve times (in seconds) at subsample fractions ${0.2, 0.4, 0.6, 0.8, 1.0}$:
>
> |    Fraction of nodes sampled | 0.2 	| 0.4  	| 0.6  	| 0.8   	| 1    	|
> |--------|---------|----------|----------|-----------|----------|
> | GW 	| 0.5938  | 12.3565  | 72.2823  | 370.3707  | 961.4030 |
> | Greedy | 0.0187  | 0.0834   | 0.1575   | 0.3231	| 0.5781   |
>
> We observe that:
>
> - An **80% subsample** cuts GW’s runtime by over **60%** (from 961s to 370s)
> - A **60% subsample** cuts GW’s runtime by roughly **90%** (to 72 s)
> - Even the very fast Greedy enjoys relative speedups
>
> Because our bounds vary smoothly with the subsample fraction, practitioners can trade a small accuracy loss for large compute savings while still reliably selecting the best algorithm.
>
> **W3: Concern about the two-oracle setting not available in practice**
>
> Our use of the distance and the ground-truth oracle matches the query‑based semi‑supervised clustering literature (e.g., Dasgupta, Long ’05; Balcan, Blum, Gupta ’09). Moreover, as we state in lines 136-137 in Section 2, the ground‑truth oracle can be instantiated by the same‑cluster query oracle, an oracle that is well-studied in prior work (Ashtiani, Kushagra, Ben-David ’16; Mazumdar and Saha, ’17; Saha,  Subramanian ’19). Thus, we argue that our model does not rely on any stronger or more “idealized” oracles than those already accepted by prior work.
>
>
> **W4: Comparison to the sublinear-time algorithm literature**
>
> Thank you for your feedback! We will expand our related work section (Section 1.2 and Appendix B.1) as follows:
>
> Our work is related in spirit to the long-standing area of sublinear-time algorithms, though with several key distinctions. Sublinear time approximation for max-cut dates back to 1998, when a constant time approximation scheme that approximates the max-cut objective for dense graphs within an error of $\epsilon n^2$ was developed (Goldreich, Holdwasser, Ron ’98). Many later works follow (Frieze, Kannan, ’99; Bhaskara, Daruki, Venkatasubramanian ’18). For example, Bhaskara et al. show that core-sets of size $O(n^{1 - \delta})$ can approximate max-cut in graphs with average degree $n^{\delta}$. Despite having weaker approximation guarantees compared to the GW algorithm, these algorithms can have much faster runtimes.
>
> The goal of size generalization for max-cut, however, differs fundamentally from the goal of sublinear time approximation algorithms. Rather than estimating the optimal cut value, we directly estimate the empirical performance of specific heuristics. This enables principled algorithm selection among the best-performing methods, instead of defaulting to the fastest algorithm with looser guarantees. We note that Reviewers 74EK and orjk emphasized this objective is well-motivated by practical applications and fills an important gap in the literature.
>
> That said, our framework could naturally be used to compare these sampling-based sublinear time algorithms alongside greedy and GW, efficiently determining instances where the sublinear time algorithms offer a comparable approximation to the more expensive algorithms and are thus preferable. Our results also don’t make restrictive assumptions on graph structure (e.g., density), unlike the sublinear approximation methods referenced, which we view as a strength.
>
> The large body of sublinear‑time clustering research (e.g., Mishra et al ’01; Czumaj, Sohler ’06; Feldman et al. ’13; Chen et al. ’22) also pursues a different goal from our paper: approximating the optimal clustering cost under specific cost models (e.g., k‑means or k‑center), often with structural assumptions (bounded dimension, cluster separation) and via uniform sampling or coreset constructions. We discuss our cost model and why head-to-head comparison with coresets is impossible further in lines 79-87.
>
> **W5:  typo in figure legend of Fig. 1(a)**
>
> Thank you for catching this. We will correct the legend in the camera-ready version.
>
> **Q1: The timing performance of both the subsampled and the original algorithms should be reported in the experiment.**
>
>
> Thank you for your suggestion! In combination with your suggestion in W1, we are scaling up our experiments and including runtime figures in addition to the accuracy or weight convergence plot in the Experiment section of the camera-ready version. Please refer to W2 for preliminary solve times experiment and analysis.
>
> We sincerely thank the reviewer for their thoughtful comments, which have helped us clarify and strengthen our work. In the camera-ready version, we will incorporate the following revisions to the Experiment section:
>
> - Correct the typo in Fig. 1(a) legend (W5)
> - Scale up the clustering experiments (W1)
> - Include runtime statistics for all subsample sizes with new figures (Q1)
> - Expand related work on sublinear‑time algorithms (W4)
>
> Please let us know whether we have addressed your concerns and what can be further improved; we are happy to continue the discussion anytime until the end of the discussion period. We respectfully ask that if you feel more positively about our paper, to please consider updating your score. Thank you!

---

> > ### Comment · Reviewer_uRvU · 2025-08-03
> > **Feedback**
> >
> > Thanks for your detailed rebuttal. However, I still feel the paper has some drawbacks that prevent me to increase the score.
> >
> > 1. Regarding W1. First, I don't quite appreciate to add more experiments to the "core experimental part" in the rebuttal stage. I mean, the purpose of rebuttal is to explain some missing details or add some ablation experiments so as to address the reviewer's concerns, NOT expand the core experimental part. Second, the current added experiments on clustering are still too simple. "n" is just several thousands and the used datasets are also very simple.
> >
> > 2. Regarding W4. I know your purpose is to estimate the performance, differing with the original goal of sublinear time algorithms. However, the sublinear time benefit of those algorithms can be adapted to quickly estimate the performance. Btw, the sublinear time clustering algorithms usually do not need structural assumptions.

---

> ### Author Response · Authors · 2025-08-05
>
> Thank you for carefully reading our responses.
>
>
> **W1**: We understand your preference not to expand the “core” experimental section during the rebuttal. Rather, our goal was to clarify that our results are not tied to the small-$n$ or $k = 2$ regime. Importantly:
> - We emphasize that our main contributions are *theoretical*. Our theory applies to arbitrarily large $n$ (dataset size), with sample size $m$ independent of $n$ in several cases (Theorems 2.2, 2.7, 3.4). As such, our scalability claim is based on provable bounds, not only on empirical scaling trends.
> - The goal of these simulations is to validate our guarantees in a controlled setting, as is common in theory-focused NeurIPS papers, since the primary focus is analytical. Datasets of these sizes are common in such papers [Shinde, Narayanan, Saunderson ’21; Yau, Karalias, Lu, Xu, Jegelka ’24], as are the specific datasets we chose (e.g., MNIST, CIFAR-10, Omniglot) [Balcan, Dick, Lang ’19; Baykal, Dikkala, Panigrahy, Rashtchian, Wang ’22]. We welcome the reviewer’s suggestions on specific datasets and scales that would be most convincing.
>
>
> **W4**: We appreciate this perspective and agree that ideas from sublinear-time algorithms (e.g., sketching, coresets) are highly relevant. We will make this connection explicit in our Related Work section.
>
> Our contribution, however, targets a different objective: **instance-specific algorithm selection**. This objective is largely orthogonal to the traditional goals of sublinear-time methods. While sublinear approaches can yield fast approximations for a single target problem, they are not designed to **evaluate or compare a given set of candidate algorithms**.
>
> We also note that Reviewer 74EK emphasized in their latest response that this ability to evaluate candidate algorithms **"is the key feature of the proposed *size generalization* scenario"**; we refer you to their illustrative example. Adapting sublinear tools to this evaluation/selection objective is non-trivial for several reasons:
>
>
> **Max-cut**: We emphasize that our goal is to estimate the empirical performance of the Goemans-Williamson (GW) algorithm—the state-of-the-art approximation algorithm—and the greedy algorithm, a classic heuristic with very different trade-offs.
>
>
> We did not aim to estimate the performance of sublinear-time approximation algorithms. Although this is an interesting direction, it is outside the scope of our work.
>
>
> That said, the approaches are complementary: in future work, sublinear-time approximation algorithms could be included as additional candidates in our selection framework, and bounds could be developed to determine the optimal probe size needed to compare their empirical performance to GW or greedy.
>
>
> **Clustering**: We emphasize that the critical distinction between our work and the literature on sublinear-time clustering is that these works target specific clustering objectives (k-means, k-median, k-center) [Bachem, Lucic, Hassani, Krause ’16; Har-Peled, Mazumdar ’18; Cohen-Addad, Saulpic, Schwiegelshohn ’22]. Crucially, our framework works with arbitrary/adversarial ground truth, which we do not assume minimizes any known clustering objective. Thus, it is impossible to provide a head-to-head comparison.
>
>
> Once again, we appreciate your thoughtful comments. Please let us know if you have any further questions or suggestions before the end of the rebuttal period, and we will be happy to address them.

---

> > ### Comment · Reviewer_uRvU · 2025-08-06
> > **Further dicussion**
> >
> > Thanks a lot for your response. Let me explain my concerns in detail. For simplicity, let us focus on the center based clustering part.
> >
> > 1. The proposed evaluation framework is only applicable to the clustering algorithms who use a selection function "f" (I call it "f-guided algorithm" for short)，not all possible clustering algorithms. However, for those greedy selection clustering algorithms, k-means++ and Gonzalez are very widely accepted in practice (I mean there is not much value to consider other types of "f").
> >
> > 2. In the main theoretical result Theorem 2.2, the sample complexity depends on a new parameter $\zeta$, which could be large. Not only this, there are two issues about this parameter. First, in a scenario we want to keep a time complexity independent of n, we cannot provide a relatively accurate estimation for $\zeta$, but in Algorithm 1, we do need this value as the input. Second, this parameter depends not only the input distribution, but also the function "f", that means, for different f-guided clustering algorithms, the value of $\zeta$ could be arbitrary (this makes the theorem not that elegant; for example, if you want to evaluate 1000 different "f"s, you should consider an upper bound of all these 1000 different $\zeta$s).
> >
> > 3. I feel sorry that I made a wrong statement in my last response. For those sublinear time clustering algorithms, they do not need assumption if we only consider the clustering cost. But if we consider the difference with ground truth, we may need some assumption, such as "clustering stability" (the authors are correct). But, I should emphasize that the proposed method in this paper actually also need an "implicit"  assumption: $\zeta$ is not too large, though the authors do not mention it explicitly.
> >
> > 4. I read the proofs in supplement. Here is a question that I forgot to mention before: In the proof of theorem 2.2 (line 1119), the equation relies on the fact that all the events within the expectation are independent. I am not sure whether this is correct, and suggest the authors to add some explanation.

---

> > > ### Author Response · Authors · 2025-08-06
> > > **Thank you for the feedback**
> > >
> > > Thank you for carefully engaging with our work and offering detailed feedback. Your comments have helped us clarify and strengthen the presentation of our results. Below, we directly address your points, highlighting how we’ll update the paper based on your feedback.
> > >
> > > > "The proposed evaluation framework is only applicable to the clustering algorithms who use a selection function 'f' [...], not all possible clustering algorithms. However, for those greedy selection clustering algorithms, k-means++ and Gonzalez are very widely accepted in practice."
> > >
> > > We agree our analysis doesn’t extend to all possible clustering algorithms (doing so would be beyond the scope of a single paper), but, as you note, it covers algorithms that are widely used in practice. We view this as a strength of our results.
> > >
> > > > If “we want to keep a time complexity independent of n, we cannot provide a relatively accurate estimation for $\zeta$, but in Algorithm 1, we do need this value as the input.”
> > >
> > > For both center-based clustering algorithms, we provide upper bounds on $\zeta$ that are independent of $n$: Theorems 2.4 and C.1 (summarized in lines 193–197). With the additional camera-ready page, we’ll move Theorem C.1 into the main body to emphasize this.
> > >
> > > To further clarify, $\zeta$—which measures how concentrated the center-selection distribution—is analogous to a condition number in numerical linear algebra: if $\zeta$ is small ("well-conditioned"), we obtain stronger guarantees. Moreover, our dependence on $\zeta$ is similar to the structure of parameterized algorithms, whose complexity or performance depends on a parameter that isolates the problem’s "hardness". For instance, in the classic online paging problem [Albers et al., STOC'02], the competitive ratio depends on a parameter that measures how many distinct page requests might appear within a given window, characterizing how "nice" the request sequence is. These measures of difficulty are often unknown a priori.
> > >
> > > We’ll explicitly make this connection clear by adding the following after line 78: “Our bounds depend on a structural parameter which functions similarly to a condition number in numerical analysis or a structural parameter in parameterized algorithms: it captures how ‘well-conditioned’ the instance is.”
> > >
> > > Finally, to highlight the parameterized nature of our results, we’ll restructure the main theorems so they hold for an arbitrary number of samples $m$, with the resulting error bounds expressed in terms of $\zeta$. For example, the bound in Theorem 1 will become $E[cost_{S'}(\mathcal{G})] - E[cost_{S}(\mathcal{G})] = O\left(k \cdot \exp\left(-\frac{m}{\zeta_{k,f}(\mathcal{X})}\right)\right),$
> > > making the dependence on the parameter $\zeta$ explicit, as is common in parameterized analysis.
> > >
> > > > “This parameter depends [on] the function 'f' [...] if you want to evaluate 1000 different "f"s, you should consider an upper bound of all these 1000 different $\zeta$s).”
> > >
> > >
> > > There’s no requirement to use a single, uniform bound across all functions $f$—to clarify this, we’ll add the following after the proof of Theorem 2.2: "If multiple selection functions are evaluated, there is no requirement to use a single uniform bound on $\zeta$; instead, one can choose a tailored sample size independently for each algorithm."
> > >
> > >
> > > > "The proposed method [has] an ‘implicit’ assumption: $\zeta$ is not too large, though the authors do not mention it explicitly.”
> > >
> > > We note that this dependence on $\zeta$ is explicitly stated in the submission:
> > > - Lines 158-161: “Our bound depends linearly on $\zeta_{k,f}(\mathcal{X})$, a parameter that quantifies the smoothness of the distribution over selected centers when sampling according to $f$ in $\mathsf{Seeding}$. Ideally, if this distribution is nearly uniform, $\zeta_{k,f}(\mathcal{X})$ is close to 1. However, if the distribution is highly skewed towards selecting certain points, then $\zeta_{k,f}(\mathcal{X})$ may be as large as $n$.”
> > > - Lines 224-225: “Higher $\zeta_{k, \mathsf{SL}}(\mathcal{X})$ indicates greater sensitivity to subsampling, while a lower value suggests robustness.“
> > >
> > > However, we understand your point that this could be clearer. We'll add the following after Theorem 2.2: "Our bounds explicitly rely on the structural parameter $\zeta$. Instances with small $\zeta$ yield stronger guarantees, while large values of $\zeta$ imply weaker guarantees.”
> > >
> > > > "In the proof of Theorem 2.2 [...], the equation relies on the fact that all the events within the expectation are independent. I'm unsure whether this assumption is correct and suggest the authors add clarification."
> > >
> > > Thanks for carefully reviewing our proofs. Fresh samples are used to estimate the expectation (line 1119), ensuring independence. To make this clearer in Theorem 2.2, we will revise the phrase “using $O(k{{\epsilon'}}^{-2}\log(\delta^{-1}))$ ground-truth queries” to “using ground-truth queries on $O(k{{\epsilon'}}^{-2}\log(\delta^{-1}))$ fresh samples from $\mathcal{X}$.”

---

> > > > ### Comment · Reviewer_uRvU · 2025-08-07
> > > >
> > > > Thanks for your quick response. I will leave more discussion to the stage of Reviewer-AC Discussions.

---

### Official Review · Reviewer_mGbx · 2025-07-02

**Clarity:** 3
**Significance:** 2
**Originality:** 2
**Rating:** 4
**Confidence:** 4

**Summary:**

This paper addresses the central challenge of **scalability in data-driven algorithm selection**. Data-driven algorithm selection is a powerful approach that evaluates candidate algorithms on a representative set of training instances and empirically chooses the best-performing one. However, running each algorithm on all training instances is computationally expensive, making it impractical for large or numerous instances.

To resolve this computational bottleneck, it is common practice to **evaluate algorithms on smaller surrogate instances** derived from the original input, but this practice has **lacked theoretical justification** until now .

This research provides the first theoretical foundation for this practice and formalizes a new concept called **"size generalization"** . Size generalization refers to the question (Question 1.1) of whether "the performance of an algorithm on large instances can be quickly estimated by evaluating it on smaller, representative instances subsampled from the original instance" .

The **main contributions** of this paper are as follows:

*   **Formalization and provision of a theoretical foundation for size generalization** :
    *   It introduces the new concept of "size generalization" to address the computational bottleneck in data-driven algorithm selection .
    *   It provides the first **formal basis and theoretical guarantees** for the concept that an algorithm's performance on large instances can be estimated by evaluating it on smaller, representative subsamples .
    *   This offers a natural framework for efficiently selecting algorithms for a single large problem instance, even when training instances are unavailable .
*   **Size generalization guarantees for key combinatorial optimization problems** :
    *   For **clustering problems**, it provides size generalization guarantees for three widely used algorithms: **single linkage, k-means++, and Gonzalez's k-center heuristic** .
        *   For center-based algorithms (smoothed variants of Gonzalez's k-center heuristic and k-means++), it shows that, under appropriate assumptions, the required sample size does not depend on the size of the original instance .
        *   For single linkage, it characterizes the conditions under which its performance can be estimated from subsamples, despite its theoretical instability .
    *   For the **Max-cut problem**, it analyzes size generalization guarantees for two representative algorithms: **Goemans-Williamson (GW) and Greedy** .
        *   For the GW algorithm, it proves a convergence bound for the SDP objective value without assumptions on the graph structure, strengthening prior work . This demonstrates that GW's performance can be efficiently estimated using subgraphs .
        *   For the Greedy algorithm, building on prior work, it identifies the rate at which the cut density of a subgraph converges to that of the full graph .
*   **Characterization of sufficient subsample sizes** :
    *   It characterizes the sufficient subsample size for the performance on a subsample to reflect the performance on the complete instance .
*   **Experimental validation** :
    *   Experiments in both clustering and max-cut problems validate that the best-performing algorithm on subsamples can effectively predict the best algorithm on the full dataset . This demonstrates the proposed theoretical insights .

**Questions:**

The problem awareness and approach of this paper were highly intriguing. As pointed out in the third weakness, while the paper focuses on two major combinatorial optimization problems—clustering and Max-cut—it does not seem to provide a concrete direction for extending the framework to other combinatorial optimization problems such as integer programming or dynamic programming. Is that a correct understanding?

**Ethical Concerns:**

["NO or VERY MINOR ethics concerns only"]

**Final Justification:**

After reading the authors' rebuttal, the important concerns I had during the review have been resolved, so I have raised my evaluation by one level.

**Limitations:**

yes

**Paper Formatting Concerns:**

No major formatting issues were observed.

**Quality:**

3

**Strengths And Weaknesses:**

Strengths
Addressing computational bottlenecks in data-driven algorithm selection: This paper tackles the primary challenge in data-driven algorithm selection, where running candidate algorithms on all training instances becomes computationally expensive. It introduces a new concept called "size generalization", demonstrating that an algorithm's performance on large instances can be efficiently estimated by evaluating it on smaller, representative instances. This significantly enhances the efficiency of algorithm selection.

Establishing a theoretical foundation for size generalization and applying it to a wide range of algorithms: This work provides the first formal theoretical justification for the common practice of evaluating algorithms on smaller proxy instances. Specifically, it presents size generalization guarantees for three widely used clustering algorithms: single linkage, k-means++, and Gonzalez's k-centers heuristic, as well as for two canonical Max-cut algorithms: Goemans-Williamson (GW) and Greedy. Furthermore, it strengthens prior work by providing convergence guarantees for the Goemans-Williamson SDP for Max-cut without imposing any assumptions on the graph structure.

Comprehensive empirical validation and reproducibility of theoretical findings: The theoretical discoveries in the paper are supported by extensive experiments using diverse datasets (MNIST, GM, Omniglot, noisy circles) and random graphs (Erdös-Réyni, Barbell, random geometric, Barabási-Albert). Experimental results show that as sample size increases, the accuracy or cut density of proxy algorithms converges to the performance of the original algorithms, highlighting the alignment between theory and practice. Additionally, code to reproduce the main experimental results is provided, ensuring the transparency and reproducibility of the research.

Weaknesses
High sample complexity or theoretical instability in some algorithms:

Gonzalez's k-centers heuristic: Directly applying the framework to this algorithm poses a challenge due to its deterministic nature leading to high sample complexity (ϑk,f(X) = n). While a smoothed variant (SoftmaxCenters) is introduced to address this, it creates a trade-off with the algorithm's strict approach.

Single linkage: This algorithm is known to be theoretically unstable, and its sample complexity for size generalization (Theorem 2.7) depends on factors such as cluster separability (ϑk,SL(X)) and minimum cluster size, potentially requiring large subsamples when these values are high.

Limitations regarding algorithm-specific assumptions:

Max-cut Greedy algorithm: The analysis of the Greedy algorithm is limited to unweighted graphs.

Goemans-Williamson (GW) algorithm: While the SDP objective value converges, the distribution of GW cut values after random rounding can differ even with different optimal SDP solutions, indicating non-Lipschitz behavior between the SDP solution and the GW cut value. This implies that estimating the cut value itself, in addition to the SDP value, involves both multiplicative and additive errors.

Limited scope of combinatorial optimization problems addressed: This paper focuses on two major combinatorial optimization problems: clustering and Max-cut. While it establishes a powerful framework for size generalization, its extension to other combinatorial optimization problems such as integer programming and dynamic programming is mentioned as "interesting future directions", implying that its current applicability is limited.

---

> ### Author Rebuttal · Authors · 2025-07-31
>
> We thank the reviewer for their review and detailed summary of our work. Below, we address their comments.
>
> **W1: trade-off characterized by smoothed Gonzalez's k-centers heuristic**
>
> > Directly applying the framework … poses a challenge due to its deterministic nature, leading to high sample complexity (ϑk,f(X) = n). While a smoothed variant (SoftmaxCenters) is introduced to address this, it creates a trade-off with the algorithm's strict approach.
>
> In Section 2.1, we analyze the exact trade-off that the reviewer has pointed out. Indeed, this trade-off is the focus of our analysis in Section 2.1 (Lines 172–192), where we show exactly how smoothing (i.e., the choice of $\beta$) affects both the sample complexity and the approximation quality.
>
> In Theorem 2.3, we provide an approximation guarantee for the k-centers objective of SoftmaxCenters, showing that this smoothed variant has nearly the same approximation ratio as Gonzalez’s k-centers heuristic. Empirically (Figure 7, Appendix C.1), we further demonstrate that even for vanishingly small $\beta$, SoftmaxCenters outperforms Gonzalez’s algorithm, achieving lower objective values. This behavior is consistent with prior observations that smoothing can improve heuristic performance (García-Díaz et al. ’17).
>
> To make these results more visible, we will use our extra camera-ready page to move these experimental results into the main text, providing a compelling demonstration of the performance of SoftmaxCenters.
>
> **W2: SL potentially requiring large subsamples**
>
> > “[Single linkage] is known to be theoretically unstable, and its sample complexity for size generalization (Theorem 2.7) depends on factors such as cluster separability (ϑk, SL(X)) and minimum cluster size, potentially requiring large subsamples when these values are high.”
>
> It is well known (and we explicitly cite in lines 198-199) that single linkage can be unstable when clusters are poorly separated. Our Theorem 2.7 does not introduce this limitation—it precisely characterizes the stability of SL to deletions of individual datapoints.
> Moreover, as we describe in lines 238-240, in Appendix C.3, we show that our sample complexity bounds are tight. To ensure these insights are front and center, we will move both the theorem statement and its matching lower bound into the main body of the camera‐ready version. We believe that clarifying the exact regimes in which SL remains stable versus when it becomes deletion‐sensitive represents a valuable contribution of this section.
>
>
> **W3: Analysis of the Greedy algorithm is limited to unweighted graphs**
>
> Indeed, our max‑cut Greedy analysis (Theorem 3.1) assumes unweighted graphs. However, this assumption is standard in the literature (e.g., Feige, Karpinski, Langberg ’02; Mathieu, Schudy ’08; Fotakis, Lampis, Paschos ’15;  Wang, Wu, Zuo ’24) and clearly stated in Sec. 3.2. Extending to general weights is orthogonal to our main goal of illustrating distinct size‑generalization behaviors across algorithmic paradigms. We will add the references mentioned here to our camera-ready version to more clearly illustrate this point.
>
> **W4: non-Lipschitzness of GW objective leading to multiplicative and additive errors**
>
> > While the SDP objective value converges, the distribution of GW cut values after random rounding can differ even with different optimal SDP solutions, indicating non-Lipschitz behavior between the SDP solution and the GW cut value. This implies that estimating the cut value itself, in addition to the SDP value, involves both multiplicative and additive errors.
>
> We see this not as a weakness but rather a summary of Theorem 3.3. Rather than a drawback, proving that no purely multiplicative bound exists—and then developing a combined multiplicative and additive estimator—is a technical advancement (Sec. 3.3). If the GW cut value were Lipschitz in the SDP solution, a purely multiplicative guarantee would be trivial. Our work tackles the hard, non‑Lipschitz regime. Thus, the fact that we are able to provide a bound **despite** this non-Lipschitzness is a strength of our analysis, not a weakness of our result.
>
> **W5: The current application does not include what’s listed in the future directions**
>
> > “Extension to other combinatorial optimization problems, such as integer programming and dynamic programming, is mentioned as 'interesting future directions', implying that its current applicability is limited.”
>
> We respectfully disagree. We already cover **five** canonical methods (three clustering, two max‑cut) to demonstrate our framework’s breadth (Sec. 1.1). Proposing future directions—integer programming and dynamic programming among them—is a standard practice to inspire follow‑up work and does not diminish our current contributions.
>
> **Q1: Extension to other combinatorial problems**
>
> In our paper, we deliberately chose a diverse set of canonical algorithms—ones that perform well in practice. These algorithms range from SDP‑based methods (Goemans–Williamson) to purely combinatorial heuristics (Gonzalez’s k‑centers, greedy max‑cut), showcasing size‑generalization behavior across fundamentally different paradigms and providing a proof-of-concept to guide future research. We view this breadth as one of our paper’s strengths.
>
> Moreover, we believe some of the methods developed in this paper can extend to a wider class of combinatorial optimization problems. For instance, the primal–dual analysis used for Max‐Cut SDP should apply to any SDP formulation with strong duality (e.g., correlation‐clustering rounding, sensor‐network localization).
>
> That said, we fully agree that a general theoretical toolkit for size generalization would greatly inspire future work. We will make this and the SDP extensions explicit in the Conclusion by adding:
>
> *Future work includes extending size‐generalization guarantees to broader classes of optimization problems, such as integer programming and additional SDP‐based algorithms (e.g., randomized rounding for correlation clustering). Another exciting direction for future work is to formulate a unified theoretical toolkit for size generalization—one that prescribes, given a new combinatorial algorithm, which structural properties to verify to guarantee that its performance on large instances extrapolates from small ones.*
>
>
> We hope this clarifies that each point raised either restates known properties or actually highlights our innovations—the $\zeta$‑smoothness mechanism, tight SL bounds, and the novel treatment of non‑Lipschitz rounding. We thank the reviewer again for their feedback and would be happy to discuss further.

---

> > ### Comment · Reviewer_mGbx · 2025-08-04
> >
> > Thank you for the authors' rebuttal. The clear responses to the points commented by the reviewer have clarified the points that the reviewer had misunderstood, and we have received confirmation that unclear points will be addressed in the camera-ready version. Since the initial concerns have been resolved, I am raising my evaluation by one level from 3 to 4.

---

> > > ### Author Response · Authors · 2025-08-04
> > >
> > > Thank you! We are glad that we addressed your concerns. Please let us know if you have any further questions or suggestions before the end of the rebuttal period, and we will be happy to address them.

---

### Official Review · Reviewer_74EK · 2025-07-03

**Clarity:** 4
**Significance:** 3
**Originality:** 3
**Rating:** 4
**Confidence:** 3

**Summary:**

This paper introduces and tackles the task of "size generalization", which is estimating algorithms' performance on a large dataset by running it on a much smaller subset of instances. This task is meaningful because it provides a principled way to accelerate algorithm selection, which would be extremely costly if every candidate algorithm is run on every instance. Specifically, the authors provide size generalization guarantees for three clustering algorithms and two max-cut algorithms.

To model the generalization behaviour of k-means++ and Gonzalez’s k-centers heuristic, the paper builds an MCMC-based surrogate and proves a sample bound in terms of a smoothness parameter $\varsigma_{k,f}$ that quantifies how sharply the seeding distribution concentrates. Notably, the sub-sample size needed to obtain the bound is independent of the total sample size. To analyse linkage-based clustering algorithms, the paper tracks how min–max path distances shift under subsampling and introduces a separation parameter $\varsigma_{k,\mathrm{SL}}$; size generalization holds when this remains moderate. For Goemans–Williamson max-cut, the paper compares the SDP objective on an induced subgraph to that on the full graph via a dual-pruning argument, then obtains accuracy guarantees. For the Greedy differencing heuristic, the paper casts the evolving cut weight as a martingale, applies concentration inequalities, and shows that sampling $t = O(\varepsilon^{-2})$ vertices suffices to approximate the full-graph cut within $O(\varepsilon)$.

Experiments on four image or synthetic datasets and four random-graph families confirm that the proxy algorithms’ accuracies and cut densities rapidly approach those on the full instances, often allowing the preferred heuristic to be identified from a tiny fraction of the data.

**Questions:**

- How applicable are the size generalization bounds and proving techniques on other combinatorial problems and algorithms? Is there a more general framework we can extract out of the separated bounds and proofs in this paper?

- Can the bounds developed in this paper guide us in selecting sub-sample sizes in real-world scenario? Is there a good way to decide or guess the smoothing parameters or other data assumptions?

**Ethical Concerns:**

["NO or VERY MINOR ethics concerns only"]

**Final Justification:**

I will keep my rating of 4. Reasons explained in another comment.

**Limitations:**

Please see the weakness section.

**Quality:**

3

**Strengths And Weaknesses:**

Strengths:
1. The task of size generalization formalizes a indeed very practical question. In fact, most people in the field of machine learning test their models on small datasets, and implicitly assumes size generalization in most times. It is good to see efforts in providing theoretical supports of size generalization, first on combinatorial algorithmic tasks.

2. The paper develops several set of tools for obtaining size generalization bounds, which can be inspiring to more combinatorial algorithms. However, I am curious whether these techniques are directly applicable to a larger set of problems and algorithms.

Weaknesses

1. All size generalization bounds in the paper rely on some data assumptions, such as the smoothness parameters $\vartheta_{k,f}$ and $\vartheta_{k,\mathrm{SL}}$, yet the authors provide little guidance on how to estimate or upper bound these quantities before running experiments. This limitation prevents people from directly using the bounds to plan computation budgets in real world scenarios. For example, when the data condition is unknown, I wonder if there is a naive workaround with a probe and stop strategy: run the target algorithm on randomly selected subsets of size 1, 2, 4, 8, 16, \dots, compare the objective values or algorithm rankings between consecutive subset sizes, and terminate when the performance curve stabilises within a predefined tolerance. If we can prove the objective converges before reaching certain size, then this empirical scheme can act as a pragmatic substitute for the theoretical sample complexity.

2. In this paper, basically each algorithm needs a set of new analysis. This approach is intrinsically limited in scope, because extending size generalization results to more / unknown combinatorial algorithms will likely require a more general framework.

---

> ### Author Rebuttal · Authors · 2025-07-31
>
> Thank you very much for your careful reading and encouraging feedback. We’re delighted that you find our formalization of size‑generalization both practically meaningful—“people in the field of machine learning test their models on small datasets, and implicitly assume size generalization most of the time”—and “inspiring to more combinatorial algorithms.” Your questions also led to an exciting new experiment that we will include in the camera-ready version. Below, we address each point.
>
> **W1: “A pragmatic substitute for the theoretical sample complexity”**
>
> This is a great point! We agree that an adaptive “probe‐and‐stop” strategy could serve as a pragmatic complement to our sample‐complexity bounds. To validate this idea, we ran preliminary experiments on four max‐cut problems (random geometric, Erdős–Rényi, barbell, and Barabási–Albert graphs) with $n=100$ nodes. For each instance, we drew 2, 4, 8, …, 64 nodes, solved the problem using both Greedy and Goemans–Williamson (GW), and averaged the objective over 10 trials. In every case, the relative ranking of Greedy vs. GW stabilized by 8–16 samples:
>
> |           	| number of nodes sampled |  2 | 4 | 8 | 16 | 32 | 64 | 100 |
> |------------------|-----------|------------:|------------:|------------:|-------------:|-------------:|-------------:|----------:|
> | **Random Geometric**| Greedy| 1.0 |     3.6 |   15.5 |   61.8 |    244.2 |   972.7 |	2381.8 |
> |                     	| GW    |   1.0 |   3.6 |    15.2 |   60.0 |    239.6 |   997.3 |	2376.9 |
> |                     	| **Best**  | Greedy, GW  | Greedy, GW  | 	Greedy  | 	Greedy   | 	Greedy   |      	GW  |	Greedy  |
> | **Erdős–Rényi**     	| Greedy	|     	0.0 |     	1.0 |     	4.0 |      	8.4 |     	40.9 |    	104.6 | 	323.1 |
> |                     	| GW    	|     	0.0 |     	1.0 |     	5.0 |      	9.0 |     	43.7 |    	140.6 | 	339.8 |
> |                     	| **Best**  | Greedy, GW  | Greedy, GW  |    	GW   |    	GW	|    	GW	|    	GW	|  	GW	|
> | **Barbell**         	| Greedy	|     	0.4 |     	2.1 |     	7.1 |     	29.6 |    	126.3 |    	511.5 |	1254.7 |
> |                     	| GW    	|     	0.4 |     	2.1 |     	6.5 |     	28.2 |    	122.9 |    	506.5 |	1249.5 |
> |                     	| **Best**  | Greedy, GW  | Greedy, GW  | 	Greedy  | 	Greedy   | 	Greedy   | 	Greedy   |	Greedy  |
> | **Barabási–Albert** 	| Greedy	|     	0.2 |     	0.3 |     	3.2 |     	11.5 |     	34.3 |    	130.1 | 	307.6 |
> |                     	| GW    	|     	0.2 |     	0.3 |     	3.3 |     	12.9 |     	36.8 |    	141.2 | 	328.2 |
> |                     	| **Best**  | Greedy, GW  | Greedy, GW  |    	GW   |    	GW	|    	GW	|    	GW	|  	GW	|
>
> We will include these results and a brief discussion in the camera-ready version and continue scaling to more instances.
>
> **W2 & Q1. On “extending size generalization results to more combinatorial algorithms [with] a more general framework”**
>
> Thank you for raising the question of whether a unified framework guides the analysis of size generalization. In our paper, we deliberately chose a diverse set of canonical algorithms—ones that perform well in practice. These algorithms range from SDP‑based methods (Goemans–Williamson) to purely combinatorial heuristics (Gonzalez’s k‑centers, greedy max‑cut), showcasing size‑generalization behavior across fundamentally different paradigms and providing a proof-of-concept to guide future research. We view this breadth as one of our paper’s strengths.
>
> Moreover, we believe some of the methods developed in this paper can extend to a wider class of combinatorial optimization problems. For instance, the primal–dual analysis used for Max‐Cut SDP should apply to any SDP formulation with strong duality (e.g., correlation‐clustering rounding, sensor‐network localization).
>
> That said, we fully agree that a general theoretical toolkit for size generalization would greatly inspire future work. We will make this and the SDP extensions explicit in the Conclusion by adding:
>
> *Future work includes extending size‐generalization guarantees to broader classes of optimization problems, such as integer programming and additional SDP‐based algorithms (e.g., randomized rounding for correlation clustering). Another exciting direction for future work is to formulate a unified theoretical toolkit for size generalization—one that prescribes, given a new combinatorial algorithm, which structural properties to verify to guarantee that its performance on large instances extrapolates from small ones.*
>
> **Q2: Ways to “decide or guess the smoothing parameters”**
>
> This point relates closely to W1—please also see our response there for an implementation of the reviewer’s subsampling method. Moreover, if by “smoothing parameter” you are referring to the $\zeta$ terms in our sample complexity bound, Theorem 2.4 shows that they can be bounded in terms of the data radius and the choice of $\beta$. Likewise, Theorem C.1 (see Line 193 for the forward reference in the main body) shows, under a high‑dimensional Gaussian data model, that one can bound the term for $k$-means++. Single‐linkage is more challenging due to its instability; in practice, one could estimate $\zeta$ from a pilot sample. If we have misunderstood your question, please let us know, and we will be happy to clarify further during the discussion period.
>
> We sincerely thank you again for your detailed review and thoughtful questions. In the camera-ready version, we will incorporate:
>
> - The probe‑and‑stop subsampling experiment (W1)
> - Explicit future‑work directions on a unified toolkit and SDP extensions (W2, Q1) in the Conclusion
>
> Please let us know whether we have addressed your concerns and what can be further improved; we are happy to continue the discussion anytime until the end of the discussion period. We respectfully ask that if you feel more positively about our paper, to please consider updating your score. Thank you!

---

> > ### Comment · Reviewer_74EK · 2025-08-05
> >
> > Thank you for your detailed rebuttal and extra experiments!
> >
> > The author said, "We view this breadth as one of our paper’s strengths." I agree that, as the first paper analysing the size generalization of combinatorial algorithms, this paper analyses a set of five algorithms, makes use of a selection of techniques, and provides insights into analysing the algorithm selection strategy for combinatorial tasks. **In this regard, I tend to keep my positive rating. I strongly encourage the authors to extend their discussion of how the proposed techniques can be reused and applied to a broader set of tasks and a broader class of algorithms.**
> >
> > Reviewer uRvU raises concern about sublinear algorithms for k-means, k-center, and max-cut. While I agree with Review uRvU that sub-linear algorithms naturally provide fast approximate solutions to the given tasks, I believe there is a fundamental difference. **Sub-linear algorithms cannot be used to evaluate a set of provided algorithms. I think this is the key feature of the proposed "size generalization" scenario.** (The authors replied Review uRvU by treating sub-linear algorithms as candidate algorithms, which I think misses the real concern). For example, a set of real-world problems is abstracted to a set of k-means instances, with special structures. Suppose that some real-world heuristics are also abstracted into k-means candidate algorithms, which are provided with size generalization bounds. Then, we can compare these provided algorithms on small sub-sampled instances with generalization guarantees to larger ones. Please correct me if I am wrong on this example and interpretation.
> >
> > In summary, I will keep my rating of 4. I think this paper raises the interesting problem of size generalization of combinatorial tasks, and presents a few initial examples. The reason for not raising my rating is that I think with the current techniques and results, size generalization is still far from achieving the above-described example and become actually useful.

---

> > > ### Author Response · Authors · 2025-08-05
> > >
> > > Thank you for carefully reading our responses. We agree with your perspective on sublinear-time approximation algorithms. Indeed, they "cannot be used to evaluate a set of provided algorithms. [This is] the key feature of the proposed *size generalization* scenario". Your example illustrates exactly the motivation behind the size-generalization guarantees we study, and we appreciate you highlighting this strength of our framework.
> > >
> > > We highly value the idea of developing a generalizable toolkit for size generalization and will make sure to discuss this in the camera-ready version! Thank you again for your time!

---

### Note · Authors · 2025-08-12

We thank the reviewers for their thoughtful feedback and engagement. We are encouraged that, across both the initial review and the rebuttal, they found our contributions valuable. Our key strengths:

- **Clear, novel motivation**: Our work formalizes a “very practical question” (74EK), “provides the first theoretical foundation” (mGbx), “has practical significance in many scenarios” (uRvU), and “fills a nice gap in the literature”—“a significant strength” (orjk).
- **Diverse theoretical tools**: Reviewers noted “several sets of tools… [inspiring] more combinatorial algorithms” (74EK) and our treatment of “three widely used clustering algorithms and two max-cut algorithms” (uRvU).
- **Strong results and technical insight**: The “proof techniques are quite nice,” and the conclusions align with intuition from related work (orjk).

Reviewers asked whether our techniques generalize to other combinatorial problems. We chose five algorithms—SDP-based and purely combinatorial—for their strong empirical performance and methodological diversity. This variety helps reveal size generalization behavior across fundamentally different approaches. Moreover, we believe that parts of our analysis, such as the primal–dual framework for the Max-Cut SDP, could extend naturally to other SDP formulations with strong duality (e.g., correlation clustering, sensor network localization).

Reviewer uRvU asked about connections to sublinear-time algorithms. As noted in the rebuttal, our goals differ fundamentally; 74EK’s example (in their comment) captures this distinction. We will make this difference more explicit in the revision.

We have polished our paper significantly in response to the comments:
1. **Experiments**: we added experiments on larger, more heterogeneous datasets, runtime statistics (uRvU), experiments with a pragmatic subsampling scheme (74EK), and a sublinear sample-size demo (uRvU).
2. **Related Work**: we clarified ties to sublinear-time algorithms.
3. **Conclusion**: we added a discussion on the importance of developing a size-generalization toolkit and; sketches of extensions to other combinatorial problems.

Overall, our paper formalizes and analyzes *size generalization*—a practically useful yet theoretically underexplored notion in learning for combinatorial optimization. We believe these contributions lay a foundation for extending this notion to a wider array of settings. We thank the reviewers once again for the helpful suggestions and discussions.

---

### Decision · Program_Chairs · 2025-09-17

**Decision:**

Accept (spotlight)

**Comment:**

This paper tackles the primary challenge in data-driven algorithm selection, where running candidate algorithms on all training instances becomes computationally expensive. It introduces a new concept called "size generalization", demonstrating that an algorithm's performance on large instances can be efficiently estimated by evaluating it on smaller, representative instances. This significantly enhances the efficiency of algorithm selection. The paper addresses a handful of comobinatorial optimization partition problems w,r,t, some common optimization algorithms. The main criticism of the submission refers to the relatively limited scope of the analysis.